# Provable Convergence of Single-Timescale Neural Actor-Critic in Continuous Spaces

## Abstract

Actor-critic (AC) algorithms have been the powerhouse behind many successful yet challenging applications. However, the theoretical understanding of finite-time convergence in AC's most practical form remains elusive. Existing research often oversimplifies the algorithm and only considers simple finite state and action spaces. We analyze the more practical single-timescale AC on continuous state and action spaces and use deep neural network approximations for both critic and actor. Our analysis reveals that the iterates of the more practical framework we consider converge towards the stationary point at rate $\widetilde{\mathcal{O}}(T^{-1/2}) + \widetilde{\mathcal{O}}(m^{-1/2})$, where $T$ is the total number of iterations and $m$ is the width of the deep neural network. To our knowledge, this is the first finite-time analysis of single-timescale AC in continuous state and action spaces, which further narrows the gap between theory and practice.

## 1 Introduction

Actor-critic (AC) algorithms have driven numerous successful applications and are state-of-the-art in reinforcement learning (Konda & Tsitsiklis, 1999; Mnih et al., 2016; Silver et al., 2017). Their practical implementation typically consists of two parallel updates: the critic update and the actor update. The critic incrementally estimates the action-value function for the current policy, while the actor adjusts the policy network in the direction suggested by the estimated policy gradient based on the action value.

Despite AC's widespread success, their theoretical understanding lags significantly behind. Most existing theoretical results focus on cases where the actor and the critic update at significantly different rates. These include algorithms that either update the critic multiple times for a fixed actor (Yang et al., 2019; Kumar et al., 2019; Agarwal et al., 2021; Xu et al., 2020a) or employ two-timescale approaches where the actor's stepsize decays faster than the critic's (Wu et al., 2020b; Chen et al., 2023; Xu et al., 2020b; Hong et al., 2023). These settings are only made to simplify analysis. In practice, the actor and critic are typically updated at a **single-timescale**, using stepsizes that are *constantly proportional* to each other (Chen et al., 2021; Olshevsky & Gharesifard, 2023; Chen & Zhao, 2024; Tian et al., 2024). Single-timescale AC is typically more sample-efficient, as it avoids artificially slowing down the actor update performed in the aforementioned AC variants (Olshevsky & Gharesifard, 2023; Chen & Zhao, 2024).

However, the theoretical analysis of single-timescale AC in *practical settings* is still largely missing in the literature. As shown in Table 1, all existing works only analyze the single-timescale AC method in solving Markov Decision Processes (MDPs) with *finite action space*. This finite action space assumption excludes all continuous policies, including commonly used Gaussian, Uniform, and Gamma policies. Given the commonness of continuous control tasks in practice and the prevalence of AC algorithms in addressing them (Lillicrap et al., 2015; Haarnoja et al., 2018), there is a pressing need for theoretical guarantees in continuous settings. Moreover, Markovian sampling and deep neural network approximation for both the actor and the critic are commonly used in practical applications (LeCun et al., 2015; Haarnoja et al., 2018). However, existing studies have typically addressed only one of these elements, failing to consider their compound effects in practice (see the summary in Table 1).

Table 1: Comparisons of existing works analyzing single-timescale AC algorithms under various settings

| Reference | MDP | | Sampling | | Function class | Convergence rate | |
|---|---|---|---|---|---|---|---|
| | State | Action | Actor | Critic | | w.r.t. $T$ | w.r.t. $m$ |
| Chen et al. (2021) | Infinite | Finite | i.i.d. | i.i.d. | Linear | $\mathcal{O}(T^{-0.5})$ | N/A |
| Olshevsky & Gharesifard (2023) | Finite | Finite | i.i.d. | i.i.d. | Linear | $\mathcal{O}(T^{-0.5})$ | N/A |
| Chen & Zhao (2024) | Infinite | Finite | Markovian | Markovian | Linear | $\tilde{\mathcal{O}}(T^{-0.5})$ | N/A |
| Tian et al. (2024) | Finite | Finite | i.i.d. | Markovian | Deep NN | $\tilde{\mathcal{O}}(T^{-0.5})$ | $\tilde{\mathcal{O}}(m^{-0.5})$ |
| Ours | Infinite | **Infinite** | Markovian | Markovian | Deep NN | $\tilde{\mathcal{O}}(T^{-0.5})$ | $\tilde{\mathcal{O}}(m^{-0.5})$ |

As highlighted in the last row of Table 1, in this paper, we establish the finite-time convergence of single-timescale AC in solving MDPs with continuous (infinite) state and action spaces, and using deep neural network approximation and Markovian sampling for both actor and critic updates. Our analysis shows that the algorithm converges to a stationary point at a rate of $\widetilde{\mathcal{O}}(T^{-1/2}) + \widetilde{\mathcal{O}}(m^{-1/2})$, where $T$ is the number of iterations, $m$ is the neural network width, and $\widetilde{\mathcal{O}}$ hides logarithmic factors. As outlined in Table 1, previous studies faced at least two of the three potentially restrictive assumptions discussed earlier (finite action space, i.i.d sampling, linear function class). In contrast, our results address all these challenges, which bridge the gap between theory and practice and advance the theoretical analysis for the single-timescale AC method.

## 1.1 MAIN CONTRIBUTIONS

Our main contributions are summarised as follows:

• We establish the convergence of single-timescale AC in continuous state and action spaces, which has not been accomplished in prior research (see Table 1). Notably, even for the simpler case of the two-timescale AC variants, existing analysis cannot establish their convergence in the continuous setting. Our work may serve as the foundation to analyze other two- or single-timescale AC algorithms in more general continuous settings.

• Our results demonstrate significant advantages over existing works on single-timescale AC. We adopt more practical settings of deep neural network approximation and Markovian sampling for both the actor and the critic. Compared to Tian et al. (2024), where the critic employs Markovian sampling to collect transition tuples, the actor still requires i.i.d. transition tuples sampled from a discounted state-action occupation measure, which demands a burdensome re-sampling. In contrast, we allow Markovian sampling for both the actor and critic, utilizing the same transition tuples, closely following the state-of-the-art practice that facilitates efficient *online learning*.

• Technically, we develop a new framework to address the challenges posed by the continuous domain in single-timescale AC analysis. To establish the main results, we formulate a general condition in Assumption 4.7 (c) and demonstrate that it is satisfied by a broad class of neural network policies (Proposition 4.8) on continuous space, and include the previous assumptions on discrete space as special cases. Moreover, we examine the neural network approximation errors of the *evolving* actor and critic, ensuring that the resulting errors do not amplify through their interactions. Our methodology enriches the analytical toolbox for single-timescale AC.

**Notation.** We use san-serif letters to denote scalars and use lower and upper case bold letters to denote vectors and matrices respectively. We also use $\|\boldsymbol{\omega}\|$ to denote the $\ell_2$-norm of a vector $\boldsymbol{\omega}$, $\|\boldsymbol{A}\|$ to denote the spectral norm of a matrix $\boldsymbol{A}$, and $\|\boldsymbol{A}\|_{\mathrm{F}}$ to denote the Frobenius norm of a matrix $\boldsymbol{A}$. For two sequences of real numbers $(x_n)$ and $(y_n)$, we write $x_n = O(y_n)$ if there exists $C < \infty$ such that $|x_n| \leq C|y_n|$ for all $n$ sufficiently large. We use $\tilde{\mathcal{O}}(\cdot)$ to further hide logarithmic factors. The total variation distance of two probability measures $\mu$ and $\nu$ on $\mathcal{X}$ is defined by $d_{TV}(\mu, \nu) := \sup_A |\mu(A) - \nu(A)|$, where $A$ runs over all measurable subsets of $\mathcal{X}$. In addition, we use $\mathbb{P}$ to denote a generic probability of some random event.

## 2 PRELIMINARIES

In this section, we introduce some basics of MDP, the AC algorithm, and deep neural networks.

**Markov Decision Process.** We consider the standard Markov Decision Process (MDP) characterized by $(\mathcal{S}, \mathcal{A}, \mathcal{P}, r)$, where $\mathcal{S}$ is the state space and $\mathcal{A}$ is the action space. The spaces $\mathcal{S}$ and $\mathcal{A}$ are allowed to be either finite sets or real vector spaces, i.e., $\mathcal{S} \subset \mathbb{R}^{d_\mathrm{s}}$ and $\mathcal{A} \subset \mathbb{R}^{d_\mathrm{a}}$. The transition kernel is denoted by $\mathcal{P}(s_{t+1}|s_t, a_t) \in \mathbb{R}_{\geq 0}$ and the reward function is $r : \mathcal{S} \times \mathcal{A} \to [-U_r, U_r]$. A policy $\pi_{\boldsymbol{\theta}}$ parameterized by $\boldsymbol{\theta} \in \mathcal{X}_\Theta$ maps a given state to a probability distribution over the action space, i.e., $\boldsymbol{a}_t \sim \pi_{\boldsymbol{\theta}}(\cdot|\boldsymbol{s}_t)$. In this work, we consider the average-reward setting (Sutton et al., 1999; Yang et al., 2019; Wu et al., 2020b; Chen & Zhao, 2024), which aims to find a policy $\pi_{\boldsymbol{\theta}}$ that maximizes the following infinite-horizon time-average reward:

$$J(\boldsymbol{\theta}) := \lim_{T \to \infty} \mathbb{E}_{\boldsymbol{\theta}} \left[ \frac{1}{T} \sum_{t=0}^{T-1} r(s_t, a_t) \right] = \mathbb{E}_{(s,a) \sim (\mu_{\boldsymbol{\theta}}, \pi_{\boldsymbol{\theta}})} \left[ r(s, a) \right].$$

In the above equation, the expectation $\mathbb{E}_{\boldsymbol{\theta}}$ is taken over the states and actions generated by following the policy $\pi_{\boldsymbol{\theta}}$ and the transition kernel $\mathcal{P}$. Additionally, $\mu_{\boldsymbol{\theta}}$ denotes the stationary state distribution induced by $\pi_{\boldsymbol{\theta}}$ and $\mathcal{P}$. The existence of this stationary distribution is guaranteed by the uniform ergodicity of the underlying MDP, which is a common assumption (See Assumption 4.6 in the sequel). Hereafter, we refer to $J(\boldsymbol{\theta})$ as the time-average reward (and exchangeably, *performance function*), which can be evaluated by the expected reward over the stationary distribution $\mu_{\boldsymbol{\theta}}$ and the policy $\pi_{\boldsymbol{\theta}}$. The state-value function is used to evaluate the overall rewards starting from a state $s$, following policy $\pi_{\boldsymbol{\theta}}$ and transition kernel $\mathcal{P}$ thereafter, which is defined as

$$V_{\boldsymbol{\theta}}(s) := \mathbb{E}_{\boldsymbol{\theta}} \left[ \sum_{t=0}^{\infty} \big( r(s_t, a_t) - J(\boldsymbol{\theta}) \big) \Big| s_0 = s \right].$$

Similarly, we define the action-value (Q-value) function to evaluate the overall rewards starting from $s$, taking action $a$, and following transition kernel $\mathcal{P}$ and policy $\pi_{\boldsymbol{\theta}}$ thereafter:

$$Q_{\boldsymbol{\theta}}(s, a) := \mathbb{E}_{\boldsymbol{\theta}} \left[ \sum_{t=0}^{\infty} \big( r(s_t, a_t) - J(\boldsymbol{\theta}) \big) \Big| s_0 = s, a_0 = a \right] = r(s, a) - J(\boldsymbol{\theta}) + \mathbb{E} \big[ V_{\boldsymbol{\theta}}(s') \big],$$

where the last expectation is taken over $s' \sim \mathcal{P}(\cdot|s, a)$.

We denote the class of real-valued functions on $\mathcal{S}$ by $\mathcal{F} := \{f \mid f : \mathcal{S} \to \mathbb{R}\}$. For a policy $\pi_{\boldsymbol{\theta}}$, we define two operators $D_{\boldsymbol{\theta}} : \mathcal{F} \to \mathcal{F}$ and $P_{\boldsymbol{\theta}} : \mathcal{F} \to \mathcal{F}$ as follows:

$$D_{\boldsymbol{\theta}} f(s) = \mu_{\boldsymbol{\theta}}(s) \cdot f(s), \quad P_{\boldsymbol{\theta}} f(s) = \int_{\mathcal{S} \times \mathcal{A}} \pi_{\boldsymbol{\theta}}(a \mid s) \mathcal{P}(s' \mid s, a) f(s') \mathrm{d}(a \times s'). \tag{1}$$

These operators will be instrumental in addressing the technical challenge associated with continuous state and action space. Lastly, for two functions $f, g \in \mathcal{F}$, their inner product is defined as

$$\langle f, g \rangle = \int_{\mathcal{S}} f(s) \cdot g(s) \mathrm{d}s, \tag{2}$$

and the norm of $f$ is defined as $\|f\|^2 = \langle f, f \rangle$.

**Actor-Critic.** In AC, typically the critic estimates the actor's value through Temporal-Difference (TD) learning, and the actor adjusts its policy parameters to maximize the performance function via stochastic gradient ascent. The policy gradient theorem (Sutton et al., 1999) provides an analytical formula of the gradient of the performance function $J(\boldsymbol{\theta})$ with respect to the policy parameter $\boldsymbol{\theta}$, which is given by

$$\nabla_{\boldsymbol{\theta}} J(\boldsymbol{\theta}) = \mathbb{E}_{s \sim \mu_{\boldsymbol{\theta}}, a \sim \pi_{\boldsymbol{\theta}}} \big[ Q_{\boldsymbol{\theta}}(s, a) \cdot \nabla_{\boldsymbol{\theta}} \log \pi_{\boldsymbol{\theta}}(a|s) \big]. \tag{3}$$

Equivalently, the policy gradient can be written as

$$\nabla J(\boldsymbol{\theta}) = \mathbb{E}_{s \sim \mu_{\boldsymbol{\theta}}, a \sim \pi_{\boldsymbol{\theta}}} [(Q_{\boldsymbol{\theta}}(s, a) - b(s)) \nabla_{\boldsymbol{\theta}} \log \pi_{\boldsymbol{\theta}}(a|s)],$$

where $b(s)$ is called the baseline function, which is employed to reduce the variance of the gradient estimate. A popular choice of baseline is the state-value function, which leads to the following so-called advantage-based policy gradient

$$\nabla_{\boldsymbol{\theta}} J(\boldsymbol{\theta}) = \mathbb{E}_{s \sim \mu_{\boldsymbol{\theta}}, a \sim \pi_{\boldsymbol{\theta}}}[\Delta_{\boldsymbol{\theta}}(s, a)\nabla_{\boldsymbol{\theta}} \log \pi_{\boldsymbol{\theta}}(a|s)], \tag{4}$$

where $\Delta_{\boldsymbol{\theta}} := Q_{\boldsymbol{\theta}}(s, a) - V_{\boldsymbol{\theta}}(s)$ is known as the advantage function.

In deep reinforcement learning, the policy and value functions are typically parameterized by deep neural networks (DNNs) due to their strong representation capabilities (Henderson et al., 2018; Zhao et al., 2020). However, the convergence and performance of training DNNs are less understood, especially in reinforcement learning. In this paper, we establish conditions and provide an asymptotic analysis for single-timescale AC algorithms utilizing DNN approximations for both the actor and the critic.

## 3 THE SINGLE-TIMESCALE NEURAL ACTOR-CRITIC ALGORITHM

In this section, we present the single-timescale neural AC algorithm to be analyzed in the sequel, incorporating key components commonly found in practical implementations.

### 3.1 PARAMETERIZATION OF THE VALUE FUNCTION AND POLICY

We consider a multi-layer neural network for estimating the true state-value function $V_{\boldsymbol{\theta}}(s)$ under a policy $\pi_{\boldsymbol{\theta}}$. The network $\widehat{V}(\boldsymbol{\omega}; \boldsymbol{s})$ has a general form of a deep neural network with a linear output layer:

$$\begin{aligned}
\boldsymbol{s}^{(0)} &= \boldsymbol{s}, \\
\boldsymbol{s}^{(k)} &= \frac{1}{\sqrt{m_k}}\sigma(\boldsymbol{W}^{(k)}\boldsymbol{s}^{(k-1)}), \text{ for } k = 1, 2, \cdots, K, \\
\widehat{V}(\boldsymbol{\omega}; \boldsymbol{s}) &= \frac{1}{\sqrt{m_K}}\boldsymbol{b}^{\top}\boldsymbol{s}^{(K)},
\end{aligned} \tag{5}$$

where $K$ is the total number of hidden layers, state $\boldsymbol{s} \in \mathbb{R}^{d_s}$ is the input to the neural network, $\sigma$ is an element-wise activation function, $\boldsymbol{b}$ is a fixed coefficient vector for the output layer, and $\boldsymbol{\omega} \in \mathcal{X}_{\Omega}$ stands for the trainable parameter of the neural network. The latter is a column vector formed by stacking the weights of different layers, $\boldsymbol{\omega} := \{\boldsymbol{W}^{(k)} \in \mathbb{R}^{m_k \times m_{k-1}}\}_{k=1}^{K}$, where $m_k \in \mathbb{N}$ is the width of the $k$-th layer and $m_0 = d_s$ is the input dimension. Without loss of generality, we assume all the hidden layers have the same width $m$, i.e., $m_k = m$ for $k \in \{1, 2, \cdots, K\}$. It is for the ease of presentation only. As shown in the proof, our analysis also applies to $m_k \geq m$. We admit some freedom to choose the activation function $\sigma(\cdot)$. It only needs to satisfy Assumption 4.1. For example, it can be sigmoid and GeLU (Hendrycks & Gimpel, 2016). Note that the above definition is general enough to encompass standard multilayer perceptrons (MLPs), convolutional neural networks (CNNs), and residual networks (ResNets) as special cases.

The policy $\pi_{\boldsymbol{\theta}}$ is allowed to have a general parameterization, including linear functions (Yang et al., 2019), deep neural networks (Wang et al., 2019), and energy-based policies (Fu et al., 2020). For the DNN case, the actor can be parameterized similarly to Eq. (5), where all the trainable parameters will be stacked into the column vector $\boldsymbol{\theta} \in \mathcal{X}_{\Theta}$.

### 3.2 ALGORITHM DESIGN

In this subsection, we first aim to update the parameter of the neural network (the critic) $\boldsymbol{\omega}$ so that $\widehat{V}(\boldsymbol{\omega}; s)$ can approximate the true value function $V_{\boldsymbol{\theta}}(s)$ of a policy $\pi_{\boldsymbol{\theta}}$. Concretely, at step $t$, we implement Stochastic Gradient Descent (SGD) methods to adjust the critic in the direction that would most reduce the mean square value error $[V(s_t) - \widehat{V}(\boldsymbol{\omega}_t; s_t)]^2$:

$$\boldsymbol{\omega}_{t+1} = \boldsymbol{\omega}_t - \frac{1}{2}\beta\nabla[V(s_t) - \widehat{V}(\boldsymbol{\omega}_t; s_t)]^2 = \boldsymbol{\omega}_t + \beta[V(s_t) - \widehat{V}(\boldsymbol{\omega}_t; s_t)]\nabla_{\boldsymbol{\omega}}\widehat{V}(\boldsymbol{\omega}_t; s_t), \tag{6}$$

where $\beta$ is the stepsize (learning rate). Since $V(s_t)$ is unknown, the semi-gradient TD(0) method approximates it by replacing $V(s_t)$ with the current target $r_t - J(\boldsymbol{\theta}) + \widehat{V}(\boldsymbol{\omega}_t; s_{t+1})$. To further

---

**Algorithm 1** Single-Timescale Neural Actor-Critic

---

1: **Input** initial actor parameter $\boldsymbol{\theta}_0$, initial critic parameter $\boldsymbol{\omega}_0$, initial reward estimator $\eta_0$, stepsizes $\alpha$ for actor, $\beta$ for critic, and $\gamma$ for reward estimator.
2: Draw $s_0$ from some initial distribution
3: **for** $t = 0, 1, 2, \cdots, T-1$ **do**
4:      Take action $a_t \sim \pi_{\boldsymbol{\theta}_t}(\cdot|s_t)$
5:      Observe next state $s_{t+1} \sim \mathcal{P}(\cdot|s_t, a_t)$ and reward $r_t = r(s_t, a_t)$
6:      $\delta_t = r_t - \eta_t + \widehat{V}(\boldsymbol{\omega}_t; \boldsymbol{s}_{t+1}) - \widehat{V}(\boldsymbol{\omega}_t; \boldsymbol{s}_t)$
7:      $\eta_{t+1} = \eta_t + \gamma(r_t - \eta_t)$
8:      $\boldsymbol{\omega}_{t+1} = proj_{\mathcal{B}_{\boldsymbol{\omega}_0}}(\boldsymbol{\omega}_t + \beta\delta_t\nabla_{\boldsymbol{\omega}}\widehat{V}(\boldsymbol{\omega}; \boldsymbol{s}_t))$
9:      $\boldsymbol{\theta}_{t+1} = \boldsymbol{\theta}_t + \alpha\delta_t\nabla_{\boldsymbol{\theta}}\log\pi_{\boldsymbol{\theta}_t}(a_t|s_t)$
10: **end for**

---

estimate the unknown time-average reward $J(\boldsymbol{\theta})$, we use the following exponential moving average update of $\eta_t$,

$$\eta_{t+1} = \eta_t + \gamma(r_t - \eta_t),$$

where $\gamma$ is the stepsize. Hereafter, we will refer to it as the *reward estimator*. This additional estimation of the time-average reward $J(\boldsymbol{\theta})$ introduces more analysis complexity compared to the discounted setting (Olshevsky & Gharesifard, 2023; Tian et al., 2024). Now, by denoting the TD error as

$$\delta_t := r_t - \eta_t + \widehat{V}(\boldsymbol{\omega}_t; \boldsymbol{s}_{t+1}) - \widehat{V}(\boldsymbol{\omega}_t; \boldsymbol{s}_t),$$

we can rewrite the update of the critic in Eq. (6) as

$$\boldsymbol{\omega}_{t+1} = \boldsymbol{\omega}_t + \beta\delta_t\nabla_{\boldsymbol{\omega}}\widehat{V}(\boldsymbol{\omega}; \boldsymbol{s}_t).$$

For the neural network specified in Section 3.1, we require its width $m$ to be sufficiently large such that the neural network is in the overparameterization regime. In this regime, the optimal solution typically resides in the neighborhood of the initialization (Du et al., 2019; Chen et al., 2021; Tian et al., 2024). Therefore, in Line 8 of Algorithm 1, we constrain the update of the critic parameter within a ball of constant radius around its initial condition, which ensures the boundedness without overlooking the optimal solution. Specifically, $proj_{\mathcal{B}_{\boldsymbol{\omega}_0}}$ stands for the projection onto a ball with a constant radius around the initial condition of the critic, i.e., $\mathcal{B}_{\boldsymbol{\omega}_0} = \{\boldsymbol{\omega}|\|\boldsymbol{\omega} - \boldsymbol{\omega}_0\| \le U_{\boldsymbol{\omega}}\}$, where $U_{\boldsymbol{\omega}}$ is a constant.

For the actor update, it is standard to use the TD error ($\delta_t$) as an approximation of the advantage function (Sutton & Barto, 2018). Therefore, based on the policy gradient theorem, the corresponding update rule for the actor can be written as

$$\boldsymbol{\theta}_{t+1} = \boldsymbol{\theta}_t + \alpha\delta_t\nabla_{\boldsymbol{\theta}}\log\pi_{\boldsymbol{\theta}_t}(a_t|s_t),$$

where $\delta_t\nabla_{\boldsymbol{\theta}}\log\pi_{\boldsymbol{\theta}_t}(a_t|s_t)$ is an approximation of the policy gradient defined in Eq. (4). The parallel updates of the critic and actor in Lines 8 and 9 aim to drive the actor towards the direction that increases the time-average reward $J(\boldsymbol{\theta})$.

Algorithm 1 is considered to be "single-timescale" if the stepsizes $\alpha, \beta, \gamma$ are only constantly proportional to each other. It is introduced in the classic textbook (Sutton & Barto, 2018) as a canonical AC algorithm with linear function approximation. We take a significant step forward to consider the more challenging neural network approximation for both the actor and the critic, which is referred to as the "neural actor-critic". Moreover, we consider the more practical Markovian sampling, starting from an initial state $s_0$, with subsequent states and actions generated according to the transition kernel and the policy, respectively. The consecutive transition tuples $(s_0, a_0, s_1, a_1, s_2, \cdots)$ form a single trajectory, thereby circumventing the time-consuming re-sampling procedure (i.i.d. sampling) mandated in prior works (Chen et al., 2021; Olshevsky & Gharesifard, 2023; Tian et al., 2024). More importantly, we aim to address the challenging settings of continuous state and action spaces that are prevalent in applications. The finite-time convergence in such contexts is of significant interest to the community but remains unresolved.

# 4 ANALYSIS OF SINGLE-TIMESCALE NEURAL ACTOR-CRITIC

In this section, we first outline several standard assumptions regarding the neural networks and the underlying MDP that facilitate the convergence analysis of single-timescale neural AC algorithm. We also discuss insights related to these conditions and their connections with relevant literature. Building upon these assumptions, we subsequently present our main results on the finite-time convergence of the algorithm.

## 4.1 ASSUMPTIONS

We first state the assumptions about the neural network defined in Eq. (5).

**Assumption 4.1** (Neural architecture and initialization). The neural network defined in Eq. (5) satisfies the following properties:

(a) (Input assumption) Any input to the neural network satisfies $\|\boldsymbol{s}^{(0)}\| \leq 1$.

(b) (Activation function assumption) $\sigma$ is $L_a$-Lipschitz and $H_a$-smooth, i.e.,

  (i) $\forall x_1, x_2 \in \mathbb{R}, |\sigma(x_1) - \sigma(x_2)| \leq L_a|x_1 - x_2|$.
  (ii) $\forall x_1, x_2 \in \mathbb{R}, |\sigma'(x_1) - \sigma'(x_2)| \leq H_a|x_1 - x_2|$, where $\sigma'$ is the derivative of $\sigma$.

(c) (Initialization assumption) Each entry of the vector $\boldsymbol{b}$ satisfies $|b_i| \leq 1, \forall i$, and the weights of the neural network $\boldsymbol{W}_0^{(k)}$ are randomly initialized from a normal distribution $\mathcal{N}(0, 1)$, with each entry being independently sampled.

This assumption mainly states the initialization and analytic properties of the neural network. We note that these assumptions are widely satisfied in various applications. For the input norm constraint, we could normalize the state space to guarantee this assumption. Regarding the activation function, we emphasize that many commonly used activation functions, such as sigmoid and GeLU, satisfy this condition. While this assumption excludes non-smooth activation functions like ReLU, alternatives such as GeLU or SiLU (smooth versions of ReLU) can be employed to maintain compliance with the assumption. The initialization assumption, furthermore, can be easily implemented during neural network training. We also note that the above assumptions are common in the theoretical analysis of neural networks (Liu et al., 2020; Tian et al., 2024).

As shown in Liu et al. (2020), with Assumption 4.1, the following assumption holds with high probability (Lemma F.4 in Liu et al. (2020)), which we state as an assumption in our work for ease of presentation.

**Assumption 4.2.** The absolute value of each entry of $\boldsymbol{s}^{(k)}$ (the output of layer $k$ of the neural network) is $\widetilde{O}(1)$ at initialization. The initial weights satisfy $\|\boldsymbol{W}_0^{(k)}\| \leq \mathcal{O}(\sqrt{m})$ for all $k$.

For the value function $V_{\boldsymbol{\theta}}(s)$ of a given policy $\boldsymbol{\theta}$, its best approximation using the neural network (Eq. (5)) is defined via

$$\epsilon_{\text{app}}(\boldsymbol{\omega}^*(\boldsymbol{\theta})) := \inf_{\boldsymbol{\omega}} \sqrt{\mathbb{E}_{s \sim \mu_{\boldsymbol{\theta}}}\left[(\widehat{V}(\boldsymbol{\omega}; s) - V_{\boldsymbol{\theta}}(s))^2\right]}, \tag{7}$$

where $\boldsymbol{\omega}^*(\boldsymbol{\theta})$ is referred to as the *optimal critic* that yields the minimal (optimal) approximation error $\epsilon_{\text{app}}(\boldsymbol{\omega}^*(\boldsymbol{\theta}))$. In this paper, we assume the optimal approximation errors for all potential policies are uniformly bounded, that is,

$$\forall \boldsymbol{\theta}, \ \epsilon_{\text{app}}(\boldsymbol{\omega}^*(\boldsymbol{\theta})) \leq \epsilon_{\text{app}},$$

for some constant $\epsilon_{\text{app}} \geq 0$. The error $\epsilon_{\text{app}}$ is zero if $V_{\boldsymbol{\theta}}$ can be exactly approximated by the neural network (Eq. (5)). Naturally, it is expected that the learning errors of Algorithm 1 depend on $\epsilon_{\text{app}}$, which represents the approximation capacity of the critic.

The assumption of a uniformly bounded approximation error is common in the literature (Chen et al., 2021; Olshevsky & Gharesifard, 2023; Chen & Zhao, 2024; Tian et al., 2024). It is more restrictive for the linear function approximation than for the neural network setting. If the true

value function is not linear, which is typically the case in practice, the approximation error $\epsilon_{\text{app}}$ can be significantly large. In contrast, the neural network approximation can arbitrarily closely approximate any continuous function according to the Universal Approximation Theorem (Hornik, 1991), and therefore can potentially keep the approximation error arbitrarily small.

We then make the following assumption for the optimal critic.

**Assumption 4.3** (Smoothness of optimal critic). For any $\boldsymbol{\theta}_1, \boldsymbol{\theta}_2 \in \mathcal{X}_\Theta$, we have

$$\|\boldsymbol{\omega}^*(\boldsymbol{\theta}_1) - \boldsymbol{\omega}^*(\boldsymbol{\theta}_2)\| \leq L_*\|\boldsymbol{\theta}_1 - \boldsymbol{\theta}_2\|,$$
$$\|\nabla\boldsymbol{\omega}^*(\boldsymbol{\theta}_1) - \nabla\boldsymbol{\omega}^*(\boldsymbol{\theta}_2)\| \leq L_s\|\boldsymbol{\theta}_1 - \boldsymbol{\theta}_2\|,$$

where $L_*$ and $L_s$ are finite positive constants.

The above assumption states that the optimal critic is $L_*$-Lipschitz and $L_s$-smooth. This assumption is commonly employed for the single-timescale AC with neural network approximation (Tian et al., 2024). In the case of linear function approximation, the above assumption is trivially implied by the linearity of the value function (Olshevsky & Gharesifard, 2023; Chen & Zhao, 2024).

Furthermore, we specify the regularity of the neural network.

**Assumption 4.4** (Regularity of the neural network). For the neural network defined in Eq. (5), there exists some constant $\lambda_1 > 0$ such that

$$\|\widehat{V}(\boldsymbol{\omega}) - \widehat{V}(\boldsymbol{\omega}^*(\boldsymbol{\theta}))\| \geq \lambda_1\|\boldsymbol{\omega} - \boldsymbol{\omega}^*(\boldsymbol{\theta})\|, \qquad \forall \boldsymbol{\theta} \in \mathcal{X}_\Theta, \boldsymbol{\omega} \in \mathcal{X}_\Omega,$$

where the norm of a function is defined based on the inner product given in Eq. (2), which involves the product of function values integrated over $s$. Assumption 4.4 states the regularity of the neural network in terms of learning the optimal value. Intuitively, it requires that the perturbation of the critic parameter around the optimal one will cause a non-zero change of the critic neural network output for any given input (the state). From the point of view of the optimization landscape of the neural network, it merely assumes that optimal and suboptimal points are distinguished. This is also a standing assumption of other analysis of AC methods with neural network approximation (Tian et al., 2024).

The next assumption pertains to the exploration of the policy $\pi_{\boldsymbol{\theta}}$ in continuous settings.

**Assumption 4.5** (Exploration). There exists a constant $\lambda_2 > 0$ such that $\left\langle \widehat{V}(\boldsymbol{\omega}), D_{\boldsymbol{\theta}}(I - P_{\boldsymbol{\theta}})\widehat{V}(\boldsymbol{\omega}) \right\rangle \geq \lambda_2\|\widehat{V}(\boldsymbol{\omega})\|^2$, for any $\boldsymbol{\theta} \in \mathcal{X}_\Theta$ and neural network $\widehat{V}(\boldsymbol{\omega}) \in \mathcal{F}$, where $D_{\boldsymbol{\theta}}, P_{\boldsymbol{\theta}}$ are operators defined in Eq. (1), $I$ denotes the identity operator, and the inner product is defined in Eq. (2).

This assumption was first introduced by us for the continuous setting with general function approximation classes. To demonstrate its connection to exploration, we show that if exploration is insufficient, the assumption fails to hold. Consequently, when the assumption holds, it implies sufficient exploration. First note that the operator $D_\theta$ essentially multiplies the stationary distribution $\mu_\theta$ to the function on its right (see the definition in Eq. (1)). If the policy $\pi_\theta$ does not sufficiently explore, there exists a subset of the state space $U \subset \mathcal{S}$ such that $\mu_\theta(U) = 0$. Furthermore, we can choose $\hat{V}(\omega)$ such that $\hat{V}(\omega; s) = 0, \forall s \in \mathcal{S} \setminus U$ and $\hat{V}(\omega; s) \geq 0, \forall s \in U$. With this choice, the left-hand side of the inequality evaluates to 0, while the right-hand side becomes positive. This violates the condition stated in Assumption 4.5. Thus, the contrapositive holds: if Assumption 4.5 is satisfied, it ensures sufficient exploration of the state space under the policy $\pi_\theta$.

Note that sufficient exploration assumption is standard in the literature of analyzing the convergence of on-policy RL algorithms (Bhandari et al., 2018; Zou et al., 2019; Wu et al., 2020b; Olshevsky & Gharesifard, 2023; Chen & Zhao, 2024). We can also drop this condition by analyzing the off-policy version of the algorithm under some sufficiently-exploring behavior policy that can be arbitrarily specified, and relates to the target policy by importance sampling. However, this is not the core focus of the problem. Therefore, we adopt Assumption 4.5 directly and concentrate on the primary challenge of analyzing the algorithm in the continuous state-action space.

The following assumption is made on the underlying MDP.

**Assumption 4.6** (Uniform ergodicity). For a Markov chain generated by the policy $\pi_{\boldsymbol{\theta}}$ and transition kernel $\mathcal{P}$, let $\mathbb{P}$ denote the corresponding state transition probability. Then there exists $C > 0$ and

$\rho \in (0, 1)$ such that the total variation distance between the state distribution at time $\tau$ and the stationary distribution $\mu_\theta$ satisfies: $d_{TV}(\mathbb{P}(s_\tau \in \cdot | s_0 = s), \mu_\theta(\cdot)) \leq C\rho^\tau$, for all $\tau \geq 0$, $s \in \mathcal{S}$.

Assumption 4.6 assumes the Markov chain is geometrically mixing, which is implied by the uniform ergodicity of the chain. It is commonly employed to characterize the noise induced by Markovian sampling in reinforcement learning algorithms (Bhandari et al., 2018; Zou et al., 2019; Wu et al., 2020b; Chen et al., 2021; Olshevsky & Gharesifard, 2023).

To justify this assumption in the continuous space, we note that all the distributions specified by the Ornstein–Uhlenbeck process satisfy this property. The OU process converges to a Gaussian distribution with the exponential mixing time. Moreover, it can also be shown that this property holds for more general diffusion processes (Del Moral & Villemonais, 2018).

Finally, we need some regularity assumptions on the policy.

**Assumption 4.7** (Smoothness of the policy). Let $\pi_\theta(a|s)$ be a policy parameterized by $\theta \in \mathcal{X}_\Theta$. There exists positive constants $B$, $L_l$ and $L_\pi$ such that for any $\theta$, $s$, and $a$, it holds that

(a) $\|\nabla \log \pi_\theta(a|s)\| \leq B$,

(b) $\|\nabla \log \pi_{\theta_1}(a|s) - \nabla \log \pi_{\theta_2}(a|s)\| \leq L_l \|\theta_1 - \theta_2\|$,

(c) $d_{TV}(\pi_{\theta_1}(\cdot|s), \pi_{\theta_2}(\cdot|s)) \leq L_\pi \|\theta_1 - \theta_2\|$.

Assumption 4.7 (a) and (b) are standard and widely adopted across the prior results presented in Table 1. For Assumption 4.7 (c), previous research considers the finite action space only and relies on a degenerated version of the condition, which is simply the Lipschitz continuity of the policy, i.e., $|\pi_{\theta_1}(a|s) - \pi_{\theta_2}(a|s)| \leq L\|\theta_1 - \theta_2\|$, where the absolute distance on the left is evaluated between two function values at a single action point. In contrast, we generalize this condition by employing the Lipschitz continuity of two *distributions* (either probability mass or density functions) under the total variation distance. Our assumption naturally accommodates continuous action spaces and encompasses the finite action space conditions considered in prior research as a special case.

Under the continuous state and action spaces settings, we further justify that Assumption 4.7 (c) is sufficiently general and can be satisfied by a broad range of parameterization methods in the following proposition.

**Proposition 4.8** (Generality of Assumption 4.7 (c)). *Under the following conditions:*

(a) *(Support Compactness) For any $\theta$, the policy $\pi_\theta(a|s)$ has compact support $\mathcal{X}_A \subset \mathbb{R}^{d_a}$.*

(b) *(Density Lipschitzness) For any $\theta$, the policy $\pi_\theta(a|s)$ is Lipschitz w.r.t $a$, i.e., $|\pi_\theta(a_1|s) - \pi_\theta(a_2|s)| \leq L_1 \|a_1 - a_2\|$ for some constant $L_1 > 0$ and all $a_1, a_2 \in \mathbb{R}^{d_a}$.*

(c) *(Neural Network Lipschitzness) Let the policy $\pi_\theta(\cdot|s)$ be a distribution with its mean value parameterized by the neural network $\bar{\mu}_\theta(s)$. For any $s$, $\bar{\mu}_\theta(\cdot)$ is Lipschitz w.r.t. $\theta$, i.e., $|\bar{\mu}_{\theta_1}(s) - \bar{\mu}_{\theta_2}(s)| \leq L_2 \|\theta_1 - \theta_2\|$ for some constant $L_2 > 0$ and all $\theta_1, \theta_2 \in \mathcal{X}_\Theta$,*

*Assumption 4.7 (c) holds with $L_\pi = L_1 L_2 |\mathcal{X}_A|$, where $\mathcal{X}_A$ is the volume of $\mathcal{X}_A$, i.e., $|\mathcal{X}_A| = \int_{\mathcal{X}_A} da$.*

Conditions (a) and (b) assert that the policy $\pi_\theta(\cdot|s)$ has compact support and is Lipschitz continuous with respect to $a$. These conditions are sufficiently general to be satisfied by a wide range of distributions, including the uniform distribution, the truncated Gaussian distribution, and the Beta distribution with $\alpha, \beta > 1$. Condition (c) holds for commonly used neural networks such as MLP and Transformer (Bartlett et al., 2017; Zhang et al., 2022). Consequently, Assumption 4.7 (c) is satisfied by a wide range of distributions with their mean parameterized by MLP or Transformer, thus demonstrating the generality of the newly proposed Assumption 4.7 (c).

## 4.2 FINITE-TIME ANALYSIS

We define the integer $\tau_T := \min\{i \geq 0 \mid C\rho^{i-1} \leq T^{-1/2}\}$ given $T$ the total number of iterations (see Algorithm 1), where $C, \rho$ are the same constants defined in Assumption 4.6. The integer $\tau_T$ represents a certain mixing time of an ergodic Markov chain, which will be used to control the

Markovian noise in the analysis. In our main results, we require that $T \geq 2\tau_T$ to ensure that the Markov chain is well-mixed and the Markovian noise is effectively bounded. We can estimate that $\tau_T = \frac{\log C\rho^{-1}}{\log \rho^{-1}} + \frac{\log T}{2\log \rho^{-1}} = \mathcal{O}(\log T)$ which results in $C\rho^{\tau_T - 1} \leq \frac{1}{\sqrt{T}}$.

We quantify the *learning errors* by defining $y_t := \eta_t - J(\boldsymbol{\theta}_t)$, which is the difference between the reward estimator and the true time-average reward $J(\boldsymbol{\theta}_t)$ at time $t$. For the critic, we define $\boldsymbol{z}_t := \boldsymbol{\omega}_t - \boldsymbol{\omega}_t^*$ with $\boldsymbol{\omega}_t^* := \boldsymbol{\omega}^*(\boldsymbol{\theta}_t)$ to measure the error between the critic and its target value at iteration $t$. The following theorem summarizes our main results.

**Theorem 4.9.** *Consider Algorithm 1 with $\alpha = \frac{c}{\sqrt{T}}, \beta = \frac{1}{\sqrt{T}}, \gamma = \frac{1}{\sqrt{T}}$, where $c$ is a constant depending on problem parameters. Suppose Assumption 4.1-4.7 hold, for $T \geq 2\tau_T$, we have*

$$\frac{1}{T - \tau_T} \sum_{t=\tau_T}^{T-1} \mathbb{E}[y_t^2] = \mathcal{O}(\frac{\log^2 T}{\sqrt{T}}) + \widetilde{\mathcal{O}}(\frac{1}{\sqrt{m}}) + \mathcal{O}(\epsilon_{\mathrm{app}}),$$

$$\frac{1}{T - \tau_T} \sum_{t=\tau_T}^{T-1} \mathbb{E}\|\boldsymbol{z}_t\|^2 = \mathcal{O}(\frac{\log^2 T}{\sqrt{T}}) + \widetilde{\mathcal{O}}(\frac{1}{\sqrt{m}}) + \mathcal{O}(\epsilon_{\mathrm{app}}),$$

$$\frac{1}{T - \tau_T} \sum_{t=\tau_T}^{T-1} \mathbb{E}\|\nabla J(\boldsymbol{\theta}_t)\|^2 = \mathcal{O}(\frac{\log^2 T}{\sqrt{T}}) + \widetilde{\mathcal{O}}(\frac{1}{\sqrt{m}}) + \mathcal{O}(\epsilon_{\mathrm{app}}).$$

Given that the problem is inherently non-convex in general, it is common to prove convergence to a stationary point. The error term $\mathcal{O}(\epsilon_{\mathrm{app}})$ represents the critic approximation error that commonly appears in the analysis of AC methods (Wu et al., 2020b; Chen & Zhao, 2024; Tian et al., 2024). If the critic approximation error $\epsilon_{\mathrm{app}}$ is zero, the reward estimator, the critic, and the actor estimation errors all vanish at a rate of $\widetilde{\mathcal{O}}(T^{-\frac{1}{2}}) + \widetilde{\mathcal{O}}(m^{-\frac{1}{2}})$, where again $m$ denotes the width of the neural networks adopted. The $\widetilde{\mathcal{O}}$ notation hides the polynomials of all other problem parameters that do not depend on $T, m$ and $\epsilon_{\mathrm{app}}$. The additional logarithmic term with respect to $T$ arises from the mixing time of the Markov chain, which can be further eliminated if considering the i.i.d. sampling model.

Compared to previous results on single-timescale AC methods, we achieve the same convergence rate of $\widetilde{\mathcal{O}}(T^{-\frac{1}{2}})$ with respect to the number of total iterations $T$. The term $\widetilde{\mathcal{O}}(m^{-\frac{1}{2}})$ emerges from neural network analysis, which is consistent with previous findings (Liu et al., 2020; Tian et al., 2024). It is important to note that in linear function approximation cases, the approximation error ($\epsilon_{\mathrm{app}}$) serves as the primary source of learning errors due to its limited expressive capacity.

Our proof analyzes and tracks the interactions of the three errors $(y_t, \boldsymbol{z}_t, \nabla J(\boldsymbol{\theta}_t))$ by deriving their implicit bounds that are dependent on each other. Subsequently, we prove their *simultaneous convergence* under a series of technical developments. Considering continuous spaces and deep neural networks substantially complicate the bounding of the error terms. For example, to analyze the inner product between $\boldsymbol{z}_t$ and the critic's mean-path update $\bar{g}(\boldsymbol{\omega}_t, \boldsymbol{\theta}_t)$ as defined in Eq. (10), we employ the Bellman equation and neural network approximation to manage error propagation. This error is controlled by leveraging the approximation capability of the neural network, the linearity of wide networks, and sufficient policy exploration (see Section E in Appendix for a detailed proof sketch). In contrast, (Chen & Zhao, 2024) manages this term through direct computation by exploiting the linearity of the value function.

Moreover, we manage to control Markovian noise in continuous state and action spaces, which involves novel results established in Lemma C.1, which characterizes the distance between stationary distributions in these continuous spaces. This approach is distinct from the finite action space setting (Chen & Zhao, 2024) and is considerably more intricate than the i.i.d. sampling scheme (Olshevsky & Gharesifard, 2023; Tian et al., 2024). Compared with the Neural Tangent Kernel (NTK) analysis (Jacot et al., 2018; Allen-Zhu et al., 2019; Liu et al., 2020) where the neural network is trained to learn a fixed mapping, the neural network in our algorithm is trained to estimate the value function of an evolving policy, which requires a novel design of the update rates and less conservative treatment of the coupling learning errors.

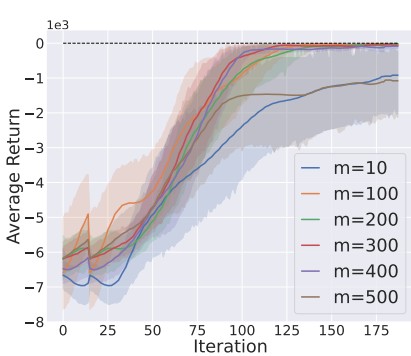
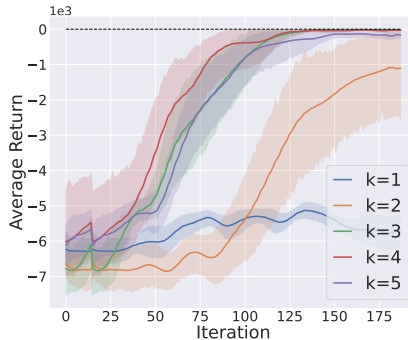

(a) Performance of Algorithm 1 against widths          (b) Performance of Algorithm 1 against depths

Figure 1: Experimental results of Algorithm 1 on the pendulum problem.

## 5 EXPERIMENTS

We evaluate the performance of Algorithm 1 in the classic benchmark "Pendulum" environment. The Pendulum environment features a continuous state space represented by $[\cos(\theta), \sin(\theta), \dot{\theta}]$, where $\theta$ is the pendulum angle and $\dot{\theta}$ is the angular velocity. The action space is also continuous, consisting of a single torque value $\tau$ typically ranging from $-2$ to $2$. The reward function is designed to penalize deviations from the upright position and the magnitude of the applied torque, calculated as $R = -(\theta^2 + 0.1\dot{\theta}^2 + 0.001\tau^2)$. In our experiment, episodes terminate after 1000 time steps. At the beginning of each run, the state is initialized at a random angle in $[-\pi, \pi]$ and a random angular velocity in $[-1, 1]$.

We employ a truncated Gaussian policy defined as $\pi_{\boldsymbol{\theta}} = \text{Truncated}(\mathcal{N}(\boldsymbol{\theta}, 1), -1, 1)$ for the actor, where the mean $\boldsymbol{\theta}$ is learned using Algorithm 1, while the variance remains fixed at 1. The mean value $\boldsymbol{\theta}$ is parameterized by the neural network defined in Eq. (5) with 2 hidden layers and 64 neurons in each layer, i.e., $K = 2, m = 64$. The parameterization of the critic $\boldsymbol{\omega}$ is specified in Eq. (5) as outlined in Section 3.1. To verify our theoretical findings, we evaluate the performance of Algorithm 1 with varying widths and depths for the critic. The tanh activation function is employed, adhering to Assumption 4.1b.

In Fig. 1, the solid lines correspond to the mean and the shaded regions correspond to 95% confidence interval over 10 independent runs. The dashed line corresponds to a value of 0, representing the theoretically achievable optimal value for this task. The average return is calculated as the mean of the last 40 returns. When the average return is around -200, it indicates that the pendulum is being kept upright. Fig. 1a and 1b show the performance of Algorithm 1 under different widths $m$ and depths $K$, respectively. In our experiment, we set the stepsizes as $5e^{-6}$ for both the critic and the actor. In Figures 1a, the number of hidden layers of the network is fixed at 2 while in Fig. 1b, the network width of each hidden layer is fixed at 200. These results indicate that the neural networks with larger sizes can outperform the smaller neural networks, which strongly corroborates our theoretical findings.

## 6 CONCLUSION AND DISCUSSION

In this paper, we present a finite-time analysis for single-timescale AC methods, achieving a convergence rate of $\widetilde{\mathcal{O}}(T^{-1/2}) + \widetilde{\mathcal{O}}(m^{-1/2})$. Our results surpass those of existing works by effectively addressing continuous state and action spaces, utilizing Markovian sampling, and employing deep neural network approximations for both critic and actor. Note that we focus on overparameterized neural networks in terms of having a much larger width than depth, i.e., $m \gg K$. In this regime, the depth has a relatively minor influence on the performance of learning (Jacot et al., 2018). In our result, the dependence of the depth is implicitly captured by the constants defined in Lemma C.5. Characterizing more general cases where depth is prominent in influencing the learning performance and its dependence order explicitly remains an open and challenging problem.

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

APPENDIX

# Table of Contents

## A  RELATED WORK

**AC methods.** The AC algorithm was initially proposed by Konda & Tsitsiklis (1999). Subsequently, Kakade (2001) extended it to the natural AC algorithm. The asymptotic convergence of AC algorithms has been well established under various settings, as demonstrated in works by Kakade (2001), Bhatnagar et al. (2009), Castro & Meir (2010), and Zhang et al. (2020b). More recently, many studies have focused on the finite-time convergence of AC methods. Under the double-loop setting, Kumar et al. (2019) investigated the finite-time local convergence of several AC variants with linear function approximation. Wang et al. (2019) explored the global convergence of AC methods with both the actor and the critic parameterized by neural networks with single hidden layers. Cayci et al. (2022) improved upon the work of Wang et al. (2019) by considering Markovian sampling and reducing sample complexity.

Under the two-timescale AC setting, Wu et al. (2020b) established the finite-time local convergence to a stationary point at a sample complexity of $\widetilde{\mathcal{O}}(\epsilon^{-2.5})$ under the undiscounted time-average reward setting. Xu et al. (2020b) studied both local convergence and global convergence for two-timescale (natural) AC, with $\widetilde{\mathcal{O}}(\epsilon^{-2.5})$ and $\widetilde{\mathcal{O}}(\epsilon^{-4})$ sample complexity, respectively, under the discounted accumulated reward. The algorithm collects multiple samples to update the critic. Hong et al. (2023) proposed a two-timescale stochastic approximation algorithm for bilevel optimization and the algorithm was subsequently employed in the context of two-timescale AC. Chen et al. (2023) established the global convergence of two-timescale AC methods for solving linear quadratic regulator (LQR), where only a single sample is used to update the critic in each iteration. However, none of these previous results utilized neural network approximation for the value function (the critic).

Under the most challenging single-timescale setting, Fu et al. (2020) considered the least-squares temporal difference (LSTD) update for the critic and obtained the optimal policy within the energy-based policy class for both linear function approximation and neural network approximation. (Zhou & Lu, 2023) studied single-timescale AC on LQR. In addition, Chen et al. (2021); Olshevsky & Gharesifard (2023); Chen & Zhao (2024) considered the single-timescale AC in general MDP cases with linear function approximation. Recently, Tian et al. (2024) built upon the results of Olshevsky & Gharesifard (2023) and improved to neural network approximation. A comprehensive review and

comparison of all existing results on single-timescale AC in general MDP settings are presented in Table 1.

## B  ADDITIONAL NOTATIONS

We make use of the following auxiliary Markov chain which was introduced in (Zou et al., 2019) to deal with the Markovian noise.

**Auxiliary Markov Chain:**

$$s_{t-\tau} \xrightarrow{\boldsymbol{\theta}_{t-\tau}} a_{t-\tau} \xrightarrow{\mathcal{P}} s_{t-\tau+1} \xrightarrow{\boldsymbol{\theta}_{t-\tau}} \widetilde{a}_{t-\tau+1} \xrightarrow{\mathcal{P}} \widetilde{s}_{t-\tau+2} \xrightarrow{\boldsymbol{\theta}_{t-\tau}} \widetilde{a}_{t-\tau+2} \cdots \xrightarrow{\mathcal{P}} \widetilde{s}_t \xrightarrow{\boldsymbol{\theta}_{t-\tau}} \widetilde{a}_t \xrightarrow{\mathcal{P}} \widetilde{s}_{t+1}. \tag{8}$$

For reference, we also show the original Markov chain.

**Original Markov Chain:**

$$s_{t-\tau} \xrightarrow{\boldsymbol{\theta}_{t-\tau}} a_{t-\tau} \xrightarrow{\mathcal{P}} s_{t-\tau+1} \xrightarrow{\boldsymbol{\theta}_{t-\tau+1}} \widetilde{a}_{t-\tau+1} \xrightarrow{\mathcal{P}} s_{t-\tau+2} \xrightarrow{\boldsymbol{\theta}_{t-\tau+2}} a_{t-\tau+2} \cdots \xrightarrow{\mathcal{P}} s_t \xrightarrow{\boldsymbol{\theta}_t} a_t \xrightarrow{\mathcal{P}} s_{t+1}. \tag{9}$$

In the sequel, we denote by $\widetilde{O}_t := (\widetilde{s}_t, \widetilde{a}_t, \widetilde{s}_{t+1})$ the tuple generated from the auxiliary Markov chain in Eq. (8) while $O_t := (s_t, a_t, s_{t+1})$ denotes the tuple generated from the original Markov chain in Eq. (9).

We define the following functions, which will benefit to decompose the errors and simplify the presentation.

$$\Delta g(O, \eta, \boldsymbol{\theta}) := [J(\boldsymbol{\theta}) - \eta]\nabla_{\boldsymbol{\omega}}\widehat{V}(\boldsymbol{\omega}; s),$$
$$g(O, \boldsymbol{\omega}, \boldsymbol{\theta}) := [r(s, a) - J(\boldsymbol{\theta}) + \widehat{V}(\boldsymbol{\omega}; s') - \widehat{V}(\boldsymbol{\omega}; s)]\nabla_{\boldsymbol{\omega}}\widehat{V}(\boldsymbol{\omega}; s),$$
$$\bar{g}(\boldsymbol{\omega}, \boldsymbol{\theta}) := \mathbb{E}_{(s,a,s')\sim(\mu_{\boldsymbol{\theta}}, \pi_{\boldsymbol{\theta}}, \mathcal{P})}[(r(s, a) - J(\boldsymbol{\theta}) + \widehat{V}(\boldsymbol{\omega}; s') - \widehat{V}(\boldsymbol{\omega}; s))\nabla_{\boldsymbol{\omega}}\widehat{V}(\boldsymbol{\omega}; s)],$$
$$\Delta h(O, \eta, \boldsymbol{\omega}, \boldsymbol{\theta}) := (J(\boldsymbol{\theta}) - \eta + \widehat{V}(\boldsymbol{\omega}; s') - \widehat{V}(\boldsymbol{\omega}; s) - \widehat{V}(\boldsymbol{\omega}^*(\boldsymbol{\theta}); s') + \widehat{V}(\boldsymbol{\omega}^*(\boldsymbol{\theta}); s))\nabla\log\pi_{\boldsymbol{\theta}}(a|s),$$
$$h(O, \boldsymbol{\theta}) := (r(s, a) - J(\boldsymbol{\theta}) + \widehat{V}(\boldsymbol{\omega}^*(\boldsymbol{\theta}); s') - \widehat{V}(\boldsymbol{\omega}^*(\boldsymbol{\theta}); s))\nabla\log\pi_{\boldsymbol{\theta}}(a|s),$$
$$\Delta h'(O, \boldsymbol{\theta}) := ((\widehat{V}(\boldsymbol{\omega}^*(\boldsymbol{\theta}); s') - V_{\boldsymbol{\theta}}(s')) - (\widehat{V}(\boldsymbol{\omega}^*(\boldsymbol{\theta}); s) - V_{\boldsymbol{\theta}}(s)))\nabla\log\pi_{\boldsymbol{\theta}}(a|s). \tag{10}$$

We also define the following functions, which characterize the Markovian noise.

$$\Phi(O, \eta, \boldsymbol{\theta}) := (\eta - J(\boldsymbol{\theta}))(r(s, a) - J(\boldsymbol{\theta})),$$
$$\Psi(O, \boldsymbol{\omega}, \boldsymbol{\theta}) := \langle \boldsymbol{\omega} - \boldsymbol{\omega}_{\boldsymbol{\theta}}^*, g(O, \boldsymbol{\omega}, \boldsymbol{\theta}) - \bar{g}(\boldsymbol{\omega}, \boldsymbol{\theta}) \rangle,$$
$$\Xi(O, \boldsymbol{\omega}, \boldsymbol{\theta}) := \langle \boldsymbol{\omega} - \boldsymbol{\omega}_{\boldsymbol{\theta}}^*, (\nabla\boldsymbol{\omega}_{\boldsymbol{\theta}}^*)^\top(\mathbb{E}_{O'_{\boldsymbol{\theta}}}[h(O'_{\boldsymbol{\theta}}, \boldsymbol{\theta})] - h(O, \boldsymbol{\theta})) \rangle,$$
$$\Theta(O, \boldsymbol{\theta}) := \langle \nabla J(\boldsymbol{\theta}), \mathbb{E}_{O'_{\boldsymbol{\theta}}}[h(O'_{\boldsymbol{\theta}}, \boldsymbol{\theta})] - h(O, \boldsymbol{\theta}) \rangle, \tag{11}$$

where $O'_{\boldsymbol{\theta}}$ is a shorthand for an independent sample from stationary distribution $s \sim \mu_{\boldsymbol{\theta}}, a \sim \pi_{\boldsymbol{\theta}}, s' \sim \mathcal{P}$.

To demonstrate the main ideas of the proof of Theorem 4.9, we use the notations $Y_T, Z_T$ and $G_T$ for the three errors that we seek to bound, namely,

$$Y_T := \frac{1}{T - \tau_T}\sum_{t=\tau_T}^{T-1}\mathbb{E}y_t^2, \ Z_T := \frac{1}{T - \tau_T}\sum_{t=\tau_T}^{T-1}\mathbb{E}\|\boldsymbol{z}_t\|^2, \ G_T := \frac{1}{T - \tau_T}\sum_{t=\tau_T}^{T-1}\mathbb{E}\|\nabla J(\boldsymbol{\theta}_t)\|^2. \tag{12}$$

Here $Y_T, Z_T,$ and $G_T$ represent the reward estimation error, critic error, and actor error (policy gradient norm), respectively. Our proof of Theorem 4.9 primarily involves analyzing and bounding these three errors relative to one another. The difficulty of this work lies in the continuous state and action spaces and the neural network approximation.

## C    PRELIMINARY LEMMAS

**Lemma C.1** (Distance between stationary distributions). *For any $\boldsymbol{\theta}_1$ and $\boldsymbol{\theta}_2$, it holds that*

$$d_{TV}(\mu_{\boldsymbol{\theta}_1}, \mu_{\boldsymbol{\theta}_2}) \leq L_\pi(\lceil \log_\rho C^{-1} \rceil + \frac{1}{1-\rho}) \|\boldsymbol{\theta}_1 - \boldsymbol{\theta}_2\|,$$

$$d_{TV}(\mu_{\boldsymbol{\theta}_1} \otimes \pi_{\boldsymbol{\theta}_1}, \mu_{\boldsymbol{\theta}_2} \otimes \pi_{\boldsymbol{\theta}_2}) \leq L_\pi(1 + \lceil \log_\rho C^{-1} \rceil + \frac{1}{1-\rho}) \|\boldsymbol{\theta}_1 - \boldsymbol{\theta}_2\|,$$

$$d_{TV}(\mu_{\boldsymbol{\theta}_1} \otimes \pi_{\boldsymbol{\theta}_1} \otimes \mathcal{P}, \mu_{\boldsymbol{\theta}_2} \otimes \pi_{\boldsymbol{\theta}_2} \otimes \mathcal{P}) \leq L_\pi(1 + \lceil \log_\rho C^{-1} \rceil + \frac{1}{1-\rho}) \|\boldsymbol{\theta}_1 - \boldsymbol{\theta}_2\|.$$

**Lemma C.2** (Wu et al. (2020b)). *Given time indexes $t$ and $\tau$ such that $t \geq \tau > 0$, consider the auxiliary Markov chain in Eq. (8). Conditioning on $s_{t-\tau+1}$ and $\boldsymbol{\theta}_{t-\tau}$, we have*

$$d_{TV}(\mathbb{P}(s_{t+1} \in \cdot), \mathbb{P}(\widetilde{s}_{t+1} \in \cdot)) \leq d_{TV}(\mathbb{P}(O_t \in \cdot), \mathbb{P}(\widetilde{O}_t \in \cdot)),$$

$$d_{TV}(\mathbb{P}(O_t \in \cdot), \mathbb{P}(\widetilde{O}_t \in \cdot)) = d_{TV}(\mathbb{P}((s_t, a_t) \in \cdot), \mathbb{P}((\widetilde{s}_t, \widetilde{a}_t) \in \cdot)),$$

$$d_{TV}(\mathbb{P}((s_t, a_t) \in \cdot), \mathbb{P}((\widetilde{s}_t, \widetilde{a}_t) \in \cdot)) \leq d_{TV}(\mathbb{P}(s_t \in \cdot), \mathbb{P}(\widetilde{s}_t \in \cdot)) + \frac{1}{2} L_\pi \mathbb{E}[\|\boldsymbol{\theta}_t - \boldsymbol{\theta}_{t-\tau}\|].$$

**Lemma C.3** (Wu et al. (2020b)). *For any $\boldsymbol{\theta}_1, \boldsymbol{\theta}_2$, we have*

$$|J(\boldsymbol{\theta}_1) - J(\boldsymbol{\theta}_2)| \leq L_J \|\boldsymbol{\theta}_1 - \boldsymbol{\theta}_2\|,$$

*where $L_J = 2U_r L_\pi(1 + \lceil \log_\rho C^{-1} \rceil + \frac{1}{1-\rho})$.*

**Lemma C.4** (Zhang et al. (2020a)). *For the performance function $J(\boldsymbol{\theta})$, there exists a constant $L_{J'} > 0$ such that for all $\boldsymbol{\theta}_1, \boldsymbol{\theta}_2 \in \mathbb{R}^d$, it holds that*

$$\|\nabla J(\boldsymbol{\theta}_1) - \nabla J(\boldsymbol{\theta}_2)\| \leq L_{J'} \|\boldsymbol{\theta}_1 - \boldsymbol{\theta}_2\|, \tag{13}$$

*which further implies*

$$J(\boldsymbol{\theta}_2) \geq J(\boldsymbol{\theta}_1) + \langle \nabla J(\boldsymbol{\theta}_1), \boldsymbol{\theta}_2 - \boldsymbol{\theta}_1 \rangle - \frac{L_{J'}}{2} \|\boldsymbol{\theta}_1 - \boldsymbol{\theta}_2\|^2, \tag{14}$$

$$J(\boldsymbol{\theta}_2) \leq J(\boldsymbol{\theta}_1) + \langle \nabla J(\boldsymbol{\theta}_1), \boldsymbol{\theta}_2 - \boldsymbol{\theta}_1 \rangle + \frac{L_{J'}}{2} \|\boldsymbol{\theta}_1 - \boldsymbol{\theta}_2\|^2. \tag{15}$$

**Lemma C.5** (Boundedness, Lipschitzness, and smoothness of the neural network). *There exists scalars $U_v, L_v,$ and $H_v$ such that for any $s \in \mathcal{S}$ and $\boldsymbol{\omega}_1, \boldsymbol{\omega}_2 \in \mathcal{X}_\Omega$,*

$$\|\widehat{V}(\boldsymbol{\omega}; s)\| \leq U_v,$$

$$\|\widehat{V}(\boldsymbol{\omega}_1; s) - \widehat{V}(\boldsymbol{\omega}_2; s)\| \leq L_v \|\boldsymbol{\omega}_1 - \boldsymbol{\omega}_2\|,$$

$$\|\nabla_{\boldsymbol{\omega}} \widehat{V}(\boldsymbol{\omega}_1; s) - \nabla_{\boldsymbol{\omega}} \widehat{V}(\boldsymbol{\omega}_2; s)\| \leq H_v \|\boldsymbol{\omega}_1 - \boldsymbol{\omega}_2\|,$$

*where $U_v = \mathcal{O}(1)$, $L_v = \mathcal{O}(1)$ and $H_v = \widetilde{O}(\frac{1}{\sqrt{m}})$ with respect to width $m$.*

## D    PROOF OF PROPOSITIONS

We provide the proof of Proposition 4.8 which justifies the generality of the newly proposed Assumption 4.7 (c).

**Proof of Proposition 4.8.**

*Proof.* We adopt neural networks to parameterize the mean value $\bar{\mu}_{\boldsymbol{\theta}}(\cdot)$ of a distribution, where $\theta \in \mathcal{X}_\Theta$ is the neural network parameter. Then the policy can be denoted as $\pi_{\boldsymbol{\theta}}(\cdot|s) = \mathcal{L}(X + \mu_{\boldsymbol{\theta}}(s))$, where $\mathcal{L}(\cdot)$ is the law of the random variables, $X$ is some zero-mean random variable, and $\bar{\mu}_{\boldsymbol{\theta}}(\cdot)$ is the neural network with parameter $\boldsymbol{\theta}$ that takes state $s$ as its input. We denote density function of $X$ as $\pi(a|s)$ whose mean value is zero. With the conditions specified in Proposition 4.8, we show that Assumption 4.7 (c) holds, i.e., $d_{TV}(\pi_{\theta_1}(\cdot|s), \pi_{\theta_2}(\cdot|s)) \leq L_\pi|\theta_1 - \theta_2|$ for some $L_\pi$.

It holds that

$$
\begin{aligned}
d_{TV}&(\pi_{\boldsymbol{\theta}_1}(\cdot|s), \pi_{\boldsymbol{\theta}_2}(\cdot|s)) \\
&= d_{TV}(\mathcal{L}(X + \bar{\mu}_{\boldsymbol{\theta}_1}(s)), \mathcal{L}(X + \bar{\mu}_{\boldsymbol{\theta}_2}(s))) \\
&= \frac{1}{2} \int_{\mathbb{R}^{d_a}} \left| \pi(a - \bar{\mu}_{\boldsymbol{\theta}_1}(s)|s) - \pi(a - \bar{\mu}_{\boldsymbol{\theta}_2}(s)|s) \right| da \\
&= \frac{1}{2} \int_{\mathcal{Y}_A} \left| \pi(a - \bar{\mu}_{\boldsymbol{\theta}_1}(s)|s) - \pi(a - \bar{\mu}_{\boldsymbol{\theta}_2}(s)|s) \right| \mathrm{d}x \\
&\leq \frac{1}{2} \int_{\mathcal{Y}_A} L_1 |\bar{\mu}_{\boldsymbol{\theta}_1}(s) - \bar{\mu}_{\boldsymbol{\theta}_2}(s)| \mathrm{d}x \\
&\leq L_1 \cdot |\mathcal{X}_A| \cdot |\bar{\mu}_{\boldsymbol{\theta}_1}(s) - \bar{\mu}_{\boldsymbol{\theta}_2}(s)|,
\end{aligned}
$$

where $\mathcal{Y}_A$ in the third equality is defined as $\mathcal{Y}_A = (\mathcal{X}_A + \bar{\mu}_{\boldsymbol{\theta}_1}(s)) \cup (\mathcal{X}_A + \bar{\mu}_{\boldsymbol{\theta}_2}(s))$. Combining this with the neural network Lipschitzness, we have that

$$
d_{TV}(\pi_{\boldsymbol{\theta}_1}(\cdot|s), \pi_{\boldsymbol{\theta}_2}(\cdot|s)) \leq L_1 \cdot L_2 \cdot |\mathcal{X}| \cdot |\boldsymbol{\theta}_1 - \boldsymbol{\theta}_2|.
$$

Thus, we conclude the proof of this proposition. $\qquad\square$

# E PROOF SKETCH

In this subsection, we sketch the main proof steps of Theorem 4.9. The key challenges and new techniques developed are also highlighted correspondingly. We first derive implicit (coupled) upper bounds for the reward estimation error $y_t$, the critic error $\boldsymbol{z}_t$, and the policy gradient $\nabla J(\boldsymbol{\theta}_t)$, respectively. Then, we solve a system of inequalities to establish finite-time convergence.

*Step 1: Reward estimation error analysis.* Using the reward estimator update rule (Line 7 of Algorithm 1), we decompose the reward estimation error into:

$$
\begin{aligned}
y_{t+1}^2 = (1 - 2\gamma)y_t^2 &+ 2\gamma y_t(r_t - J(\boldsymbol{\theta}_t)) \\
&+ 2y_t(J(\boldsymbol{\theta}_t) - J(\boldsymbol{\theta}_{t+1})) + (J(\boldsymbol{\theta}_t) - J(\boldsymbol{\theta}_{t+1}) + \gamma(r_t - \eta_t))^2.
\end{aligned}
\tag{16}
$$

The second term on the right-hand side of Eq. (16) is a bias term caused by the Markovian sample, which requires **characterizing the distance between stationary distributions under continuous state and action spaces** as shown in Lemma C.1. This error term is further handled in Lemma F.1. The third term captures the variation of the moving targets $J(\boldsymbol{\theta}_t)$ tracked by the reward estimation error. We employ the smoothness of $J(\boldsymbol{\theta})$ (see Lemma C.4) and derive an implicit upper bound for this term as a function of the norm of $y_t$ and $\nabla J(\boldsymbol{\theta}_t)$. This bound will be combined with the implicit bounds derived in Step 2 and Step 3 below to establish the non-asymptotic convergence altogether. The last term in Eq. (16) reflects the variance in reward estimation, which is bounded by $\mathcal{O}(\gamma)$ after utilizing the Lipschitzness of $J(\boldsymbol{\theta})$ in Lemma C.3.

*Step 2: Critic error analysis.* Using the critic update rule (Line 8 of Algorithm 1), we decompose the squared error by (we neglect the projection for the time being for the ease of comprehension. The complete analysis can be found in the appendix.)

$$
\begin{aligned}
\|\boldsymbol{z}_{t+1}\|^2 =& \|\boldsymbol{z}_t\|^2 + 2\beta\langle \boldsymbol{z}_t, \bar{g}(\boldsymbol{\omega}_t, \boldsymbol{\theta}_t)\rangle + 2\beta\Psi(O_t, \boldsymbol{\omega}_t, \boldsymbol{\theta}_t) + 2\beta\langle \boldsymbol{z}_t, \Delta g(O_t, \eta_t, \boldsymbol{\theta}_t)\rangle \\
&+ 2\langle \boldsymbol{z}_t, \boldsymbol{\omega}_t^* - \boldsymbol{\omega}_{t+1}^* \rangle + \|\boldsymbol{\omega}_t^* - \boldsymbol{\omega}_{t+1}^* + \beta(g(O_t, \boldsymbol{\omega}_t, \boldsymbol{\theta}_t) + \Delta g(O_t, \eta_t, \boldsymbol{\theta}_t))\|^2,
\end{aligned}
\tag{17}
$$

where $O_t := (s_t, a_t, s_{t+1})$ denotes the tuple generated from the original Markov chain in Eq. (9) and the definitions of $g, \bar{g}, \Delta g$, and $\Psi$ can be found in Eq. (10) and Eq. (11) in Appendix B. Without diving into the detailed definitions, here we focus on illustrating the high-level insights of our proof. First of all, the second term on the right-hand side of Eq. (17) is the inner product between the critic error $\boldsymbol{z}_t$ and the critic's mean-path update $\bar{g}(\boldsymbol{\omega}_t, \boldsymbol{\theta}_t)$, which serves as the key to the convergence. Our analysis for this term is **distinct from all previous results** since considering continuous spaces and deep neural networks substantially complicate the bounding process. we employ the Bellman equation and neural network approximation to manage error propagation and control the error by leveraging the **approximation capability of the neural network** (Eq. (7)), the **linearity of wide networks** (third inequality in Lemma C.5), and **sufficient policy exploration** (see Eq. (22)). It provides an explicit characterization of how sufficient exploration can help the convergence of learning.

The third term is a Markovian noise, which is again **characterized by the distance between stationary distributions under continuous state and action spaces** and further bounded implicitly in Lemma F.3. The fourth term is caused by inaccurate reward and critic estimations, which can be bounded by the norm of $y_t$ and $z_t$ after applying **the Lipschitzness of $\widehat{V}$ as shown in Lemma C.5**. The fifth term tracks both the critic estimation performance $z_t$ and the difference between the drifting critic targets $\boldsymbol{\omega}_t^*$. Similar to the case of Step 1, we establish an implicit upper bound for this term as a function of $y_t$ and $z_t$ by utilizing the smoothness of the optimal critic proved in Assumption 4.3. Finally, the last term reflects the variances of various estimations, which is bounded by $\mathcal{O}(\beta)$.

*Step 3: Policy gradient norm analysis.* Using the actor update rule (Line 9 of Algorithm 1) and the smoothness property of $J(\boldsymbol{\theta})$ (see Lemma C.4), we derive

$$
\begin{aligned}
\|\nabla J(\boldsymbol{\theta}_t)\|^2 \leq {}& \frac{1}{\alpha}(J(\boldsymbol{\theta}_{t+1}) - J(\boldsymbol{\theta}_t)) + \Theta(O_t, \boldsymbol{\theta}_t) - \langle \nabla J(\boldsymbol{\theta}_t), \Delta h(O_t, \eta_t, \boldsymbol{\omega}_t, \boldsymbol{\theta}_t)\rangle \\
& - \langle \nabla J(\boldsymbol{\theta}_t), \mathbb{E}_{O_t'}[\Delta h'(O_t', \boldsymbol{\theta}_t)]\rangle + \alpha \frac{L_{J'}}{2}\|\delta_t \nabla \log \pi_{\boldsymbol{\theta}_t}(a_t|s_t)\|^2,
\end{aligned}
\tag{18}
$$

where $O_t'$ is a shorthand for an independent sample from stationary distribution $s \sim \mu_{\boldsymbol{\theta}_t}, a \sim \pi_{\boldsymbol{\theta}_t}, s' \sim \mathcal{P}(\cdot|s,a)$, $\Theta$ is defined in Eq. (11), and $L_{J'}$ is a constant. The first term on the right-hand side of Eq. (18) compares the actor's performances between consecutive updates, which can be bounded via Abel summation by parts. The second term is a noise term introduced by Markovian sampling, which is **characterized by the distance between stationary distributions under continuous state and action spaces** and handled in Lemma F.6. The third term is an error introduced by the inaccurate estimations of both the time-average reward and the critic. After employing the **the Lipschitzness of $\widehat{V}$ as shown in Lemma C.5**, we control this term by providing an implicit bound depending on $y_t$, $z_t$, and $\nabla J(\boldsymbol{\theta}_t)$. The fourth term comes from the linear function approximation error. The final term represents the variance of the stochastic gradient update, which is controlled by $\mathcal{O}(\alpha)$ due to the **boundedness of $\widehat{V}$, a result we specifically derived in Lemma C.5**.

*Step 4: Interconnected iteration system analysis.* Taking the expectation of and summing Eq. (16), Eq. (17), and Eq. (18) from $\tau_T$ to $T - 1$, respectively, we obtain the following system of inequalities in terms of $Y_T$, $Z_T$, $G_T$:

$$
Y_T \leq \mathcal{O}(\frac{\log^2 T}{\sqrt{T}}) + l_1\sqrt{Y_T G_T},
$$

$$
Z_T \leq \mathcal{O}(\frac{\log^2 T}{\sqrt{T}}) + \mathcal{O}(\epsilon_{\text{app}}) + \widetilde{\mathcal{O}}(\frac{1}{\sqrt{m}}) + l_2\sqrt{Y_T Z_T} + l_3\sqrt{Z_T(2Y_T + l_4 Z_T)} + l_5\sqrt{Z_T G_T},
$$

$$
G_T \leq \mathcal{O}(\frac{\log^2 T}{\sqrt{T}}) + \mathcal{O}(\epsilon_{\text{app}}) + l_6\sqrt{G_T(2Y_T + l_4 Z_T)}.
$$

where $l_1, l_2, l_3, l_4, l_5, l_6$ are positive constants. By solving the above system of inequalities, we further prove that if

$$
(1 + \frac{1}{2}l_4)l_3 \leq \frac{1}{4}, \ 2l_4 l_5^2 l_6^2 \leq \frac{1}{2}, \ l_1(1 + 2l_6^2 + 4l_4 l_6^2(l_2^2 + l_3 + 2l_5^2 l_6^2)) \leq 1,
$$

then $Y_T, Z_T, G_T$ converge at a rate of $\mathcal{O}(\frac{\log^2 T}{\sqrt{T}}) + \mathcal{O}(\epsilon_{\text{app}}) + \widetilde{\mathcal{O}}(\frac{1}{\sqrt{m}})$. This condition can be easily satisfied by choosing the stepsize ratio $c$ to be smaller than a threshold identified in Equation (34). Thus, it completes the proof.

# F   PROOF OF MAIN THEOREM

In this section, we aim to show the proof of Theorem 4.9. Define $U_\delta := 2U_r + 2U_{\boldsymbol{\omega}} + 2U_v$ so that we have $|\delta_t| \leq U_\delta$, where $U_v$ is defined in Lemma C.5 and $\delta_t$ is the TD error which comes from Line 6 in Algorithm 1. Note that from Assumption 4.7, we have $\|\delta \nabla \log \pi_{\boldsymbol{\theta}}\| \leq G := U_\delta B$. The norm of $\boldsymbol{\omega}$ is defined by $\|\boldsymbol{\omega}\| =: (\sum_{k=1}^{K} \|\boldsymbol{W}^{(k)}\|_{\text{F}}^2)^{1/2}$, where $\|\cdot\|_{\text{F}}$ is the Frobenius norm of a matrix.

We decompose the whole proof into four steps.

## F.1  STEP 1: REWARD ESTIMATION ERROR ANALYSIS

In this subsection, we will establish an implicit bound for estimator.

**Lemma F.1.** *From any $t \geq \tau > 0$, we have*

$$\mathbb{E}[\Phi(O_t, \eta_t, \boldsymbol{\theta}_t)] \leq 4U_r L_J \|\boldsymbol{\theta}_t - \boldsymbol{\theta}_{t-\tau}\| + 2U_r |\eta_t - \eta_{t-\tau}|$$

$$+ 2U_r^2 L_\pi \sum_{i=t-\tau}^{t} \mathbb{E}\|\boldsymbol{\theta}_i - \boldsymbol{\theta}_{t-\tau}\| + 4U_r^2 C \rho^{\tau-1}.$$

**Theorem F.2.** *Choose $\alpha = \frac{c}{\sqrt{T}}, \beta = \gamma = \frac{1}{\sqrt{T}}$, we have*

$$Y_T \leq \mathcal{O}(\frac{\log^2 T}{\sqrt{T}}) + cG\sqrt{Y_T G_T}. \tag{19}$$

*Proof.* From the update rule of reward estimator in Line 7 of Algorithm 1, we have

$$\eta_{t+1} - J(\boldsymbol{\theta}_{t+1}) = \eta_t - J(\boldsymbol{\theta}_t) + J(\boldsymbol{\theta}_t) - J(\boldsymbol{\theta}_{t+1}) + \gamma(r_t - \eta_t),$$

which implies

$$\begin{aligned}
y_{t+1}^2 &= (y_t + J(\boldsymbol{\theta}_t) - J(\boldsymbol{\theta}_{t+1}) + \gamma(r_t - \eta_t))^2 \\
&\leq y_t^2 + 2y_t(J(\boldsymbol{\theta}_t) - J(\boldsymbol{\theta}_{t+1})) + 2\gamma y_t(r_t - \eta_t) \\
&\quad + 2(J(\boldsymbol{\theta}_t) - J(\boldsymbol{\theta}_{t+1}))^2 + 2\gamma^2(r_t - \eta_t)^2 \\
&= (1 - 2\gamma)y_t^2 + 2\gamma y_t(r_t - J(\boldsymbol{\theta}_t)) + 2y_t(J(\boldsymbol{\theta}_t) - J(\boldsymbol{\theta}_{t+1})) \\
&\quad + 2(J(\boldsymbol{\theta}_t) - J(\boldsymbol{\theta}_{t+1}))^2 + 2\gamma^2(r_t - \eta_t)^2.
\end{aligned}$$

Taking expectation up to $s_{t+1}$ (the whole trajectory), rearranging and summing from $\tau_T$ to $T-1$, we have

$$\sum_{t=\tau_T}^{T-1} \mathbb{E}[y_t^2] \leq \underbrace{\sum_{t=\tau_T}^{T-1} \frac{1}{2\gamma} \mathbb{E}(y_t^2 - y_{t+1}^2)}_{I_1} + \underbrace{\sum_{t=\tau_T}^{T-1} \mathbb{E}[y_t(r_t - J(\boldsymbol{\theta}_t))]}_{I_2} + \underbrace{\sum_{t=\tau_T}^{T-1} \frac{1}{\gamma} \mathbb{E}[y_t(J(\boldsymbol{\theta}_t) - J(\boldsymbol{\theta}_{t+1})]}_{I_3}$$

$$+ \underbrace{\sum_{t=\tau_T}^{T-1} \frac{1}{\gamma} \mathbb{E}[(J(\boldsymbol{\theta}_t) - J(\boldsymbol{\theta}_{t+1}))^2]}_{I_4} + \underbrace{\sum_{t=\tau_T}^{T-1} \gamma \mathbb{E}[(r_t - \eta_t)^2]}_{I_5}.$$

For term $I_1$, from Abel summation by parts, we have

$$\begin{aligned}
I_1 &= \sum_{t=\tau_T}^{T-1} \frac{1}{2\gamma} \mathbb{E}(y_t^2 - y_{t+1}^2) \\
&\leq \frac{2U_r^2}{\gamma} \\
&= 2U_r^2 \sqrt{T}.
\end{aligned}$$

For term $I_2$, from Lemma F.1, we have

$$\begin{aligned}
\mathbb{E}[y_t(r_t - J(\boldsymbol{\theta}_t))] &\leq 4U_r L_J \|\boldsymbol{\theta}_t - \boldsymbol{\theta}_{t-\tau}\| + 2U_r |\eta_t - \eta_{t-\tau}| \\
&\quad + 2U_r^2 L_\pi \sum_{i=t-\tau}^{t} \mathbb{E}\|\boldsymbol{\theta}_i - \boldsymbol{\theta}_{t-\tau}\| + 4U_r^2 C \rho^{\tau-1} \\
&\leq 4U_r L_J G\tau\alpha + 4U_r^2 \tau\gamma + 2U_r^2 L_\pi \tau(\tau+1)G\alpha + 4U_r^2 C \rho^{\tau-1} \\
&\leq (4U_r L_J G\tau + 2U_r^2 L_\pi G\tau(\tau+1))\alpha + 4U_r^2 \tau\gamma + 4U_r^2 C \rho^{\tau-1}.
\end{aligned}$$

Choose $\tau = \tau_T$, we have

$$I_2 = \sum_{t=\tau_T}^{T-1} \mathbb{E}[y_t(r_t - J(\boldsymbol{\theta}_t))]$$

$$\leq (4U_r L_J G \tau_T + 2U_r^2 L_\pi G \tau_T (\tau_T + 1)) \sum_{t=\tau_T}^{T-1} \alpha$$

$$+ 4U_r^2 \tau_T \sum_{t=\tau_T}^{T-1} \gamma + 4U_r^2 \sum_{t=\tau_T}^{T-1} \frac{1}{\sqrt{T}}$$

$$= (4cU_r L_J G \tau_T + 2cU_r^2 L_\pi G \tau_T (\tau_T + 1) + 4U_r^2 \tau_T + 4U_r^2) \frac{T - \tau_T}{\sqrt{T}}.$$

For $I_3$, if $y_t > 0$, from Eq. (14), we have

$$y_t(J(\boldsymbol{\theta}_t) - J(\boldsymbol{\theta}_{t+1})) \leq y_t(\frac{L_{J'}}{2}\|\boldsymbol{\theta}_t - \boldsymbol{\theta}_{t+1}\|^2 + \langle \nabla J(\boldsymbol{\theta}_t), \boldsymbol{\theta}_t - \boldsymbol{\theta}_{t+1}\rangle)$$

$$\leq L_{J'} U_r \|\boldsymbol{\theta}_t - \boldsymbol{\theta}_{t+1}\|^2 + |y_t|\|\boldsymbol{\theta}_t - \boldsymbol{\theta}_{t+1}\|\|\nabla J(\boldsymbol{\theta}_t)\|.$$

If $y_t \leq 0$, from Eq. (15), we have

$$y_t(J(\boldsymbol{\theta}_t) - J(\boldsymbol{\theta}_{t+1})) \leq y_t(-\frac{L_{J'}}{2}\|\boldsymbol{\theta}_t - \boldsymbol{\theta}_{t+1}\|^2 + \langle \nabla J(\boldsymbol{\theta}_t), \boldsymbol{\theta}_t - \boldsymbol{\theta}_{t+1}\rangle)$$

$$\leq L_{J'} U_r \|\boldsymbol{\theta}_t - \boldsymbol{\theta}_{t+1}\|^2 + |y_t|\|\boldsymbol{\theta}_t - \boldsymbol{\theta}_{t+1}\|\|\nabla J(\boldsymbol{\theta}_t)\|.$$

Overall, we get

$$I_3 = \sum_{t=\tau_T}^{T-1} \frac{1}{\gamma}\mathbb{E}[y_t(J(\boldsymbol{\theta}_t) - J(\boldsymbol{\theta}_{t+1}))]$$

$$\leq \sum_{t=\tau_T}^{T-1} \frac{1}{\gamma}\mathbb{E}[L_{J'} U_r \|\boldsymbol{\theta}_t - \boldsymbol{\theta}_{t+1}\|^2 + |y_t|\|\boldsymbol{\theta}_t - \boldsymbol{\theta}_{t+1}\|\|\nabla J(\boldsymbol{\theta}_t)\|]$$

$$\leq \sum_{t=\tau_T}^{T-1} \mathbb{E}[cL_{J'} U_r G^2 \alpha + cG|y_t|\|\nabla J(\boldsymbol{\theta}_t)\|]$$

$$\leq c^2 L_{J'} U_r G^2 \frac{T - \tau_T}{\sqrt{T}} + cG(\sum_{t=\tau_T}^{T-1} \mathbb{E}y_t^2)^{\frac{1}{2}}(\sum_{t=\tau_T}^{T-1} \mathbb{E}\|\nabla J(\boldsymbol{\theta}_t)\|^2)^{\frac{1}{2}}.$$

For term $I_4$, we have

$$I_4 = \sum_{t=\tau_T}^{T-1} \frac{1}{\gamma}\mathbb{E}[(J(\boldsymbol{\theta}_t) - J(\boldsymbol{\theta}_{t+1}))^2]$$

$$\leq \sum_{t=\tau_T}^{T-1} \frac{1}{\gamma}L_J^2 \mathbb{E}\|\boldsymbol{\theta}_t - \boldsymbol{\theta}_{t+1}\|^2$$

$$\leq \sum_{t=\tau_T}^{T-1} \frac{1}{\gamma}L_J^2 G^2 \alpha^2 = L_J^2 G^2 c^2 \frac{T - \tau_T}{\sqrt{T}}.$$

For term $I_5$, we have

$$I_5 = \sum_{t=\tau_T}^{T-1} \gamma \mathbb{E}[(r_t - J(\boldsymbol{\theta}_t))^2]$$

$$\leq \sum_{t=\tau_T}^{T-1} 4U_r^2 \gamma = 4U_r^2 \frac{T - \tau_T}{\sqrt{T}}.$$

Therefore, we get

$$\sum_{t=\tau_T}^{T-1} \mathbb{E}[y_t^2] \leq (4cU_rL_JG\tau_T + 2cU_r^2L_\pi G\tau_T(\tau_T + 1)$$

$$+ 4U_r^2(\tau_T + 2) + c^2G^2(L_{J'}U_r + L_J^2))\frac{T - \tau_T}{\sqrt{T}}$$

$$+ 2U_r^2\sqrt{T} + cG(\sum_{t=\tau_T}^{T-1} \mathbb{E}y_t^2)^{\frac{1}{2}}(\sum_{t=\tau_T}^{T-1} \mathbb{E}\|\nabla J(\boldsymbol{\theta}_t)\|^2)^{\frac{1}{2}}.$$

Since $\tau_T = \mathcal{O}(\log T)$, we have $\frac{\sqrt{T}}{T-\tau_T} \leq \frac{2}{\sqrt{T}}$ for large $T$. Then we get

$$\frac{1}{T - \tau_T}\sum_{t=\tau_T}^{T-1} \mathbb{E}[y_t^2] \leq (4cU_rL_JG\tau_T + 2cU_r^2L_\pi G\tau_T(\tau_T + 1)$$

$$+ 4U_r^2(\tau_T + 3) + c^2G^2(L_{J'}U_r + L_J^2))\frac{1}{\sqrt{T}}$$

$$+ cG(\frac{1}{T - \tau_T}\sum_{t=\tau_T}^{T-1} \mathbb{E}y_t^2)^{\frac{1}{2}}(\frac{1}{T - \tau_T}\sum_{t=\tau_T}^{T-1} \mathbb{E}\|\nabla J(\boldsymbol{\theta}_t)\|^2)^{\frac{1}{2}}$$

$$= \mathcal{O}(\frac{\log^2 T}{\sqrt{T}}) + cG(\frac{1}{T - \tau_T}\sum_{t=\tau_T}^{T-1} \mathbb{E}y_t^2)^{\frac{1}{2}}(\frac{1}{T - \tau_T}\sum_{t=\tau_T}^{T-1} \mathbb{E}\|\nabla J(\boldsymbol{\theta}_t)\|^2)^{\frac{1}{2}}.$$

Thus we finish the proof. $\qquad\square$

### F.2 STEP 2: CRITIC ERROR ANALYSIS

In this subsection, we will establish an implicit upper bound for critic.

**Lemma F.3.** *For any $t \geq \tau > 0$, we have*

$$\mathbb{E}[\Psi(O_t, \boldsymbol{\omega}_t, \boldsymbol{\theta}_t)] \leq C_1\|\boldsymbol{\theta}_t - \boldsymbol{\theta}_{t-\tau}\| + C_2\|\boldsymbol{\omega}_t - \boldsymbol{\omega}_{t-\tau}\| + U_\delta^2 L_v L_\pi G\tau(\tau + 1)\alpha + 2U_\delta^2 L_v C\rho^{\tau-1},$$

*where*

$$C_1 = 2U_\delta^2 L_\pi(1 + \lceil\log_\rho C^{-1}\rceil + \frac{1}{1 - \rho}) + 2U_\delta L_J L_v + 2U_\delta L_* L_v,$$

$$C_2 = 2U_\delta(U_v H_v + L_v^2 + U_r H_v + L_v).$$

**Lemma F.4.** *For any $t \geq \tau > 0$, we have*

$$\mathbb{E}[\Xi(O_t, \boldsymbol{\omega}_t, \boldsymbol{\theta}_t)] \leq C_3\|\boldsymbol{\theta}_t - \boldsymbol{\theta}_{t-\tau}\| + 2U_\delta BL_*\|\boldsymbol{\omega}_t - \boldsymbol{\omega}_{t-\tau}\|$$

$$+ 2U_\delta^2 BL_* L_\pi G\tau(\tau + 1)\alpha + 4U_\delta^2 BL_* C\rho^{\tau-1}.$$

*where $C_3 := 3U_\delta L_*(U_\delta L_l + 4BU_\delta L_J + 2BL_v L_*) + 2U_\delta BL_*^2 + 2U_\delta^2 BL_s$.*

**Theorem F.5.** *Choose $\alpha = \frac{c}{\sqrt{T}}, \beta = \gamma = \frac{1}{\sqrt{T}}$, we have*

$$Z_T \leq \mathcal{O}(\frac{\log^2 T}{\sqrt{T}}) + \widetilde{\mathcal{O}}(\frac{1}{\sqrt{m}}) + \mathcal{O}(\epsilon_{\text{app}}) + \frac{2U_v}{\lambda}\sqrt{Y_T Z_T}$$

$$+ \frac{2cBL_*}{\lambda}\sqrt{Z_T(2Y_T + 8L_v^2 Z_T)} + \frac{2cL_*}{\lambda}\sqrt{Z_T G_T} \qquad (20)$$

*Proof.* From the update rule of critic in Line 8 of Algorithm 1, we have

$$\|\boldsymbol{\omega}_{t+1} - \boldsymbol{\omega}_{t+1}^*\| = \|\Pi_{U_\omega}(\boldsymbol{\omega}_t + \beta\delta_t\nabla_{\boldsymbol{\omega}}\widehat{V}(\boldsymbol{\omega}_t; s_t)) - \boldsymbol{\omega}_{t+1}^*\|$$

$$= \|\Pi_{U_\omega}(\boldsymbol{\omega}_t + \beta\delta_t\nabla_{\boldsymbol{\omega}}\widehat{V}(\boldsymbol{\omega}_t; s_t)) - \Pi_{U_\omega}(\boldsymbol{\omega}_{t+1}^*)\|$$

$$\leq \|\boldsymbol{\omega}_t + \beta\delta_t\nabla_{\boldsymbol{\omega}}\widehat{V}(\boldsymbol{\omega}_t; s_t) - \boldsymbol{\omega}_{t+1}^*\|$$

$$= \|\boldsymbol{\omega}_t - \boldsymbol{\omega}_t^* + \boldsymbol{\omega}_t^* - \boldsymbol{\omega}_{t+1}^* + \beta\delta_t\nabla_{\boldsymbol{\omega}}\widehat{V}(\boldsymbol{\omega}_t; s_t)\|$$

Therefore, we have

$$
\begin{aligned}
\|\boldsymbol{z}_{t+1}\|^2 &= \|\boldsymbol{z}_t + \beta(g(O_t, \boldsymbol{\omega}_t, \boldsymbol{\theta}_t) + \Delta g(O_t, \eta_t, \boldsymbol{\theta}_t)) + \boldsymbol{\omega}_t^* - \boldsymbol{\omega}_{t+1}^*\|^2 \\
&= \|\boldsymbol{z}_t\|^2 + 2\beta\langle \boldsymbol{z}_t, g(O_t, \boldsymbol{\omega}_t, \boldsymbol{\theta}_t)\rangle + 2\beta\langle \boldsymbol{z}_t, \Delta g(O_t, \eta_t, \boldsymbol{\theta}_t)\rangle \\
&\quad + 2\langle \boldsymbol{z}_t, \boldsymbol{\omega}_t^* - \boldsymbol{\omega}_{t+1}^*\rangle + \|\beta(g(O_t, \boldsymbol{\omega}_t, \boldsymbol{\theta}_t) + \Delta g(O_t, \eta_t, \boldsymbol{\theta}_t)) + \boldsymbol{\omega}_t^* - \boldsymbol{\omega}_{t+1}^*\|^2 \\
&= \|\boldsymbol{z}_t\|^2 + 2\beta\langle \boldsymbol{z}_t, \bar{g}(\boldsymbol{\omega}_t, \boldsymbol{\theta}_t)\rangle + 2\beta\Psi(O_t, \boldsymbol{\omega}_t, \boldsymbol{\theta}_t) + 2\beta\langle \boldsymbol{z}_t, \Delta g(O_t, \eta_t, \boldsymbol{\theta}_t)\rangle \qquad (21) \\
&\quad + 2\langle \boldsymbol{z}_t, \boldsymbol{\omega}_t^* - \boldsymbol{\omega}_{t+1}^*\rangle + \|\beta(g(O_t, \boldsymbol{\omega}_t, \boldsymbol{\theta}_t) + \Delta g(O_t, \eta_t, \boldsymbol{\theta}_t)) + \boldsymbol{\omega}_t^* - \boldsymbol{\omega}_{t+1}^*\|^2 \\
&\leq \|\boldsymbol{z}_t\|^2 + 2\beta\langle \boldsymbol{z}_t, \bar{g}(\boldsymbol{\omega}_t, \boldsymbol{\theta}_t)\rangle + 2\beta\Psi(O_t, \boldsymbol{\omega}_t, \boldsymbol{\theta}_t) + 2\beta\langle \boldsymbol{z}_t, \Delta g(O_t, \eta_t, \boldsymbol{\theta}_t)\rangle \\
&\quad + 2\langle \boldsymbol{z}_t, \boldsymbol{\omega}_t^* - \boldsymbol{\omega}_{t+1}^*\rangle + 2U_\delta^2 L_v^2\beta^2 + 2\|\boldsymbol{\omega}_t^* - \boldsymbol{\omega}_{t+1}^*\|^2.
\end{aligned}
$$

We then analyse the mean-path update $\bar{g}(\boldsymbol{\omega}_t, \boldsymbol{\theta}_t)$. From the definition in Eq. (10), we have

$$
\begin{aligned}
\bar{g}(\boldsymbol{\omega}_t, \boldsymbol{\theta}_t) &:= \mathbb{E}_{s_t, a_t, s_{t+1}}[(r(s_t, a_t) - J(\boldsymbol{\theta}_t) + \widehat{V}(\boldsymbol{\omega}_t; s_{t+1}) - \widehat{V}(\boldsymbol{\omega}_t; s_t))\nabla_{\boldsymbol{\omega}}\widehat{V}(\boldsymbol{\omega}_t; s_t)] \\
&\overset{(1)}{=} \mathbb{E}_{s_t, a_t, s_{t+1}}[(V(s_t) - V(s_{t+1}) + \widehat{V}(\boldsymbol{\omega}_t; s_{t+1}) - \widehat{V}(\boldsymbol{\omega}_t; s_t))\nabla_{\boldsymbol{\omega}}\widehat{V}(\boldsymbol{\omega}_t; s_t)] \\
&= \mathbb{E}_{s_t}[(V(s_t) - \widehat{V}(\boldsymbol{\omega}_t, s_t) - \mathbb{E}_{s_{t+1}, a_t}[V(s_{t+1}) - \widehat{V}(\boldsymbol{\omega}_t, s_{t+1})|s_t])\nabla_{\boldsymbol{\omega}}\widehat{V}(\boldsymbol{\omega}_t; s_t)]
\end{aligned}
$$

where (1) comes from the Bellman equation. For $\mathbb{E}_{s_{t+1}, a_t}[V(s_{t+1}) - \widehat{V}(\boldsymbol{\omega}_t, s_{t+1})|s_t]$, it can be shown that

$$
\begin{aligned}
&\mathbb{E}_{s_{t+1}, a_t}[V(s_{t+1}) - \widehat{V}(\boldsymbol{\omega}_t, s_{t+1})|s_t] \\
&= \int_{\mathcal{S}}\int_{\mathcal{A}} \pi_{\boldsymbol{\theta}_t}(a_t|s_t)\mathcal{P}(s_{t+1}|s_t, a_t)(V(s_{t+1}) - \widehat{V}(\boldsymbol{\omega}_t; s_{t+1}))d(a_t \times s_{t+1}).
\end{aligned}
$$

By the definition of operator $P_{\boldsymbol{\theta}}$, we have

$$
P_{\boldsymbol{\theta}}(V(s) - \widehat{V}(\boldsymbol{\omega}, s)) = \int_{\mathcal{S}}\int_{\mathcal{A}} \pi_{\boldsymbol{\theta}}(a|s)\mathcal{P}(s'|s, a)(V(s') - \widehat{V}(\boldsymbol{\omega}; s'))d(a \times s').
$$

Then for $\bar{g}(\boldsymbol{\omega}_t, \boldsymbol{\theta}_t)$, it follows that

$$
\bar{g}(\boldsymbol{\omega}_t, \boldsymbol{\theta}_t) = \mathbb{E}_{s_t}[(I - P_{\boldsymbol{\theta}_t})(V(s_t) - \widehat{V}(\boldsymbol{\omega}_t, s_t))\nabla_{\boldsymbol{\omega}}\widehat{V}(\boldsymbol{\omega}_t; s_t)],
$$

where $I$ is the identity operator. Therefore, we have

$$
\begin{aligned}
\langle \boldsymbol{z}_t, \bar{g}(\boldsymbol{\omega}_t, \boldsymbol{\theta}_t)\rangle =& \mathbb{E}\langle \boldsymbol{z}_t, (I - P_{\boldsymbol{\theta}_t})(V(s_t) - \widehat{V}(\boldsymbol{\omega}_t; s_t))\nabla_{\boldsymbol{\omega}}\widehat{V}(\boldsymbol{\omega}_t; s_t)\rangle \\
=& \mathbb{E}\langle \boldsymbol{z}_t, (I - P_{\boldsymbol{\theta}_t})(V(s_t) - \widehat{V}(\boldsymbol{\omega}_t^*; s_t) + \widehat{V}(\boldsymbol{\omega}_t^*; s_t) - \widehat{V}(\boldsymbol{\omega}_t; s_t))\nabla_{\boldsymbol{\omega}}\widehat{V}(\boldsymbol{\omega}_t; s_t)\rangle \\
=& \mathbb{E}\langle \boldsymbol{z}_t, (I - P_{\boldsymbol{\theta}_t})(V(s_t) - \widehat{V}(\boldsymbol{\omega}_t^*; s_t))\nabla_{\boldsymbol{\omega}}\widehat{V}(\boldsymbol{\omega}_t; s_t)\rangle \\
&+ \mathbb{E}\langle \boldsymbol{z}_t, (I - P_{\boldsymbol{\theta}_t})(\widehat{V}(\boldsymbol{\omega}_t^*; s_t) - \widehat{V}(\boldsymbol{\omega}_t; s_t))\nabla_{\boldsymbol{\omega}}\widehat{V}(\boldsymbol{\omega}_t; s_t)\rangle \\
=& 4U_{\boldsymbol{\omega}} L_v \epsilon_{\mathrm{app}} + \mathbb{E}[(\boldsymbol{z}_t^\top \nabla_{\boldsymbol{\omega}}\widehat{V}(\boldsymbol{\omega}_t; s_t) + (\widehat{V}(\boldsymbol{\omega}_t^*; s_t) - \widehat{V}(\boldsymbol{\omega}_t; s_t)) \\
&- (\widehat{V}(\boldsymbol{\omega}_t^*; s_t) - \widehat{V}(\boldsymbol{\omega}_t, s_t)))(I - P_{\boldsymbol{\theta}_t})(\widehat{V}(\boldsymbol{\omega}_t^*; s_t) - \widehat{V}(\boldsymbol{\omega}_t; s_t))] \\
\overset{(1)}{=}& \mathbb{E}[\boldsymbol{z}_t^\top(\nabla_{\boldsymbol{\omega}}\widehat{V}(\boldsymbol{\omega}_t; s_t) - \nabla_{\boldsymbol{\omega}}\widehat{V}(\boldsymbol{\omega}_{\mathrm{mid}}; s_t))(I - P_{\boldsymbol{\theta}_t})(\widehat{V}(\boldsymbol{\omega}_t^*; s_t) - \widehat{V}(\boldsymbol{\omega}_t; s_t))] \\
&- \langle \widehat{V}(\boldsymbol{\omega}_t^*) - \widehat{V}(\boldsymbol{\omega}_t), D_{\boldsymbol{\theta}}(I - P_{\boldsymbol{\theta}_t})(\widehat{V}(\boldsymbol{\omega}_t^*) - \widehat{V}(\boldsymbol{\omega}_t))\rangle + 2U_\delta L_v \epsilon_{\mathrm{app}} \\
\overset{(2)}{\leq}& -\lambda_1^2\lambda_2\|\boldsymbol{z}_t\|^2 + 2L_v H_v\|\boldsymbol{z}_t\|^3 + 2U_\delta L_v \epsilon_{\mathrm{app}}
\end{aligned}
$$
(22)

where (1) comes from the mean-value theorem with $\boldsymbol{\omega}_{\mathrm{mid}} = \lambda_3\boldsymbol{\omega}_t + (1 - \lambda_3)\boldsymbol{\omega}_t^*$ where $\lambda_3 \in [0, 1]$; (2) follows from Assumption 4.4 and Assumption 4.5. Hereafter, we define $\lambda := \lambda_1^2\lambda_2$.

Substituting the above result into Eq. (21), it holds that

$$
\begin{aligned}
\|\boldsymbol{z}_{t+1}\|^2 \leq& \|\boldsymbol{z}_t\|^2 - 2\lambda\beta\|\boldsymbol{z}_t\|^2 + 2\beta\Psi(O_t, \boldsymbol{\omega}_t, \boldsymbol{\theta}_t) + 2\beta\langle \boldsymbol{z}_t, \Delta g(O_t, \eta_t, \boldsymbol{\theta}_t)\rangle \\
&+ 2\langle \boldsymbol{z}_t, \boldsymbol{\omega}_t^* - \boldsymbol{\omega}_{t+1}^*\rangle + 2\|\boldsymbol{\omega}_t^* - \boldsymbol{\omega}_{t+1}^*\|^2 + 2U_\delta^2\beta^2 + 4\beta L_v H_v U_\delta^3 + 4U_\delta L_v \beta\epsilon_{\mathrm{app}}
\end{aligned}
$$

Taking expectation up to $s_{t+1}$, we have

$$
\begin{aligned}
\mathbb{E}\|\boldsymbol{z}_{t+1}\|^2 &\leq (1-2\lambda\beta)\mathbb{E}\|\boldsymbol{z}_t\|^2 + 2\beta\mathbb{E}\Psi(O_t,\boldsymbol{\omega}_t,\boldsymbol{\theta}_t) + 2\beta\mathbb{E}\langle\boldsymbol{z}_t,\Delta g(O_t,\eta_t,\boldsymbol{\theta}_t)\rangle \\
&\quad +2\mathbb{E}\langle\boldsymbol{z}_t,\boldsymbol{\omega}_t^* - \boldsymbol{\omega}_{t+1}^*\rangle + 2\mathbb{E}\|\boldsymbol{\omega}_t^* - \boldsymbol{\omega}_{t+1}^*\|^2 + 2U_\delta^2\beta^2 + 4\beta L_v H_v U_\delta^3 + 4U_\delta L_v\beta\epsilon_{\text{app}} \\
&\leq (1-2\lambda\beta)\mathbb{E}\|\boldsymbol{z}_t\|^2 + 2\beta\mathbb{E}\Psi(O_t,\boldsymbol{\omega}_t,\boldsymbol{\theta}_t) + 2\beta\mathbb{E}\langle\boldsymbol{z}_t,\Delta g(O_t,\eta_t,\boldsymbol{\theta}_t)\rangle \\
&\quad + 2\mathbb{E}\langle\boldsymbol{z}_t,\boldsymbol{\omega}_t^* - \boldsymbol{\omega}_{t+1}^*\rangle + 2U_\delta^2\beta^2 + 2\mathbb{E}\|\boldsymbol{\omega}_t^* - \boldsymbol{\omega}_{t+1}^*\|^2 + 4\beta L_v H_v U_\delta^3 + 4U_\delta L_v\beta\epsilon_{\text{app}} \\
&\leq (1-2\lambda\beta)\mathbb{E}\|\boldsymbol{z}_t\|^2 + 2\beta\mathbb{E}\Psi(O_t,\boldsymbol{\omega}_t,\boldsymbol{\theta}_t) + 2\beta\mathbb{E}\langle\boldsymbol{z}_t,\Delta g(O_t,\eta_t,\boldsymbol{\theta}_t)\rangle \\
&\quad + 2\mathbb{E}\langle\boldsymbol{z}_t,\boldsymbol{\omega}_t^* - \boldsymbol{\omega}_{t+1}^* + (\nabla\boldsymbol{\omega}_t^*)^\top(\boldsymbol{\theta}_{t+1}-\boldsymbol{\theta}_t)\rangle + 2\mathbb{E}\langle\boldsymbol{z}_t,(\nabla\boldsymbol{\omega}_t^*)^\top(\boldsymbol{\theta}_t-\boldsymbol{\theta}_{t+1})\rangle \\
&\quad + 2U_\delta^2\beta^2 + 2\mathbb{E}\|\boldsymbol{\omega}_t^* - \boldsymbol{\omega}_{t+1}^*\|^2 + 4\beta L_v H_v U_\delta^3 + 4U_\delta L_v\beta\epsilon_{\text{app}}
\end{aligned}
$$

It can be shown that

$$
\begin{aligned}
\mathbb{E}\|\boldsymbol{z}_{t+1}\|^2 &\overset{(1)}{\leq} (1-2\lambda\beta)\mathbb{E}\|\boldsymbol{z}_t\|^2 + 2\beta\mathbb{E}\Psi(O_t,\boldsymbol{\omega}_t,\boldsymbol{\theta}_t) + 2\beta U_v\mathbb{E}\|\boldsymbol{z}_t\|\|y_t\| + L_s\mathbb{E}\|\boldsymbol{z}_t\|\|\boldsymbol{\theta}_{t+1}-\boldsymbol{\theta}_t\|^2 \\
&\quad + 2\alpha\mathbb{E}\langle\boldsymbol{z}_t,-(\nabla\boldsymbol{\omega}_t^*)^\top\delta_t\nabla\log\pi_{\boldsymbol{\theta}_t}(a_t|s_t)\rangle + 2U_\delta^2\beta^2 \\
&\quad + 2L_*^2\mathbb{E}\|\boldsymbol{\theta}_t-\boldsymbol{\theta}_{t+1}\|^2 + 4\beta L_v H_v U_\delta^3 + 4U_\delta L_v\beta\epsilon_{\text{app}} \\
&\leq (1-2\lambda\beta)\mathbb{E}\|\boldsymbol{z}_t\|^2 + 2\beta\mathbb{E}\Psi(O_t,\boldsymbol{\omega}_t,\boldsymbol{\theta}_t) + 2\beta U_v\sqrt{\mathbb{E}y_t^2}\sqrt{\mathbb{E}\|\boldsymbol{z}_t\|^2} \\
&\quad + \frac{L_s}{2}\mathbb{E}\|\boldsymbol{z}_t\|^2\|\boldsymbol{\theta}_{t+1}-\boldsymbol{\theta}_t\|^2 + \frac{L_s}{2}\mathbb{E}\|\boldsymbol{\theta}_{t+1}-\boldsymbol{\theta}_t\|^2 + 2U_\delta^2\beta^2 + 2L_*^2 G^2\alpha^2 \\
&\quad + 2\alpha\mathbb{E}\langle\boldsymbol{z}_t,-(\nabla\boldsymbol{\omega}_t^*)^\top\delta_t\nabla\log\pi_{\boldsymbol{\theta}_t}(a_t|s_t)\rangle + 4\beta L_v H_v U_\delta^3 + 4U_\delta L_v\beta\epsilon_{\text{app}} \\
&\leq (1-2\lambda\beta)\mathbb{E}\|\boldsymbol{z}_t\|^2 + 2\beta\mathbb{E}\Psi(O_t,\boldsymbol{\omega}_t,\boldsymbol{\theta}_t) + 2\beta U_v\sqrt{\mathbb{E}y_t^2}\sqrt{\mathbb{E}\|\boldsymbol{z}_t\|^2} + \frac{L_s G^2}{2}\alpha^2\mathbb{E}\|\boldsymbol{z}_t\|^2 \\
&\quad + 2U_\delta^2\beta^2 + (2L_*^2 + \frac{L_s}{2})G^2\alpha^2 + 2\alpha\mathbb{E}\langle\boldsymbol{z}_t,-(\nabla\boldsymbol{\omega}_t^*)^\top\delta_t\nabla\log\pi_{\boldsymbol{\theta}_t}(a_t|s_t)\rangle \\
&\quad + 4\beta L_v H_v U_\delta^3 + 4U_\delta L_v\beta\epsilon_{\text{app}} \\
&\overset{(2)}{\leq} (1-\lambda\beta)\mathbb{E}\|\boldsymbol{z}_t\|^2 + 2\beta\mathbb{E}\Psi(O_t,\boldsymbol{\omega}_t,\boldsymbol{\theta}_t) + 2\beta U_v\sqrt{\mathbb{E}y_t^2}\sqrt{\mathbb{E}\|\boldsymbol{z}_t\|^2} \\
&\quad + 2U_\delta^2\beta^2 + (2L_*^2 + \frac{L_s}{2})G^2\alpha^2 + 2\alpha\mathbb{E}\langle\boldsymbol{z}_t,-(\nabla\boldsymbol{\omega}_t^*)^\top\delta_t\nabla\log\pi_{\boldsymbol{\theta}_t}(a_t|s_t)\rangle \\
&\quad + 4\beta L_v H_v U_\delta^3 + 4U_\delta L_v\beta\epsilon_{\text{app}}
\end{aligned}
$$
(23)

where (1) follows from the $L_s$-smoothness of $\boldsymbol{\omega}^*$ in Assumption 4.3; (2) uses $\frac{L_s G^2}{2}\alpha^2 \leq \lambda\beta$ for large $T$.

For term $\mathbb{E}\langle\boldsymbol{z}_t,-(\nabla\boldsymbol{\omega}_t^*)^\top\delta_t\nabla\log\pi_{\boldsymbol{\theta}_t}(a_t|s_t)\rangle$, we have

$$
\begin{aligned}
&\mathbb{E}\langle\boldsymbol{z}_t,-(\nabla\boldsymbol{\omega}_t^*)^\top\delta_t\nabla\log\pi_{\boldsymbol{\theta}_t}(a_t|s_t)\rangle \\
&= \mathbb{E}\langle\boldsymbol{z}_t,(\nabla\boldsymbol{\omega}_t^*)^\top(-\Delta h(O_t,\eta_t,\boldsymbol{\omega}_t,\boldsymbol{\theta}_t) - h(O_t,\boldsymbol{\theta}_t))\rangle \\
&= -\mathbb{E}\langle\boldsymbol{z}_t,(\nabla\boldsymbol{\omega}_t^*)^\top\Delta h(O_t,\eta_t,\boldsymbol{\omega}_t,\boldsymbol{\theta}_t)\rangle \\
&\quad + \mathbb{E}\langle\boldsymbol{z}_t,(\nabla\boldsymbol{\omega}_t^*)^\top(\mathbb{E}_{O_t'}[h(O_t',\boldsymbol{\theta}_t)] - h(O_t,\boldsymbol{\theta}_t) - \mathbb{E}_{O_t'}[h(O_t',\boldsymbol{\theta}_t)])\rangle \\
&= \mathbb{E}[\Xi(O_t,\boldsymbol{\omega}_t,\boldsymbol{\theta}_t)] - \mathbb{E}\langle\boldsymbol{z}_t,(\nabla\boldsymbol{\omega}_t^*)^\top\mathbb{E}_{O_t'}[h(O_t',\boldsymbol{\theta}_t)]\rangle \\
&\quad - \mathbb{E}\langle\boldsymbol{z}_t,(\nabla\boldsymbol{\omega}_t^*)^\top\Delta h(O_t,\eta_t,\boldsymbol{\omega}_t,\boldsymbol{\theta}_t)\rangle
\end{aligned}
$$

Note that from Cauchy-Schwartz inequality and $L_*$ is the Lipschitz constant of $\boldsymbol{\omega}^*$ in Assumption 4.3, we have

$$
-\mathbb{E}\langle\boldsymbol{z}_t,(\nabla\boldsymbol{\omega}_t^*)^\top\Delta h(O_t,\eta_t,\boldsymbol{\omega}_t,\boldsymbol{\theta}_t)\rangle \leq BL_*\sqrt{\mathbb{E}\|\boldsymbol{z}_t\|^2}\sqrt{2\mathbb{E}y_t^2 + 8L_v^2\mathbb{E}\|\boldsymbol{z}_t\|^2}. \tag{24}
$$

From the fact that

$$
\begin{aligned}
\mathbb{E}_{O_t'}[h(O_t',\boldsymbol{\theta}_t) - \Delta h'(O_t',\boldsymbol{\theta}_t)] &= \mathbb{E}_{O_t'}[(r(s_t,a_t) - J(\boldsymbol{\theta}_t) + V_{\boldsymbol{\theta}_t}(s_t') - V_{\boldsymbol{\theta}_t}(s_t))\nabla\log\pi_{\boldsymbol{\theta}_t}(a|s)] \\
&= \nabla J(\boldsymbol{\theta}_t),
\end{aligned}
$$

we obtain

$$\mathbb{E}\langle \boldsymbol{z}_t, (\nabla \boldsymbol{\omega}_t^*)^\top \mathbb{E}_{O_t'}[h(O_t', \theta_t)]\rangle = \mathbb{E}\langle \boldsymbol{z}_t, (\nabla \boldsymbol{\omega}_t^*)^\top \nabla J(\theta_t)\rangle + \mathbb{E}\langle \boldsymbol{z}_t, (\nabla \boldsymbol{\omega}_t^*)^\top \mathbb{E}_{O_t'}[\Delta h'(O_t', \theta_t)]\rangle.$$

It follows that

$$-\mathbb{E}\langle \boldsymbol{z}_t, (\nabla \boldsymbol{\omega}_t^*)^\top \nabla J(\theta_t)\rangle \leq L_* \sqrt{\mathbb{E}\|\boldsymbol{z}_t\|^2} \sqrt{\mathbb{E}\|\nabla J(\theta_t)\|^2}.$$

Furthermore, it holds that

$$\begin{aligned}
\mathbb{E}_{O'}\|\Delta h'(O, \boldsymbol{\theta})\|^2 &= \mathbb{E}_{O'}\|((\widehat{V}(\boldsymbol{\omega}^*(\boldsymbol{\theta}); s') - V_{\boldsymbol{\theta}}(s')) - (\widehat{V}(\boldsymbol{\omega}^*(\boldsymbol{\theta}); s) - V_{\boldsymbol{\theta}}(s)))\nabla \log \pi_{\boldsymbol{\theta}}(a|s)\|^2 \\
&\leq \mathbb{E}_{O'}[2B^2((\widehat{V}(\boldsymbol{\omega}^*(\boldsymbol{\theta}); s') - V_{\boldsymbol{\theta}}(s'))^2 + (\widehat{V}(\boldsymbol{\omega}^*(\boldsymbol{\theta}); s) - V_{\boldsymbol{\theta}}(s))^2)] \\
&= 4B^2\mathbb{E}_{O'}[(\widehat{V}(\boldsymbol{\omega}^*(\boldsymbol{\theta}); s) - V_{\boldsymbol{\theta}}(s))^2] \\
&= 4B^2\epsilon_{\text{app}}^2.
\end{aligned}$$

Therefore, we have

$$\begin{aligned}
-\langle \boldsymbol{z}_t, (\nabla \boldsymbol{\omega}_t^*)^\top \mathbb{E}_{O_t'}[h(O_t', \theta_t)]\rangle &\leq U_\delta L_* \sqrt{\|\mathbb{E}_{O'}[\Delta h'(O_t, \boldsymbol{\theta}_t)]\|^2} + L_* \sqrt{\mathbb{E}\|\boldsymbol{z}_t\|^2}\sqrt{\mathbb{E}\|\nabla J(\theta_t)\|^2} \\
&\leq U_\delta L_* \sqrt{\mathbb{E}_{O'}\|\Delta h'(O_t, \boldsymbol{\theta}_t)\|^2} + L_* \sqrt{\mathbb{E}\|\boldsymbol{z}_t\|^2}\sqrt{\mathbb{E}\|\nabla J(\theta_t)\|^2} \\
&\leq 2BU_\delta L_*\epsilon_{\text{app}} + L_* \sqrt{\mathbb{E}\|\boldsymbol{z}_t\|^2}\sqrt{\mathbb{E}\|\nabla J(\theta_t)\|^2}. \quad (25)
\end{aligned}$$

Substituting Eq. (24) and Eq. (25) into Eq. (24) yields

$$\begin{aligned}
\mathbb{E}\langle \boldsymbol{z}_t, -(\nabla \boldsymbol{\omega}_t^*)^\top \delta_t \nabla \log \pi_{\boldsymbol{\theta}_t}(a_t|s_t)\rangle &\leq \mathbb{E}\Xi(O_t, \boldsymbol{\omega}_t, \boldsymbol{\theta}_t) + 2BU_\delta L_*\epsilon_{\text{app}} \\
&\quad + BL_* \sqrt{\mathbb{E}\|\boldsymbol{z}_t\|^2}\sqrt{2\mathbb{E}y_t^2 + 8L_v^2\mathbb{E}\|\boldsymbol{z}_t\|^2} \quad (26) \\
&\quad + L_* \sqrt{\mathbb{E}\|\boldsymbol{z}_t\|^2}\sqrt{\mathbb{E}\|\nabla J(\theta_t)\|^2}.
\end{aligned}$$

Plugging Eq. (26) into Eq. (23), we have

$$\begin{aligned}
\mathbb{E}\|\boldsymbol{z}_{t+1}\|^2 &\leq (1 - \lambda\beta)\mathbb{E}\|\boldsymbol{z}_t\|^2 + 2\beta\mathbb{E}\Psi(O_t, \boldsymbol{\omega}_t, \boldsymbol{\theta}_t) + 2\alpha\mathbb{E}\Xi(O_t, \boldsymbol{\omega}_t, \boldsymbol{\theta}_t) \\
&\quad + 2\beta U_v \sqrt{\mathbb{E}y_t^2}\sqrt{\mathbb{E}\|\boldsymbol{z}_t\|^2} + 2BL_*\alpha \sqrt{\mathbb{E}\|\boldsymbol{z}_t\|^2}\sqrt{2\mathbb{E}y_t^2 + 8L_v^2\mathbb{E}\|\boldsymbol{z}_t\|^2} \\
&\quad + 2\alpha L_* \sqrt{\mathbb{E}\|\boldsymbol{z}_t\|^2}\sqrt{\mathbb{E}\|\nabla J(\theta_t)\|^2} + 2U_\delta^2\beta^2 + (2L_*^2 + \frac{L_s}{2})G^2\alpha^2 \quad (27) \\
&\quad + 4\beta L_v H_v U_\delta^3 + (2\alpha BU_\delta L_* + 4U_\delta L_v\beta)\epsilon_{\text{app}}.
\end{aligned}$$

Rearranging and summing from $\tau_T$ to $T - 1$ gives

$$\begin{aligned}
\lambda \sum_{\tau_T}^{T-1} \mathbb{E}\|\boldsymbol{z}_t\|^2 &\leq \underbrace{\sum_{t=\tau_T}^{T-1} \frac{1}{\beta}(\mathbb{E}\|\boldsymbol{z}_t\|^2 - \mathbb{E}\|\boldsymbol{z}_{t+1}\|^2)}_{I_1} + 2\underbrace{\sum_{t=\tau_T}^{T-1} \mathbb{E}\Psi(O_t, \boldsymbol{\omega}_t, \boldsymbol{\theta}_t)}_{I_2} + 2c\underbrace{\sum_{t=\tau_T}^{T-1} \mathbb{E}\Xi(O_t, \boldsymbol{\omega}_t, \boldsymbol{\theta}_t)}_{I_3} \\
&\quad + 2U_v \underbrace{\sum_{t=\tau_T}^{T-1} \sqrt{\mathbb{E}y_t^2}\sqrt{\mathbb{E}\|\boldsymbol{z}_t\|^2}}_{I_4} + 2cBL_* \underbrace{\sum_{t=\tau_T}^{T-1} \sqrt{\mathbb{E}\|\boldsymbol{z}_t\|^2}\sqrt{2\mathbb{E}y_t^2 + 8L_v^2\mathbb{E}\|\boldsymbol{z}_t\|^2}}_{I_5} \\
&\quad + 2cL_* \underbrace{\sum_{t=\tau_T}^{T-1} \sqrt{\mathbb{E}\|\boldsymbol{z}_t\|^2}\sqrt{\mathbb{E}\|\nabla J(\theta_t)\|^2}}_{I_6} \\
&\quad + \sum_{t=\tau_T}^{T-1} (2U_\delta^2\beta + c(2L_*^2 + \frac{L_s}{2})G^2\alpha + (2cBU_\delta L_* + 4U_\delta L_v)\epsilon_{\text{app}} + 4L_v H_v U_\delta^3).
\end{aligned}$$

In the sequel, we will tackle $I_1, I_2, I_3, I_4, I_5, I_6$ respectively.

For term $I_1$, we have

$$I_1 = \sum_{t=\tau_T}^{T-1} \frac{1}{\beta}(\mathbb{E}\|\boldsymbol{z}_t\|^2 - \mathbb{E}\|\boldsymbol{z}_{t+1}\|^2) \leq U_\delta^2 \sqrt{T}.$$

For term $I_2$, from Lemma F.3, choose $\tau = \tau_T$, we have

$$\mathbb{E}\Psi(O_t, \boldsymbol{\omega}_t, \boldsymbol{\theta}_t) \leq C_1\|\boldsymbol{\theta}_t - \boldsymbol{\theta}_{t-\tau}\| + C_2\|\boldsymbol{\omega}_t - \boldsymbol{\omega}_{t-\tau}\| + U_\delta^2 L_v L_\pi G \tau(\tau + 1)\alpha + 2U_\delta^2 L_v C \rho^{\tau-1}$$

$$\leq C_1 \sum_{k=t-\tau_T}^{t-1} G\alpha + C_2 \sum_{k=t-\tau_T}^{t-1} U_\delta \beta + U_\delta^2 L_v L_\pi G \tau_T(\tau_T + 1)\alpha + \frac{2U_\delta^2 L_v}{\sqrt{T}}$$

$$\leq (C_1 G \tau_T + U_\delta^2 L_v L_\pi G \tau_T(\tau_T + 1))\alpha + C_2 U_\delta \tau_T \beta + \frac{2U_\delta^2}{\sqrt{T}}.$$

Then we get

$$I_2 = 2 \sum_{T=\tau_T}^{T-1} \mathbb{E}\Psi(O_t, \boldsymbol{\omega}_t, \boldsymbol{\theta}_t) \leq 2 \sum_{T=\tau_T}^{T-1} ((C_1 G \tau_T + U_\delta^2 L_v L_\pi G \tau_T(\tau_T + 1))\alpha + C_2 U_\delta \tau_T \beta + \frac{2U_\delta^2}{\sqrt{T}}).$$

For term $I_3$, from Lemma F.4, choose $\tau = \tau_T$, we have

$$\mathbb{E}[\Xi(O_t, \boldsymbol{\omega}_t, \boldsymbol{\theta}_t)] \leq C_3\|\boldsymbol{\theta}_t - \boldsymbol{\theta}_{t-\tau_T}\| + 2U_\delta BL_*\|\boldsymbol{\omega}_t - \boldsymbol{\omega}_{t-\tau_T}\|$$

$$+ 2U_\delta^2 BL_* L_\pi G \tau_T(\tau_T + 1)\alpha + 4U_\delta^2 BL_* C \rho^{\tau_T - 1}$$

$$\leq C_3 \sum_{k=t-\tau_T}^{t-1} G\alpha + 2U_\delta BL_* \sum_{k=t-\tau_T}^{t-1} U_\delta \beta$$

$$+ 2U_\delta^2 BL_* L_\pi G \tau_T(\tau_T + 1)\alpha + 4U_\delta^2 BL_* C \rho^{\tau_T - 1}$$

$$\leq (C_3 G \tau_T + 2U_\delta^2 BL_* L_\pi G \tau_T(\tau_T + 1))\alpha + 2U_\delta^2 BL_* \tau_T \beta + \frac{4U_\delta^2 BL_*}{\sqrt{T}}.$$

Therefore, we have

$$I_3 = 2c \sum_{t=\tau_T}^{T-1} \mathbb{E}\Xi(O_t, \boldsymbol{\omega}_t, \boldsymbol{\theta}_t)$$

$$\leq 2c \sum_{t=\tau_T}^{T-1} ((C_3 G \tau_T + 2U_\delta^2 BL_* L_\pi G \tau_T(\tau_T + 1))\alpha + 2U_\delta^2 BL_* \tau_T \beta + \frac{4U_\delta^2 BL_*}{\sqrt{T}}).$$

For term $I_4$, $I_5$, and $I_6$, from Cauchy-Schwartz inequality, we have

$$I_4 \leq 2U_v \Big(\sum_{t=\tau_T}^{T-1} \mathbb{E}y_t^2\Big)^{\frac{1}{2}} \Big(\sum_{t=\tau_T}^{T-1} \mathbb{E}\|\boldsymbol{z}_t\|^2\Big)^{\frac{1}{2}},$$

$$I_5 \leq 2cBL_* \Big(\sum_{t=\tau_T}^{T-1} \mathbb{E}\|\boldsymbol{z}_t\|^2\Big)^{\frac{1}{2}} \Big(2\sum_{t=\tau_T}^{T-1} \mathbb{E}y_t^2 + 8L_v^2 \sum_{t=\tau_T}^{T-1} \mathbb{E}\|\boldsymbol{z}_t\|^2\Big)^{\frac{1}{2}},$$

$$I_6 \leq 2cL_* \Big(\sum_{t=\tau_T}^{T-1} \mathbb{E}\|\boldsymbol{z}_t\|^2\Big)^{\frac{1}{2}} \Big(\sum_{t=\tau_T}^{T-1} \mathbb{E}\|\nabla J(\boldsymbol{\theta}_t)\|\Big)^{\frac{1}{2}}.$$

Overall, we get

$$\lambda \sum_{t=\tau_T}^{T-1} \mathbb{E}\|\boldsymbol{z}_t\|^2 \le 2U_v \Big(\sum_{t=\tau_T}^{T-1} \mathbb{E}y_t^2\Big)^{\frac{1}{2}} \Big(\sum_{t=\tau_T}^{T-1} \mathbb{E}\|\boldsymbol{z}_t\|^2\Big)^{\frac{1}{2}}$$

$$+ 2cBL_* \Big(\sum_{t=\tau_t}^{T-1} \mathbb{E}\|\boldsymbol{z}_t\|^2\Big)^{\frac{1}{2}} \Big(2\sum_{t=\tau_T}^{T-1} \mathbb{E}y_t^2 + 8L_v^2 \sum_{t=\tau_T}^{T-1} \mathbb{E}\|\boldsymbol{z}_t\|^2\Big)^{\frac{1}{2}}$$

$$+ 2cL_* \Big(\sum_{t=\tau_T}^{T-1} \mathbb{E}\|\boldsymbol{z}_t\|^2\Big)^{\frac{1}{2}} \Big(\sum_{t=\tau_T}^{T-1} \mathbb{E}\|\nabla J(\boldsymbol{\theta}_t)\|\Big)^{\frac{1}{2}}$$

$$+ U_\delta^2 \sqrt{T} + 2 \sum_{T=\tau_T}^{T-1} \Big((C_1 G\tau_T + U_\delta^2 L_v L_\pi G\tau_T(\tau_T+1))\alpha + C_2 U_\delta \tau_T \beta + \frac{2U_\delta^2}{\sqrt{T}}\Big)$$

$$+ 2c \sum_{t=\tau_T}^{T-1} \Big((C_3 G\tau_T + 2U_\delta^2 BL_* L_\pi G\tau_T(\tau_T+1))\alpha + 2U_\delta^2 BL_* \tau_T \beta + \frac{4U_\delta^2 BL_*}{\sqrt{T}}\Big)$$

$$+ \sum_{t=\tau_T}^{T-1} \Big(2U_\delta^2 \beta + c(2L_*^2 + \frac{L_s}{2})G^2\alpha + (2cBU_\delta L_* + 4U_\delta L_v)\epsilon_{\mathrm{app}} + 4L_v H_v U_\delta^3\Big).$$

Therefore, we have

$$Z_T \overset{(1)}{\le} \frac{2U_v}{\lambda}\Big(\frac{1}{T-\tau_T}\sum_{t=\tau_T}^{T-1} \mathbb{E}y_t^2\Big)^{\frac{1}{2}} \Big(\frac{1}{T-\tau_T}\sum_{t=\tau_T}^{T-1} \mathbb{E}\|\boldsymbol{z}_t\|^2\Big)^{\frac{1}{2}}$$

$$+ \frac{2cBL_*}{\lambda}\Big(\frac{1}{T-\tau_T}\sum_{t=\tau_t}^{T-1} \mathbb{E}\|\boldsymbol{z}_t\|^2\Big)^{\frac{1}{2}} \Big(2\frac{1}{T-\tau_T}\sum_{t=\tau_T}^{T-1} \mathbb{E}y_t^2 + 8L_v^2 \frac{1}{T-\tau_T}\sum_{t=\tau_T}^{T-1} \mathbb{E}\|\boldsymbol{z}_t\|^2\Big)^{\frac{1}{2}}$$

$$+ \frac{2cL_*}{\lambda}\Big(\frac{1}{T-\tau_T}\sum_{t=\tau_T}^{T-1} \mathbb{E}\|\boldsymbol{z}_t\|^2\Big)^{\frac{1}{2}} \Big(\frac{1}{T-\tau_T}\sum_{t=\tau_T}^{T-1} \mathbb{E}\|\nabla J(\boldsymbol{\theta}_t)\|\Big)^{\frac{1}{2}}$$

$$+ \frac{1}{\lambda}\Big(\frac{2U_\delta^2}{\sqrt{T}} + 2((C_1 G\tau_T + U_\delta^2 L_v L_\pi G\tau_T(\tau_T+1))\alpha + C_2 U_\delta \tau_T \beta + \frac{2U_\delta^2}{\sqrt{T}})$$

$$+ 2c((C_3 G\tau_T + 2U_\delta^2 BL_* L_\pi G\tau_T(\tau_T+1))\alpha + 2U_\delta^2 BL_* \tau_T \beta + \frac{4U_\delta^2 BL_*}{\sqrt{T}})$$

$$+ 2U_\delta^2 \beta + c(2L_*^2 + \frac{L_s}{2})G^2\alpha + (2cBU_\delta L_* + 4U_\delta L_v)\epsilon_{\mathrm{app}} + 4L_v H_v U_\delta^3)$$

$$= \mathcal{O}\Big(\frac{\log^2 T}{\sqrt{T}}\Big) + \widetilde{\mathcal{O}}\Big(\frac{1}{\sqrt{m}}\Big) + \mathcal{O}(\epsilon_{\mathrm{app}}) + \frac{2U_v}{\lambda}\Big(\frac{1}{T-\tau_T}\sum_{t=\tau_T}^{T-1} \mathbb{E}y_t^2\Big)^{\frac{1}{2}} \Big(\frac{1}{T-\tau_T}\sum_{t=\tau_T}^{T-1} \mathbb{E}\|\boldsymbol{z}_t\|^2\Big)^{\frac{1}{2}}$$

$$+ \frac{2cBL_*}{\lambda}\Big(\frac{1}{T-\tau_T}\sum_{t=\tau_t}^{T-1} \mathbb{E}\|\boldsymbol{z}_t\|^2\Big)^{\frac{1}{2}} \Big(2\frac{1}{T-\tau_T}\sum_{t=\tau_T}^{T-1} \mathbb{E}y_t^2 + 8L_v^2 \frac{1}{T-\tau_T}\sum_{t=\tau_T}^{T-1} \mathbb{E}\|\boldsymbol{z}_t\|^2\Big)^{\frac{1}{2}}$$

$$+ \frac{2cL_*}{\lambda}\Big(\frac{1}{T-\tau_T}\sum_{t=\tau_T}^{T-1} \mathbb{E}\|\boldsymbol{z}_t\|^2\Big)^{\frac{1}{2}} \Big(\frac{1}{T-\tau_T}\sum_{t=\tau_T}^{T-1} \mathbb{E}\|\nabla J(\boldsymbol{\theta}_t)\|\Big)^{\frac{1}{2}},$$

where (1) follows from $\tau_T = \mathcal{O}(\log T)$ so that $T - \tau_T \ge \frac{1}{2}T$ for large $T$ and the term $\widetilde{\mathcal{O}}\big(\frac{1}{\sqrt{m}}\big)$ comes from the fact $H_v = \widetilde{\mathcal{O}}\big(\frac{1}{\sqrt{m}}\big)$ as shown in Lemma C.5. Therefore, we have

$$Z_T \le \mathcal{O}\Big(\frac{\log^2 T}{\sqrt{T}}\Big) + \widetilde{\mathcal{O}}\Big(\frac{1}{\sqrt{m}}\Big) + \mathcal{O}(\epsilon_{\mathrm{app}}) + \frac{2U_v}{\lambda}\sqrt{Y_T Z_T}$$

$$+ \frac{2cBL_*}{\lambda}\sqrt{Z_T(2Y_T + 8L_v^2 Z_T)} + \frac{2cL_*}{\lambda}\sqrt{Z_T G_T},$$

which completes the proof. $\square$

### F.3 STEP 3: POLICY GRADIENT NORM ANALYSIS

In this subsection, we will establish an implicit upper bound for policy gradient norm.

**Lemma F.6.** *For any $t \geq \tau > 0$, it holds that*

$$\mathbb{E}[\Theta(O_t, \boldsymbol{\theta}_t)] \leq C_4 \tau(\tau + 1)G\alpha + C_5 C \rho^{\tau - 1},$$

*where $C_4 = \max\{2U_\delta BL_{J'} + 3L_J(U_\delta L_l + 2BL_v L_* + 4BU_\delta L_J), 2U_\delta BL_J L_\pi\}$, $C_5 = 4U_\delta BL_J$.*

**Theorem F.7.** *We have*

$$G_T \leq \mathcal{O}(\frac{\log^2 T}{\sqrt{T}}) + \mathcal{O}(\epsilon_{\text{app}}) + B\sqrt{G_T(2Y_T + 8L_v^2 Z_T)}. \tag{28}$$

*Proof.* From the update rule of actor in Line 9 of Algorithm 1 and Eq. (14), we have

$$J(\boldsymbol{\theta}_{t+1}) \geq J(\boldsymbol{\theta}_t) + \langle \nabla J(\boldsymbol{\theta}_t), \boldsymbol{\theta}_{t+1} - \boldsymbol{\theta}_t \rangle - \frac{L_{J'}}{2}\|\boldsymbol{\theta}_t - \boldsymbol{\theta}_{t+1}\|^2$$

$$= J(\boldsymbol{\theta}_t) + \alpha\langle \nabla J(\boldsymbol{\theta}_t), \delta_t \nabla \log \pi_{\boldsymbol{\theta}_t}(a_t|s_t) \rangle - \frac{L_{J'}}{2}\alpha^2\|\delta_t \nabla \log \pi_{\boldsymbol{\theta}_t}(a_t|s_t)\|^2$$

$$= J(\boldsymbol{\theta}_t) + \alpha\langle \nabla J(\boldsymbol{\theta}_t), \Delta h(O_t, \eta_t, \boldsymbol{\omega}_t, \boldsymbol{\theta}_t) \rangle + \alpha\langle \nabla J(\boldsymbol{\theta}_t), h(O_t, \boldsymbol{\theta}_t) \rangle$$
$$\quad - \frac{L_{J'}}{2}\alpha^2\|\delta_t \nabla \log \pi_{\boldsymbol{\theta}_t}(a_t|s_t)\|^2$$

$$= J(\boldsymbol{\theta}_t) + \alpha\langle \nabla J(\boldsymbol{\theta}_t), \Delta h(O_t, \eta_t, \boldsymbol{\omega}_t, \boldsymbol{\theta}_t) \rangle - \alpha\Theta(O_t, \boldsymbol{\theta}_t)$$
$$\quad + \alpha\langle \nabla J(\boldsymbol{\theta}_t), \mathbb{E}_{O_t'}[h(O_t', \boldsymbol{\theta}_t)] \rangle - \frac{L_{J'}}{2}\alpha^2\|\delta_t \nabla \log \pi_{\boldsymbol{\theta}_t}(a_t|s_t)\|^2$$

$$= J(\boldsymbol{\theta}_t) + \alpha\langle \nabla J(\boldsymbol{\theta}_t), \Delta h(O_t, \eta_t, \boldsymbol{\omega}_t, \boldsymbol{\theta}_t) \rangle - \alpha\Theta(O_t, \boldsymbol{\theta}_t) + \alpha\|\nabla J(\boldsymbol{\theta}_t)\|^2$$
$$\quad + \alpha\langle \nabla J(\boldsymbol{\theta}_t), \mathbb{E}_{O_t'}[\Delta h'(O_t', \boldsymbol{\theta}_t)] \rangle - \frac{L_{J'}}{2}\alpha^2\|\delta_t \nabla \log \pi_{\boldsymbol{\theta}_t}(a_t|s_t)\|^2,$$

where the last equality is due to the fact

$$\mathbb{E}_{O'}[h(O', \boldsymbol{\theta}) - \Delta h'(O', \boldsymbol{\theta})] = \mathbb{E}_{O'}[(r(s, a) - J(\boldsymbol{\theta}) + V_{\boldsymbol{\theta}}(s') - V_{\boldsymbol{\theta}}(s))\nabla \log \pi_{\boldsymbol{\theta}}(a|s)] = \nabla J(\boldsymbol{\theta}).$$

Rearranging the above inequality and taking expectation, we have

$$\mathbb{E}\|\nabla J(\boldsymbol{\theta}_t)\|^2 \leq \frac{1}{\alpha}(\mathbb{E}[J(\boldsymbol{\theta}_{t+1}) - J(\boldsymbol{\theta}_t)]) - \mathbb{E}\langle \nabla J(\boldsymbol{\theta}_t), \Delta h(O_t, \eta_t, \boldsymbol{\omega}_t, \boldsymbol{\theta}_t) \rangle + \mathbb{E}[\Theta(O_t, \boldsymbol{\theta}_t)]$$
$$\quad - \mathbb{E}\langle \nabla J(\boldsymbol{\theta}_t), \mathbb{E}_{O_t'}[\Delta h'(O_t', \boldsymbol{\theta}_t)] \rangle + \frac{L_{J'}}{2}\alpha\mathbb{E}\|\delta_t \nabla \log \pi_{\boldsymbol{\theta}_t}(a_t|s_t)\|^2.$$

Note that from Cauchy-Schwartz inequality, we have

$$-\mathbb{E}\langle \nabla J(\boldsymbol{\theta}_t), \Delta h(O_t, \eta_t, \boldsymbol{\omega}_t, \boldsymbol{\theta}_t) \rangle \leq B\sqrt{\mathbb{E}\|\nabla J(\boldsymbol{\theta}_t)\|^2}\sqrt{2\mathbb{E}y_t^2 + 8L_v^2\mathbb{E}\|\boldsymbol{z}_t\|^2}.$$

From Lemma F.6 and choosing $\tau = \tau_T$, we have

$$\mathbb{E}[\Theta(O_t, \boldsymbol{\theta}_t)] \leq C_4 \tau_T(\tau_T + 1)G\alpha + C_5 C \rho^{\tau - 1}$$

$$\leq C_4 \tau_T(\tau_T + 1)G\alpha + C_5 \frac{1}{\sqrt{T}}.$$

It has been shown that

$$\mathbb{E}_{O'}\|\Delta h'(O, \boldsymbol{\theta})\|^2 \leq 4B^2 \epsilon_{\text{app}}^2.$$

Therefore, we have

$$-\langle \nabla J(\boldsymbol{\theta}_t), \mathbb{E}_{O_t'}[\Delta h'(O_t', \boldsymbol{\theta}_t)] \rangle \leq L_J\sqrt{\|\mathbb{E}_{O'}[\Delta h'(O_t', \boldsymbol{\theta}_t)]\|^2}$$
$$\leq L_J\sqrt{\mathbb{E}_{O'}\|\Delta h'(O_t', \boldsymbol{\theta}_t)\|^2}$$
$$\leq 2BL_J\epsilon_{\text{app}},$$

where we use $\|\nabla J(\boldsymbol{\theta})\| \le L_J$ which comes from Lemma C.3. Plugging the three terms yields

$$\mathbb{E}\|\nabla J(\boldsymbol{\theta}_t)\|^2 \le \frac{1}{\alpha}(\mathbb{E}[J(\boldsymbol{\theta}_{t+1})] - \mathbb{E}[J(\boldsymbol{\theta}_t)]) + B\sqrt{\mathbb{E}\|\nabla J(\boldsymbol{\theta}_t)\|^2}\sqrt{2\mathbb{E}y_t^2 + 8L_v^2\mathbb{E}\|\boldsymbol{z}_t\|^2}$$

$$+ 2BL_J\epsilon_{\text{app}} + C_4\tau_T(\tau_T + 1)G\alpha + C_5\frac{1}{\sqrt{T}} + \frac{L_{J'}}{2}G^2\alpha.$$

Summing over $t$ from $\tau_T$ to $T - 1$ gives

$$\sum_{t=\tau_T}^{T-1} \mathbb{E}\|\nabla J(\boldsymbol{\theta}_t)\|^2 \le \underbrace{\sum_{t=\tau_T}^{T-1} \frac{1}{\alpha}(\mathbb{E}[J(\boldsymbol{\theta}_{t+1}) - \mathbb{E}[J(\boldsymbol{\theta}_t)])}_{I_1} + B\sum_{t=\tau_T}^{T-1}\sqrt{\mathbb{E}\|\nabla J(\boldsymbol{\theta}_t)\|^2}\sqrt{2\mathbb{E}y_t^2 + 8L_v^2\mathbb{E}\|\boldsymbol{z}_t\|^2}$$

$$+ (C_4\tau_T(\tau_T + 1)G + C_5 + \frac{L_{J'}}{2}G^2)\frac{T - \tau_T}{\sqrt{T}} + 2BL_J\epsilon_{\text{app}}(T - \tau_T).$$

For term $I_1$, we have

$$I_1 = \sum_{t=\tau_T}^{T-1} \frac{1}{\alpha}(\mathbb{E}[J(\boldsymbol{\theta}_{t+1})] - \mathbb{E}[J(\boldsymbol{\theta}_t)])$$

$$\le \frac{2U_r}{c}\sqrt{T}.$$

Overall, we have

$$\sum_{t=\tau_T}^{T-1} \mathbb{E}\|\nabla J(\boldsymbol{\theta}_t)\|^2 \le \frac{2U_r}{c}\sqrt{T} + (C_4\tau_T(\tau_T + 1)G + C_5 + \frac{L_{J'}}{2}G^2)\frac{T - \tau_T}{\sqrt{T}} + 2BL_J\epsilon_{\text{app}}(T - \tau_T)$$

$$+ B\sum_{t=\tau_T}^{T-1}\sqrt{\mathbb{E}\|\nabla J(\boldsymbol{\theta}_t)\|^2}\sqrt{2\mathbb{E}y_t^2 + 8L_v^2\mathbb{E}\|\boldsymbol{z}_t\|^2}$$

$$\le \frac{2U_r}{c}\sqrt{T} + (C_4\tau_T(\tau_T + 1)G + C_5 + \frac{L_{J'}}{2}G^2)\frac{T - \tau_T}{\sqrt{T}} + 2BL_J\epsilon_{\text{app}}(T - \tau_T)$$

$$+ B(\sum_{t=\tau_T}^{T-1} \mathbb{E}\|\nabla J(\boldsymbol{\theta}_t)\|^2)^{\frac{1}{2}}(2\sum_{t=\tau_T}^{T-1} \mathbb{E}y_t^2 + 8L_v^2\sum_{t=\tau_T}^{T-1} \mathbb{E}\|\boldsymbol{z}_t\|^2)^{\frac{1}{2}}.$$

Therefore, we get

$$G_T \le (\frac{4U_r}{c} + C_4\tau_T(\tau_T + 1)G + C_5 + L_{J'}G^2)\frac{1}{\sqrt{T}} + 2BL_J\epsilon_{\text{app}} + B\sqrt{G_T(2Y_T + 8L_v^2Z_T)}$$

$$= \mathcal{O}(\frac{\log^2 T}{\sqrt{T}}) + \mathcal{O}(\epsilon_{\text{app}}) + B\sqrt{G_T(2Y_T + 8L_v^2Z_T)},$$

which concludes the proof. $\qquad\square$

### F.4 STEP 4: INTERCONNECTED ITERATION SYSTEM ANALYSIS

In this subsection, we perform an interconnected iteration system analysis to prove Theorem 4.9.

**Proof of Theorem 4.9.**

*Proof.* Combining Eq. (19), Eq. (20), and Eq. (28), we have

$$Y_T \le \mathcal{O}(\frac{\log^2 T}{\sqrt{T}}) + cG\sqrt{Y_TG_T},$$

$$Z_T \le \mathcal{O}(\frac{\log^2 T}{\sqrt{T}}) + \widetilde{\mathcal{O}}(\frac{1}{\sqrt{m}}) + \mathcal{O}(\epsilon_{\text{app}}) + \frac{2U_v}{\lambda}\sqrt{Y_TZ_T}$$

$$+ \frac{2cBL_*}{\lambda}\sqrt{Z_T(2Y_T + 8L_v^2Z_T)} + \frac{2cL_*}{\lambda}\sqrt{Z_TG_T}$$

$$G_T \le \mathcal{O}(\frac{\log^2 T}{\sqrt{T}}) + \mathcal{O}(\epsilon_{\text{app}}) + B\sqrt{G_T(2Y_T + 8L_v^2Z_T)}.$$

Denote

$$l_1 := cG, l_2 := \frac{2U_v}{\lambda}, l_3 := \frac{2cBL_*}{\lambda}, l_4 := 8L_v^2, l_5 := \frac{2cL_*}{\lambda}, l_6 := B. \tag{29}$$

Then we have

$$Y_T \leq \mathcal{O}(\frac{\log^2 T}{\sqrt{T}}) + l_1\sqrt{Y_T G_T},$$

$$Z_T \leq \mathcal{O}(\frac{\log^2 T}{\sqrt{T}}) + \mathcal{O}(\epsilon_{\mathrm{app}}) + \widetilde{\mathcal{O}}(\frac{1}{\sqrt{m}}) + l_2\sqrt{Y_T Z_T} + l_3\sqrt{Z_T(2Y_T + l_4 Z_T)} + l_5\sqrt{Z_T G_T},$$

$$G_T \leq \mathcal{O}(\frac{\log^2 T}{\sqrt{T}}) + \mathcal{O}(\epsilon_{\mathrm{app}}) + l_6\sqrt{G_T(2Y_T + l_4 Z_T)}.$$

For $G_T$, we get

$$G_T \leq \mathcal{O}(\frac{\log^2 T}{\sqrt{T}}) + \mathcal{O}(\epsilon_{\mathrm{app}}) + \frac{1}{2}G_T + l_6^2(Y_T + \frac{1}{2}l_4 Z_T),$$

$$G_T \leq \mathcal{O}(\frac{\log^2 T}{\sqrt{T}}) + \mathcal{O}(\epsilon_{\mathrm{app}}) + l_6^2(2Y_T + l_4 Z_T). \tag{30}$$

For $Z_T$, we have

$$Z_T \leq \mathcal{O}(\frac{\log^2 T}{\sqrt{T}}) + \mathcal{O}(\epsilon_{\mathrm{app}}) + \widetilde{\mathcal{O}}(\frac{1}{\sqrt{m}}) + \frac{1}{4}Z_T + l_2^2 Y_T + (1 + \frac{1}{2}l_4)l_3 Z_T + l_3 Y_T + \frac{1}{4}Z_T + l_5^2 G_T.$$

If it satisfies $(1 + \frac{1}{2}l_4)l_3 \leq \frac{1}{4}$, we further have

$$Z_T \leq \mathcal{O}(\frac{\log^2 T}{\sqrt{T}}) + \mathcal{O}(\epsilon_{\mathrm{app}}) + \widetilde{\mathcal{O}}(\frac{1}{\sqrt{m}}) + (2l_2^2 + 2l_3)Y_T + 2l_5^2 G_T. \tag{31}$$

Plugging Eq. (30) into Eq. (31), it holds that

$$Z_T \leq \mathcal{O}(\frac{\log^2 T}{\sqrt{T}}) + \mathcal{O}(\epsilon_{\mathrm{app}}) + \widetilde{\mathcal{O}}(\frac{1}{\sqrt{m}}) + (2l_2^2 + 2l_3 + 4l_5^2 l_6^2)Y_T + 2l_4 l_5^2 l_6^2 Z_T.$$

If it satisfies $2l_4 l_5^2 l_6^2 \leq \frac{1}{2}$, we have

$$Z_T \leq \mathcal{O}(\frac{\log^2 T}{\sqrt{T}}) + \mathcal{O}(\epsilon_{\mathrm{app}}) + \widetilde{\mathcal{O}}(\frac{1}{\sqrt{m}}) + 4(l_2^2 + l_3 + 2l_5^2 l_6^2)Y_T. \tag{32}$$

For $Y_T$, we get

$$Y_T \leq \mathcal{O}(\frac{\log^2 T}{\sqrt{T}}) + \frac{l_1}{2}(Y_T + G_T). \tag{33}$$

Plugging Eq. (30) and Eq. (32) into Eq. (33) gives

$$Y_T \leq \mathcal{O}(\frac{\log^2 T}{\sqrt{T}}) + \mathcal{O}(\epsilon_{\mathrm{app}}) + \frac{l_1}{2}(Y_T + 2l_6^2 Y_T + l_4 l_6^2 Z_T)$$

$$\leq \mathcal{O}(\frac{\log^2 T}{\sqrt{T}}) + \mathcal{O}(\epsilon_{\mathrm{app}}) + \widetilde{\mathcal{O}}(\frac{1}{\sqrt{m}}) + \frac{l_1}{2}(Y_T + 2l_6^2 Y_T + 4l_4 l_6^2(l_2^2 + l_3 + 2l_5^2 l_6^2))Y_T$$

$$= \mathcal{O}(\frac{\log^2 T}{\sqrt{T}}) + \mathcal{O}(\epsilon_{\mathrm{app}}) + \widetilde{\mathcal{O}}(\frac{1}{\sqrt{m}}) + \frac{l_1}{2}(1 + 2l_6^2 + 4l_4 l_6^2(l_2^2 + l_3 + 2l_5^2 l_6^2))Y_T.$$

Therefore, if $l_1(1 + 2l_6^2 + 4l_4 l_6^2(l_2^2 + l_3 + 2l_5^2 l_6^2)) \leq 1$, we have

$$Y_T \leq \mathcal{O}(\frac{\log^2 T}{\sqrt{T}}) + \mathcal{O}(\epsilon_{\mathrm{app}}) + \widetilde{\mathcal{O}}(\frac{1}{\sqrt{m}}).$$

Overall, we require

$$(1 + \frac{1}{2}l_4)l_3 \leq \frac{1}{4}, \ 2l_4 l_5^2 l_6^2 \leq \frac{1}{2}, \ l_1(1 + 2l_6^2 + 4l_4 l_6^2(l_2^2 + l_3 + 2l_5^2 l_6^2)) \leq 1.$$

According to the definition of $l_1, l_2, l_3, l_4, l_5, l_6$, we have

$$(1 + 4L_v^2)\frac{2cBL_*}{\lambda} \leq \frac{1}{4},$$

$$\frac{64L_v^2 c^2 L_*^2 B^2}{\lambda^2} \leq \frac{1}{2},$$

$$cG(1 + 2B^2 + 32L_v^2 B^2(\frac{4U_v^2}{\lambda^2} + \frac{2cBL_*}{\lambda} + \frac{8c^2 B^2 L_*^2}{\lambda^2})) \leq 1.$$

Thus we choose

$$c \leq \min\{\frac{\lambda}{16c(1 + 4L_v^2)BL_*}, \frac{\lambda^2}{G((1 + 2B^2 + 32L_v^2 B^2)\lambda^2 + 128L_v^2 U_v^2 B^2)}\}, \tag{34}$$

which satisfies the above two inequalities. Therefore, we have

$$Y_T = \mathcal{O}(\frac{\log^2 T}{\sqrt{T}}) + \mathcal{O}(\epsilon_{\mathrm{app}}) + \widetilde{\mathcal{O}}(\frac{1}{\sqrt{m}}),$$

and consequently,

$$Z_T = \mathcal{O}(\frac{\log^2 T}{\sqrt{T}}) + \mathcal{O}(\epsilon_{\mathrm{app}}) + \widetilde{\mathcal{O}}(\frac{1}{\sqrt{m}}),$$

$$G_T = \mathcal{O}(\frac{\log^2 T}{\sqrt{T}}) + \mathcal{O}(\epsilon_{\mathrm{app}}) + \widetilde{\mathcal{O}}(\frac{1}{\sqrt{m}}).$$

Thus we conclude our proof. $\qquad\square$

## G  PROOF OF PRELIMINARY LEMMAS

The following preliminary lemmas have been established in prior research (Zou et al., 2019; Zhang et al., 2020a; Wu et al., 2020b; Liu et al., 2020). In this paper, we make modifications to accommodate continuous action spaces.

**Proof of Lemma C.1**.

*Proof.* For any $\theta_1$ and $\theta_2$, define the transition kernels respectively as follows:

$$P_i(s, ds') = \int_{\mathcal{A}} \mathcal{P}(ds'|s, a)\pi_{\theta_i}(a|s), \quad i = 1, 2$$

Following from Theorem 3.1 in Mitrophanov (2005), we obtain

$$d_{TV}(\mu_{\theta_1}, \mu_{\theta_2}) \leq (\lceil \log_\rho C^{-1}\rceil + \frac{1}{1 - \rho})\|P_1 - P_2\|_{\mathrm{op}},$$

where $\|\cdot\|_{\mathrm{op}}$ is the operator norm defined in Mitrophanov (2005): $\|A\| := \sup_{\|q\|_{\mathrm{TV}}=1}\|qA\|_{\mathrm{TV}}$, and $\|\cdot\|_{\mathrm{TV}}$ denotes the total-variation norm. Then we have

$$\begin{aligned}
\|P_1 - P_2\|_{\mathrm{op}} &= \sup_{\|q\|_{\mathrm{TV}}=1} \|\int_{\mathcal{S}} q(ds)(P_1 - P_2)(s, \cdot)\|_{\mathrm{TV}} \\
&= \sup_{\|q\|_{\mathrm{TV}}=1} \int_{\mathcal{S}} |\int_{\mathcal{S}} q(ds)(P_1 - P_2)(s, ds')| \\
&\leq \sup_{\|q\|_{\mathrm{TV}}=1} \int_{\mathcal{S}} \int_{\mathcal{S}} q(ds)|(P_1 - P_2)(s, ds')| \\
&= \sup_{\|q\|_{\mathrm{TV}}=1} \int_{\mathcal{S}} \int_{\mathcal{S}} q(ds)|\int_{\mathcal{A}} \mathcal{P}(ds'|s, a)(\pi_{\theta_1}(da|s) - \pi_{\theta_2}(da|s))| \\
&= \sup_{\|q\|_{\mathrm{TV}}=1} \int_{\mathcal{S}} \int_{\mathcal{S}} q(ds)\int_{\mathcal{A}} \mathcal{P}(ds'|s, a)|(\pi_{\theta_1}(da|s) - \pi_{\theta_2}(da|s))| \\
&= \sup_{\|q\|_{\mathrm{TV}}=1} \int_{\mathcal{S}} q(ds)\int_{\mathcal{A}} |(\pi_{\theta_1}(da|s) - \pi_{\theta_2}(da|s))| \\
&\leq L_\pi\|\theta_1 - \theta_2\|.
\end{aligned}$$

The first equation results from the definition of the operation norm, the second equation results from the definition of total variation. Therefore, we have

$$d_{TV}(\mu_{\boldsymbol{\theta}_1}, \mu_{\boldsymbol{\theta}_2}) \leq L_\pi(\lceil \log_\rho C^{-1} \rceil + \frac{1}{1-\rho})\|\boldsymbol{\theta}_1 - \boldsymbol{\theta}_2\|.$$

For the second inequality, we have

$$\begin{aligned}
d_{TV}(\mu_{\boldsymbol{\theta}_1} \otimes \pi_{\boldsymbol{\theta}_1}, \mu_{\boldsymbol{\theta}_2} \otimes \pi_{\boldsymbol{\theta}_2}) &= \int_{\mathcal{S}} \int_{\mathcal{A}} |\mu_{\theta_1}(ds)\pi_{\theta_1}(a|s) - \mu_{\theta_2}(ds)\pi_{\theta_2}(a|s)| \\
&\leq \int_{\mathcal{S}} \int_{\mathcal{A}} |\mu_{\theta_1}(ds)(\pi_{\theta_1}(a|s) - \pi_{\theta_2}(a|s))| \\
&\quad + \int_{\mathcal{S}} \int_{\mathcal{A}} |(\mu_{\theta_1}(ds) - \mu_{\theta_2}(ds))\pi_{\theta_2}(a|s))| \\
&= d_{TV}(\pi_{\theta_1}, \pi_{\theta_2}) + d_{TV}(\mu_{\theta_1}, \mu_{\theta_2}) \\
&\leq L_\pi\|\theta_1 - \theta_2\| + C(\lceil \log_\rho C^{-1} \rceil + \frac{1}{1-\rho})\|\boldsymbol{\theta}_1 - \boldsymbol{\theta}_2\| \\
&= L_\pi(1 + \lceil \log_\rho C^{-1} \rceil + \frac{1}{1-\rho})\|\boldsymbol{\theta}_1 - \boldsymbol{\theta}_2\|.
\end{aligned}$$

For the third inequality, we have

$$\begin{aligned}
&d_{TV}(\mu_{\boldsymbol{\theta}_1} \otimes \pi_{\boldsymbol{\theta}_1} \otimes \mathcal{P}, \mu_{\boldsymbol{\theta}_2} \otimes \pi_{\boldsymbol{\theta}_2} \otimes \mathcal{P}) \\
&= \frac{1}{2} \int_{\mathcal{S}} \int_{\mathcal{A}} \int_{\mathcal{S}} |\mu_{\theta_1}(ds)\pi_{\theta_1}(a|s)\mathcal{P}(ds'|s,a) - \mu_{\theta_2}(ds)\pi_{\theta_2}(a|s)\mathcal{P}(ds'|s,a)| \\
&= \frac{1}{2} \int_{\mathcal{S}} \int_{\mathcal{A}} |\mu_{\theta_1}(ds)\pi_{\theta_1}(a|s) - \mu_{\theta_2}(ds)\pi_{\theta_2}(a|s)| \\
&= d_{TV}(\mu_{\boldsymbol{\theta}_1} \otimes \pi_{\boldsymbol{\theta}_1}, \mu_{\boldsymbol{\theta}_2} \otimes \pi_{\boldsymbol{\theta}_2}),
\end{aligned}$$

which concludes the proof. □

**Proof of Lemma C.2.**

*Proof.* From the fact that

$$\mathbb{P}(s_{t+1} \in \cdot) = \int_{\mathcal{S}} \int_{\mathcal{A}} \mathbb{P}(s_t = ds, a_t = da, s_{t+1} \in \cdot),$$

we have

$$\begin{aligned}
&2d_{TV}(\mathbb{P}(s_{t+1} \in \cdot), \mathbb{P}(\tilde{s}_{t+1} \in \cdot)) \\
&= \int_{\mathcal{S}} |\int_{\mathcal{S}} \int_{\mathcal{A}} \mathbb{P}(s_t = ds, a_t = da, s_{t+1} = ds') - \int_{\mathcal{S}} \int_{\mathcal{A}} \mathbb{P}(\tilde{s}_t = ds, \tilde{a}_t = da, \tilde{s}_{t+1} = ds')| \\
&\leq \int_{\mathcal{S}} \int_{\mathcal{S}} \int_{\mathcal{A}} |\mathbb{P}(s_t = ds, a_t = da, s_{t+1} = ds') - \mathbb{P}(\tilde{s}_t = ds, \tilde{a}_t = da, \tilde{s}_{t+1} = ds')| \\
&= \int_{\mathcal{S}} \int_{\mathcal{S}} \int_{\mathcal{A}} |\mathbb{P}(O_t = (ds, da, ds')) - \mathbb{P}(\tilde{O}_t = (ds, da, ds'))| \\
&= 2d_{TV}(\mathbb{P}(O_t \in \cdot), \mathbb{P}(\tilde{O} \in \cdot)),
\end{aligned}$$

where the last equality requires the exchange of integral which is guaranteed by Fubini's theorem since $\mathbb{P}$ is an absolute integrable function.

For the second equality, we have

$$2d_{TV}(\mathbb{P}(O_t \in \cdot), \mathbb{P}(\tilde{O}_t \in \cdot))$$

$$= \int_{\mathcal{S}} \int_{\mathcal{A}} \int_{\mathcal{S}} |\mathbb{P}(O_t = (ds, da, ds')) - \mathbb{P}(\tilde{O}_t = (ds, da, ds'))|$$

$$= \int_{\mathcal{S}} \int_{\mathcal{A}} \int_{\mathcal{S}} |\mathcal{P}(ds'|s, a)\mathbb{P}((s_t, a_t) = (ds, da)) - \mathcal{P}(ds'|s, a)\mathbb{P}((\tilde{s}_t, \tilde{a}_t) = (ds, da))|$$

$$= \int_{\mathcal{S}} \int_{\mathcal{A}} \int_{\mathcal{S}} \mathcal{P}(ds'|s, a)|\mathbb{P}((s_t, a_t) = (ds, da)) - \mathbb{P}((\tilde{s}_t, \tilde{a}_t) = (ds, da))|$$

$$= \int_{\mathcal{S}} \int_{\mathcal{A}} |\mathbb{P}((s_t, a_t) = (ds, da)) - \mathbb{P}((\tilde{s}_t, \tilde{a}_t) = (ds, da))|$$

$$= 2d_{TV}(\mathbb{P}((s_t, a_t) \in \cdot), \mathbb{P}((\tilde{s}_t, \tilde{a}_t) \in \cdot)).$$

For the third inequality, since $\boldsymbol{\theta}_t$ is dependent on $s_t$ as shown in Eq. (9), it holds that

$$2d_{TV}(\mathbb{P}((s_t, a_t) \in \cdot), \mathbb{P}((\tilde{s}_t, \tilde{a}_t) \in \cdot))$$

$$= \int_{\mathcal{S}} \int_{\mathcal{A}} |\mathbb{P}(s_t = ds, a_t = da) - \mathbb{P}(\tilde{s}_t = ds, \tilde{a}_t = da)|$$

$$= \int_{\mathcal{S}} \int_{\mathcal{A}} | \int_{\boldsymbol{\theta}} \mathbb{P}(s_t = ds)\mathbb{P}(\boldsymbol{\theta}_t = d\boldsymbol{\theta}|s_t = s)\mathbb{P}(a_t = da|s_t = s, \boldsymbol{\theta}_t = \boldsymbol{\theta}) - \mathbb{P}(\tilde{s}_t = ds, \tilde{a}_t = da)|$$

$$= \int_{\mathcal{S}} \int_{\mathcal{A}} |\mathbb{P}(s_t = ds) \int_{\boldsymbol{\theta}} \mathbb{P}(\boldsymbol{\theta}_t = d\boldsymbol{\theta}|s_t = s)\pi_{\boldsymbol{\theta}_t}(da|s) - \mathbb{P}(\tilde{s}_t = ds)\pi_{\boldsymbol{\theta}_{t-\tau}}(da|s)|$$

$$= \int_{\mathcal{S}} \int_{\mathcal{A}} |\mathbb{P}(s_t = ds)\mathbb{E}[\pi_{\boldsymbol{\theta}_t}(da|s)|s_t = s] - \mathbb{P}(\tilde{s}_t = ds)\pi_{\boldsymbol{\theta}_{t-\tau}}(da|s)|$$

$$= \int_{\mathcal{S}} \int_{\mathcal{A}} |\mathbb{P}(s_t = ds)\mathbb{E}[\pi_{\boldsymbol{\theta}_t}(da|s)|s_t = s] - \mathbb{P}(s_t = ds)\pi_{\boldsymbol{\theta}_{t-\tau}}(da|s)|$$

$$+ \int_{\mathcal{S}} \int_{\mathcal{A}} |\mathbb{P}(s_t = ds)\pi_{\boldsymbol{\theta}_{t-\tau}}(da|s) - \mathbb{P}(\tilde{s}_t = ds)\pi_{\boldsymbol{\theta}_{t-\tau}}(da|s)|$$

$$= \int_{\mathcal{S}} \mathbb{P}(s_t = ds) \int_{\mathcal{A}} |\mathbb{E}[\pi_{\boldsymbol{\theta}_t}(da|s)|s_t = s] - \pi_{\boldsymbol{\theta}_{t-\tau}}(da|s)|$$

$$+ 2d_{TV}(\mathbb{P}(s_t \in \cdot), \mathbb{P}(\tilde{s}_t \in \cdot))$$

$$\leq L_\pi \mathbb{E}\|\boldsymbol{\theta}_t - \boldsymbol{\theta}_{t-\tau}\| + 2d_{TV}(\mathbb{P}(s_t \in \cdot), \mathbb{P}(\tilde{s}_t \in \cdot)),$$

where the last inequality holds due to the Lipschitz continuity of policy made in Assumption 4.7. $\square$

**Proof of Lemma C.3.**

*Proof.* By definition, we have

$$J(\theta_1) - J(\theta_2) = \mathbb{E}[r(s^1, a^1) - r(s^2, a^2)],$$

where $s^i \sim \mu_{\boldsymbol{\theta}_i}, a^i \sim \pi_{\boldsymbol{\theta}_i}$. Therefore, it holds that

$$J(\boldsymbol{\theta}_1) - J(\boldsymbol{\theta}_2) = \mathbb{E}[r(s^1, a^1) - r(s^1, a^1)]$$

$$\leq 2U_r d_{TV}(\mu_{\boldsymbol{\theta}_1} \otimes \pi_{\boldsymbol{\theta}_1}, \mu_{\boldsymbol{\theta}_2} \otimes \pi_{\boldsymbol{\theta}_2})$$

$$\leq 2U_r L_\pi (1 + \lceil \log_\rho C^{-1} \rceil + \frac{1}{1-\rho})\|\boldsymbol{\theta}_1 - \boldsymbol{\theta}_2\|$$

$$= L_J\|\boldsymbol{\theta}_1 - \boldsymbol{\theta}_2\|.$$

$\square$

**Proof of Lemma C.4.**

*Proof.* The proof of this lemma can be found in Lemma 3.2 of (Zhang et al., 2020a). $\square$

**Proof of Lemma C.5**.

*Proof.* We will divide the proof of this lemma into four steps.

**Step 1:** show that for all $k \in \{1, 2, \cdots, K\}$, we have

$$\|\boldsymbol{W}^{(k)}\| \leq \mathcal{O}(\sqrt{m}). \tag{35}$$

It can be shown that

$$\begin{aligned}
\|\boldsymbol{W}^{(k)}\| &\leq \|\boldsymbol{W}^{(k)} - \boldsymbol{W}_0^{(k)}\| + \|\boldsymbol{W}_0^{(K)}\| \\
&\leq U_{\boldsymbol{\omega}} + \|\boldsymbol{W}_0^{(k)}\| \\
&\leq \mathcal{O}(\sqrt{m}),
\end{aligned}$$

where the last inequality id due to Assumption 4.2 and the fact that $U_{\boldsymbol{\omega}}$ is constant to $m$.

**Step 2:** show that for all $k \in \{1, 2, \cdots, K\}$, we have

$$\|s^{(k)}\| \leq \mathcal{O}(\sqrt{m}). \tag{36}$$

From Assumption 4.1, we have $\|s^{(0)}\| \leq 1$. From Eq. (35), it holds that

$$\begin{aligned}
\|s^{(1)}\| = \|\frac{1}{\sqrt{m}}\sigma(\boldsymbol{W}^{(1)}s^{(0)})\| \\
\leq \frac{1}{m}L_a^2\|\boldsymbol{W}^{(1)}\|^2\|s^{(0)}\|^2 + \|\sigma(0)\|^2 \\
\leq \mathcal{O}(m).
\end{aligned}$$

By induction, suppose $\|s^{(k)}\|^2 \leq \mathcal{O}(m)$. We have

$$\begin{aligned}
\|s^{(k+1)}\|^2 = \|\frac{1}{\sqrt{m}}\sigma(\boldsymbol{W}^{(k+1)}s^{(k)})\|^2 \\
\leq \frac{1}{m}L_a^2\|\boldsymbol{W}^{(k+1)}\|^2\|s^{(k)}\|^2 + \|\sigma(0)\|^2 \\
\leq \mathcal{O}(m),
\end{aligned}$$

which concludes the proof. Therefore, from Eq. (36), it can be shown that

$$\|\widehat{V}(\boldsymbol{\omega}; s)\| = \|\frac{1}{\sqrt{m}}\boldsymbol{b}^\top s^{(K)}\| \leq \mathcal{O}(1).$$

**Step 3:** show that for all $k \in \{1, 2, \cdots, K\}$, we have

$$\|\nabla_{s^{(k-1)}}s^{(k)}\| \leq \mathcal{O}(1). \tag{37}$$

From the chain rule, we have

$$\nabla_{s^{(k-1)}}s^{(k)}(i, j) = \frac{1}{\sqrt{m}}\sigma'(\sum_j \boldsymbol{W}^{(k)}(i, j)s^{(k-1)}(j))\boldsymbol{W}^{(k)}(i, j).$$

Therefore, we get

$$\begin{aligned}
\|\nabla_{s^{(k-1)}}s^{(k)}\|^2 &= \sup_{\|v\|=1} \sum_{i=1}^m (\sum_j \nabla_{s^{(k-1)}}s^{(k)}(i, j)v_j)^2 \\
&= \sup_{\|v\|=1} \frac{1}{m}\|\Sigma'\boldsymbol{W}^{(k)}v\|^2 \\
&\leq \frac{1}{m}\|\Sigma'\|^2 \cdot \|\boldsymbol{W}^{(k)}\|^2 \\
&\leq \mathcal{O}(1),
\end{aligned}$$

where $\Sigma'$ is a diagonal matrix with $\Sigma'(i, i) = \sigma'(\Sigma_j \boldsymbol{W}^{(k)}(i, j)s^{(k-1)}(j)) := \xi(i)$.

**Step 4:** show that for all $k \in \{1, 2, \cdots, K\}$, we have

$$\|\nabla_{\boldsymbol{W}^{(k)}} s^{(k)}\| \leq \mathcal{O}(1), \tag{38}$$

where $\nabla_{\boldsymbol{W}^{(k)}} s^{(k)}$ is defined to be a matrix whose $(I, (j-i)m + h)$'th entry $\nabla_{\boldsymbol{W}^{(k)}} s^{(k)}(i, j, h)$ is given by

$$\nabla_{\boldsymbol{W}^{(k)}} s^{(k)}(i, j, h) = \frac{\partial s^{(k)}(i)}{\partial \boldsymbol{W}^{(k)}(j, h)}.$$

It holds that

$$\nabla_{\boldsymbol{W}^{(k)}} s^{(k)}(i, j, j') = \frac{1}{\sqrt{m}} \mathbf{1}\{i - j\} \sigma'(\sum_h \boldsymbol{W}^{(k)}(i, h) s^{(k-1)}(h)) s^{(k-1)}(j'),$$

which can be written as

$$\nabla_{\boldsymbol{W}^{(k)}} s^{(k)}(i, j, j') = \frac{1}{\sqrt{m}} \mathbf{1}\{i = j\} \xi(i) s^{(k-1)}(j').$$

Therefore, we get

$$
\begin{aligned}
\|\nabla_{\boldsymbol{W}^{(k)} s^{(k)}}\|^2 &= \sup_{\|V\|_{\mathrm{F}}=1} \sum_{i=1}^m (\sum_{j,j'} \nabla_{\boldsymbol{W}^{(k)}} s^{(k)}(i, j, j') V_{j,j'})^2 \\
&= \frac{1}{m} \sup_{\|V\|_{\mathrm{F}}=1} \sum_{i=1}^m (\sum_{j,j'} \mathbf{1}\{i = j\} \xi(i) s^{(k-1)}(j') V_{j,j'})^2 \\
&= \frac{1}{m} \sup_{\|V\|_{\mathrm{F}}=1} \sum_{i=1}^m (\sum_{j,j'} \mathbf{1}\{i = j\} \xi(i) [V s^{(k-1)}]_j)^2 \\
&= \frac{1}{m} \sup_{\|V\|_{\mathrm{F}}=1} \sum_{i=1}^m \xi(i)^2 [V s^{(k-1)}]_i^2 \\
&= \sup_{\|V\|_{\mathrm{F}}=1} \frac{1}{m} \|\Sigma' V s^{(k-1)}\|^2 \\
&\leq \frac{1}{m} \|\Sigma'\|^2 \cdot \|s^{(k-1)}\|^2 \\
&\leq \mathcal{O}(1),
\end{aligned}
$$

where the last inequality follows Eq. (36).

We then show the Lipschitzness of the neural network. Since each entry of $b$ satisfies $|b_i| \leq 1$, it is easy to see that

$$\|\nabla_{s^{(K)}} \widehat{V}(\boldsymbol{\omega}; s)\| = \frac{1}{\sqrt{m}} \|\boldsymbol{b}\| \leq 1.$$

By Eq. (37),Eq. (38), and the chain rule, we have

$$\|\nabla_{\boldsymbol{W}^{(k)}} V(\boldsymbol{\omega}; s) = \|\nabla_{\boldsymbol{W}^{(K)}} V(\boldsymbol{\omega}; s) \nabla_{\boldsymbol{W}^{(K-1)}} s^{(K)} \cdots \nabla_{s^{(k)}} s^{(k+1)} \nabla_{\boldsymbol{W}^{(k)}} s^{(k)}\| \leq \mathcal{O}(1).$$

It can be shown that

$$\|\nabla_{\boldsymbol{\omega}} \widehat{V}(\boldsymbol{\omega}; s)\|^2 = \sup_{\|V\|_{\mathrm{F}}=1} \sum_{k=1}^K (\nabla_{\boldsymbol{W}^{(k)}} \widehat{V}(\boldsymbol{\omega}; s) V_k)^2 \leq \mathcal{O}(1),$$

which concludes the proof of Lipschitzness.

The proof of smoothness property has been shown in Liu et al. (2020). $\qquad\square$

# H    PROOF OF SUPPORTING LEMMAS

The following four lemmas only deal with the Markovian noise, which are originally proved in Wu et al. (2020b) and updated in Wu et al. (2020a). We include the proof with slight modifications for proving Theorem 4.9.

**Proof of Lemma F.1**.

*Proof.* We will divide the proof of this lemma into four steps.

**Step 1:** show that for any $\boldsymbol{\theta}_1, \boldsymbol{\theta}_2, \eta, O = (s, a, s')$, we have

$$|\Phi(O, \eta, \boldsymbol{\theta}_1) - \Phi(O, \eta, \boldsymbol{\theta}_2)| \leq 4U_r L_J \|\boldsymbol{\theta}_1 - \boldsymbol{\theta}_2\|. \tag{39}$$

By the definition of $\Phi(O, \eta, \boldsymbol{\theta})$ in Eq. (11), we have

$$
\begin{aligned}
|\Phi(O, \eta, \boldsymbol{\theta}_1) - \Phi(O, \boldsymbol{\theta}, \boldsymbol{\theta}_2)| &= |(\eta - J(\boldsymbol{\theta}_1))(r - J(\boldsymbol{\theta}_1)) - (\eta - J(\boldsymbol{\theta}_2))(r - J(\boldsymbol{\theta}_2))| \\
&\leq |(\eta - J(\boldsymbol{\theta}_1))(r - J(\boldsymbol{\theta}_1)) - (\eta - J(\boldsymbol{\theta}_1))(r - J(\boldsymbol{\theta}_2))| \\
&\quad + |(\eta - J(\boldsymbol{\theta}_1))(r - J(\boldsymbol{\theta}_2)) - (\eta - J(\boldsymbol{\theta}_2))(r - J(\boldsymbol{\theta}_2))| \\
&\leq 4U_r |J(\boldsymbol{\theta}_1) - J(\boldsymbol{\theta}_2)| \\
&\leq 4U_r L_J \|\boldsymbol{\theta}_1 - \boldsymbol{\theta}_2\|.
\end{aligned}
$$

**Step 2:** show that for any $\boldsymbol{\theta}, \eta_1, \eta_2, O$, we have

$$|\Phi(O, \eta_1, \boldsymbol{\theta}) - \Phi(O, \eta_2, \boldsymbol{\theta})| \leq 2U_r |\eta_1 - \eta_2|. \tag{40}$$

By definition, we have

$$
\begin{aligned}
|\Phi(O, \eta_1, \boldsymbol{\theta}) - \Phi(O, \eta_2, \boldsymbol{\theta})| &= |(\eta_1 - J(\boldsymbol{\theta}))(r - J(\boldsymbol{\theta})) - (\eta_2 - J(\boldsymbol{\theta}))(r - J(\boldsymbol{\theta}))| \\
&\leq 2U_r |\eta_1 - \eta_2|.
\end{aligned}
$$

**Step 3:** show that for original tuple $O_t$ and the auxiliary tuple $\widetilde{O}_t$, conditioned on $s_{t-\tau+1}$ and $\boldsymbol{\theta}_{t-\tau}$, we have

$$|\mathbb{E}[\Phi(O_t, \eta_{t-\tau}, \boldsymbol{\theta}_{t-\tau})] - \mathbb{E}[\Phi(\widetilde{O}_t, \eta_{t-\tau}, \boldsymbol{\theta}_{t-\tau})]| \leq 2U_r^2 L_\pi \sum_{k=t-\tau}^{t} \mathbb{E}\|\boldsymbol{\theta}_k - \boldsymbol{\theta}_{t-\tau}\|. \tag{41}$$

By definition, we have

$$\mathbb{E}[\Phi(O_t, \eta_{t-\tau}, \boldsymbol{\theta}_{t-\tau})] - \mathbb{E}[\Phi(\widetilde{O}_t, \eta_{t-\tau}, \boldsymbol{\theta}_{t-\tau})] = (\eta_{t-\tau} - J(\boldsymbol{\theta}_{t-\tau}))\mathbb{E}[r(s_t, a_t) - r(\widetilde{s}_t, \widetilde{a}_t)].$$

By definition of total variation norm, we have

$$\mathbb{E}[r(s_t, a_t) - r(\widetilde{s}_t, \widetilde{a}_t)] \leq 2U_r d_{TV}(\mathbb{P}(O_t \in \cdot | s_{t-\tau+1}, \boldsymbol{\theta}_{t-\tau}), \mathbb{P}(\widetilde{O}_t \in \cdot | s_{t-\tau+1}, \boldsymbol{\theta}_{t-\tau})). \tag{42}$$

By Lemma C.2, we get

$$
\begin{aligned}
&d_{TV}(\mathbb{P}(O_t \in \cdot | s_{t-\tau+1}, \boldsymbol{\theta}_{t-\tau}), \mathbb{P}(\widetilde{O}_t \in \cdot | s_{t-\tau+1}, \boldsymbol{\theta}_{t-\tau})) \\
&= d_{TV}(\mathbb{P}((s_t, a_t) \in \cdot | s_{t-\tau+1}, \boldsymbol{\theta}_{t-\tau}), \mathbb{P}((\widetilde{s}_t, \widetilde{a}_t) \in \cdot | s_{t-\tau+1}, \boldsymbol{\theta}_{t-\tau})) \\
&\leq d_{TV}(\mathbb{P}(s_t \in \cdot | s_{t-\tau+1}, \boldsymbol{\theta}_{t-\tau}), \mathbb{P}(\widetilde{s}_t \in \cdot | s_{t-\tau+1}, \boldsymbol{\theta}_{t-\tau})) + \frac{1}{2}L_\pi \mathbb{E}\|\boldsymbol{\theta}_t - \boldsymbol{\theta}_{t-\tau}\| \\
&\leq d_{TV}(\mathbb{P}(O_{t-1} \in \cdot | s_{t-\tau+1}, \boldsymbol{\theta}_{t-\tau}), \mathbb{P}(\widetilde{O}_{t-1} \in \cdot | s_{t-\tau+1}, \boldsymbol{\theta}_{t-\tau})) + \frac{1}{2}L_\pi \mathbb{E}\|\boldsymbol{\theta}_t - \boldsymbol{\theta}_{t-\tau}\|.
\end{aligned}
$$

Repeat the above argument from $t$ to $t - \tau$, we have

$$d_{TV}(\mathbb{P}(O_t \in \cdot | s_{t-\tau+1}, \boldsymbol{\theta}_{t-\tau}), \mathbb{P}(\widetilde{O}_t \in \cdot | s_{t-\tau+1}, \boldsymbol{\theta}_{t-\tau})) \leq \frac{1}{2}L_\pi \sum_{k=t-\tau}^{t} \mathbb{E}\|\boldsymbol{\theta}_k - \boldsymbol{\theta}_{t-\tau}\|. \tag{43}$$

Plugging Eq. (43) into Eq. (42), we have

$$|\mathbb{E}[\Phi(O_t, \eta_{t-\tau}, \boldsymbol{\theta}_{t-\tau})] - \mathbb{E}[\Phi(\widetilde{O}_t, \eta_{t-\tau}, \boldsymbol{\theta}_{t-\tau})]| \leq 2U_r^2 L_\pi \sum_{k=t-\tau}^{t} \mathbb{E}\|\boldsymbol{\theta}_k - \boldsymbol{\theta}_{t-\tau}\|.$$

**Step 4:** show that conditioned on $s_{t-\tau+1}$ and $\boldsymbol{\theta}_{t-\tau}$, we have

$$\mathbb{E}[\Phi(\widetilde{O}_t, \eta_{t-\tau}, \boldsymbol{\theta}_{t-\tau})] \leq 4U_r^2 C\rho^{\tau-1}. \tag{44}$$

Note that according to definition, we have

$$\mathbb{E}[\Phi(O'_{t-\tau}, \eta_{t-\tau}, \boldsymbol{\theta}_{t-\tau})|\boldsymbol{\theta}_{t-\tau}] = 0,$$

where $O'_{t-\tau} = (s'_{t-\tau}, a'_{t-\tau}, s'_{t-\tau+1})$ is the tuple generated by $s'_{t-\tau} \sim \mu_{\boldsymbol{\theta}_{t-\tau}}, a'_{t-\tau} \sim \pi_{\boldsymbol{\theta}_{t-\tau}}, s'_{t-\tau+1} \sim \mathcal{P}$. From the uniform ergodicity in Assumption 4.6, it shows that

$$d_{TV}(\mathbb{P}(\widetilde{s}_t = \cdot|s_{t-\tau+1}, \boldsymbol{\theta}_{t-\tau}), \mu_{\boldsymbol{\theta}_{t-\tau}}) \leq C\rho^{\tau-1}.$$

Then we have

$$\begin{aligned}
\mathbb{E}[\Phi(\widetilde{O}_t, \eta_{t-\tau}, \boldsymbol{\theta}_{t-\tau})] &= \mathbb{E}[\Phi(\widetilde{O}_t, \eta_{t-\tau}, \boldsymbol{\theta}_{t-\tau}) - \Phi(O'_{t-\tau}, \eta_{t-\tau}, \boldsymbol{\theta}_{t-\tau})] \\
&= \mathbb{E}[(\eta_{t-\tau} - J(\boldsymbol{\theta}_{t-\tau}))(r(\widetilde{s}_t, \widetilde{a}_t) - r(s'_{t-\tau}, a'_{t-\tau}))] \\
&\leq 4U_r^2 d_{TV}(\mathbb{P}(\widetilde{O}_{t-\tau} = \cdot|s_{t-\tau+1}, \boldsymbol{\theta}_{t-\tau}), \mu_{\boldsymbol{\theta}_{t-\tau}} \otimes \pi_{\boldsymbol{\theta}_{t-\tau}} \otimes \mathcal{P}) \\
&\leq 4U_r^2 C\rho^{\tau-1}.
\end{aligned}$$

Combing Eq. (39), Eq. (40), Eq. (41), and Eq. (44), we have

$$\begin{aligned}
\mathbb{E}[\Phi(O_t, \eta_t, \boldsymbol{\theta}_t)] &= \mathbb{E}[\Phi(O_t, \eta_t, \boldsymbol{\theta}_t) - \Phi(O_t, \eta_t, \boldsymbol{\theta}_{t-\tau})] + \mathbb{E}[\Phi(O_t, \eta_t, \boldsymbol{\theta}_{t-\tau}) - \Phi(O_t, \eta_{t-\tau}, \boldsymbol{\theta}_{t-\tau})] \\
&\quad + \mathbb{E}[\Phi(O_t, \eta_{t-\tau}, \boldsymbol{\theta}_{t-\tau}) - \Phi(\widetilde{O}_t, \eta_{t-\tau}, \boldsymbol{\theta}_{t-\tau})] + \mathbb{E}[\Phi(\widetilde{O}_t, \eta_{t-\tau}, \boldsymbol{\theta}_{t-\tau})] \\
&\leq 4U_r L_J\|\boldsymbol{\theta}_t - \boldsymbol{\theta}_{t-\tau}\| + 2U_r|\eta_t - \eta_{t-\tau}| + 2U_r^2 L_\pi \sum_{i=t-\tau}^t \mathbb{E}\|\boldsymbol{\theta}_i - \boldsymbol{\theta}_{t-\tau}\| \\
&\quad + 4U_r^2 C\rho^{\tau-1},
\end{aligned}$$

which concludes the proof. $\qquad\square$

**Proof of Lemma F.3.**

*Proof.* We will divide the proof of this lemma into four steps.

**Step 1:** show that for any $\boldsymbol{\theta}_1, \boldsymbol{\theta}_2, \boldsymbol{\omega}$ and tuple $O = (s, a, s')$, we have

$$|\Psi(O, \boldsymbol{\omega}, \boldsymbol{\theta}_1) - \Psi(O, \boldsymbol{\omega}, \boldsymbol{\theta}_2) \leq C_1\|\boldsymbol{\theta}_1 - \boldsymbol{\theta}_2\|, \tag{45}$$

where $C_1 = 2U_\delta^2 L_\pi(1 + \lceil\log_\rho C^{-1}\rceil + \frac{1}{1-\rho}) + 2U_\delta L_J L_v + 2U_\delta L_* L_v$.

By definition of $\Psi(O, \boldsymbol{\omega}, \boldsymbol{\theta})$ in Eq. (11), we have

$$\begin{aligned}
&|\Psi(O, \boldsymbol{\omega}, \boldsymbol{\theta}_1) - \Psi(O, \boldsymbol{\omega}, \boldsymbol{\theta}_2)| \\
&= |\langle\boldsymbol{\omega} - \boldsymbol{\omega}_1^*, g(O, \boldsymbol{\omega}, \boldsymbol{\theta}_1) - \bar{g}(\boldsymbol{\omega}, \boldsymbol{\theta}_1)\rangle - \langle\boldsymbol{\omega} - \boldsymbol{\omega}_2^*, g(O, \boldsymbol{\omega}, \boldsymbol{\theta}_2) - \bar{g}(\boldsymbol{\omega}, \boldsymbol{\theta}_2)\rangle| \\
&\leq \underbrace{|\langle\boldsymbol{\omega} - \boldsymbol{\omega}_1^*, g(O, \boldsymbol{\omega}, \boldsymbol{\theta}_1) - \bar{g}(\boldsymbol{\omega}, \boldsymbol{\theta}_1)\rangle - \langle\boldsymbol{\omega} - \boldsymbol{\omega}_1^*, g(O, \boldsymbol{\omega}, \boldsymbol{\theta}_2) - \bar{g}(\boldsymbol{\omega}, \boldsymbol{\theta}_2)\rangle|}_{I_1} \\
&\quad + \underbrace{|\langle\boldsymbol{\omega} - \boldsymbol{\omega}_1^*, g(O, \boldsymbol{\omega}, \boldsymbol{\theta}_2) - \bar{g}(\boldsymbol{\omega}, \boldsymbol{\theta}_2)\rangle - \langle\boldsymbol{\omega} - \boldsymbol{\omega}_2^*, g(O, \boldsymbol{\omega}, \boldsymbol{\theta}_2) - \bar{g}(\boldsymbol{\omega}, \boldsymbol{\theta}_2)\rangle|}_{I_2}.
\end{aligned}$$

For term $I_1$, we have

$$\begin{aligned}
I_1 &= |\langle\boldsymbol{\omega} - \boldsymbol{\omega}_1^*, g(O, \boldsymbol{\omega}, \boldsymbol{\theta}_1) - \bar{g}(\boldsymbol{\omega}, \boldsymbol{\theta}_1)\rangle - \langle\boldsymbol{\omega} - \boldsymbol{\omega}_1^*, g(O, \boldsymbol{\omega}, \boldsymbol{\theta}_2) - \bar{g}(\boldsymbol{\omega}, \boldsymbol{\theta}_2)\rangle| \\
&= |\langle\boldsymbol{\omega} - \boldsymbol{\omega}_1^*, g(O, \boldsymbol{\omega}, \boldsymbol{\theta}_1) - g(O, \boldsymbol{\omega}, \boldsymbol{\theta}_2)\rangle| + |\langle\boldsymbol{\omega} - \boldsymbol{\omega}_1^*, \bar{g}(\boldsymbol{\omega}, \boldsymbol{\theta}_1) - \bar{g}(\boldsymbol{\omega}, \boldsymbol{\theta}_2)\rangle| \\
&= |\langle\boldsymbol{\omega} - \boldsymbol{\omega}_1^*, (J(\boldsymbol{\theta}_1) - J(\boldsymbol{\theta}_2))\nabla_{\boldsymbol{\omega}}\widehat{V}(\boldsymbol{\omega}; s)\rangle| + |\langle\boldsymbol{\omega} - \boldsymbol{\omega}_1^*, \bar{g}(\boldsymbol{\omega}, \boldsymbol{\theta}_1) - \bar{g}(\boldsymbol{\omega}, \boldsymbol{\theta}_2)\rangle| \\
&\leq 2U_{\boldsymbol{\omega}} L_J L_v\|\boldsymbol{\theta}_1 - \boldsymbol{\theta}_2\| + 2U_{\boldsymbol{\omega}}\|\bar{g}(\boldsymbol{\omega}, \boldsymbol{\theta}_1) - \bar{g}(\boldsymbol{\omega}, \boldsymbol{\theta}_2)\| \\
&\leq 2U_{\boldsymbol{\omega}} L_J L_v\|\boldsymbol{\theta}_1 - \boldsymbol{\theta}_2\| + 2U_{\boldsymbol{\omega}} \cdot 2U_\delta d_{TV}(\mu_{\boldsymbol{\theta}_1} \otimes \pi_{\boldsymbol{\theta}_1} \otimes \mathcal{P}, \mu_{\boldsymbol{\theta}_2} \otimes \pi_{\boldsymbol{\theta}_2} \otimes \mathcal{P}) \\
&\leq 2U_{\boldsymbol{\omega}} L_J L_v\|\boldsymbol{\theta}_1 - \boldsymbol{\theta}_2\| + 2U_\delta^2 d_{TV}(\mu_{\boldsymbol{\theta}_1} \otimes \pi_{\boldsymbol{\theta}_1} \otimes \mathcal{P}, \mu_{\boldsymbol{\theta}_2} \otimes \pi_{\boldsymbol{\theta}_2} \otimes \mathcal{P}) \\
&\leq (2U_\delta L_J L_v + 2U_\delta^2 L_\pi(1 + \lceil\log_\rho C^{-1}\rceil + \frac{1}{1-\rho}))\|\boldsymbol{\theta}_1 - \boldsymbol{\theta}_2\|,
\end{aligned}$$

where we use the fact that $U_\delta = 2U_r + 2U_\omega + 2U_v$ and the last inequality comes from Lemma C.1.

For term $I_2$, from Cauchy-Schwartz inequality, we have

$$
\begin{aligned}
I_2 &= |\langle \boldsymbol{\omega} - \boldsymbol{\omega}_1^*, g(O, \boldsymbol{\omega}, \boldsymbol{\theta}_2) - \bar{g}(\boldsymbol{\omega}, \boldsymbol{\theta}_2)\rangle - \langle \boldsymbol{\omega} - \boldsymbol{\omega}_2^*, g(O, \boldsymbol{\omega}, \boldsymbol{\theta}_2) - \bar{g}(\boldsymbol{\omega}, \boldsymbol{\theta}_2)\rangle| \\
&= |\langle \boldsymbol{\omega}_1^* - \boldsymbol{\omega}_2^*, g(O, \boldsymbol{\omega}, \boldsymbol{\theta}_2) - \bar{g}(\boldsymbol{\omega}, \boldsymbol{\theta}_2)\rangle| \\
&\leq 2U_\delta L_v \|\boldsymbol{\omega}_1^* - \boldsymbol{\omega}_2^*\| \\
&\leq 2U_\delta L_v L_* \|\boldsymbol{\theta}_1 - \boldsymbol{\theta}_2\|.
\end{aligned}
$$

Combining the results from $I_1$ and $I_2$, we get

$$
|\Psi(O, \boldsymbol{\omega}, \boldsymbol{\theta}_1) - \Psi(O, \boldsymbol{\omega}, \boldsymbol{\theta}_2) \leq C_1 \|\boldsymbol{\theta}_1 - \boldsymbol{\theta}_2\|,
$$

where $C_1 = 2U_\delta^2 L_\pi (1 + \lceil \log_\rho C^{-1} \rceil + \frac{1}{1-\rho}) + 2U_\delta L_J L_v + 2U_\delta L_* L_v$.

**Step 2:** show that for any $\boldsymbol{\theta}, \boldsymbol{\omega}_1, \boldsymbol{\omega}_2$ and tuple $O(s, a, s')$, we have

$$
|\Psi(O, \boldsymbol{\omega}_1, \boldsymbol{\theta}) - \Psi(O, \boldsymbol{\omega}_2, \boldsymbol{\theta})| \leq 2U_\delta (U_v H_v + L_v^2 + U_r H_v + L_v) \|\boldsymbol{\omega}_1 - \boldsymbol{\omega}_2\|. \tag{46}
$$

By definition, we have

$$
\begin{aligned}
&|\Psi(O, \boldsymbol{\omega}_1, \boldsymbol{\theta}) - \Psi(O, \boldsymbol{\omega}_2, \boldsymbol{\theta})| \\
&= |\langle \boldsymbol{\omega}_1 - \boldsymbol{\omega}^*, g(O, \boldsymbol{\omega}_1, \boldsymbol{\theta}) - \bar{g}(\boldsymbol{\omega}_1, \boldsymbol{\theta})\rangle - \langle \boldsymbol{\omega}_2 - \boldsymbol{\omega}^*, g(O, \boldsymbol{\omega}_2, \boldsymbol{\theta}) - \bar{g}(\boldsymbol{\omega}_2, \boldsymbol{\theta})\rangle| \\
&\leq |\langle \boldsymbol{\omega}_1 - \boldsymbol{\omega}^*, g(O, \boldsymbol{\omega}_1, \boldsymbol{\theta}) - \bar{g}(\boldsymbol{\omega}_1, \boldsymbol{\theta})\rangle - \langle \boldsymbol{\omega}_1 - \boldsymbol{\omega}^*, g(O, \boldsymbol{\omega}_2, \boldsymbol{\theta}) - \bar{g}(\boldsymbol{\omega}_2, \boldsymbol{\theta})\rangle| \\
&\quad + |\langle \boldsymbol{\omega}_1 - \boldsymbol{\omega}^*, g(O, \boldsymbol{\omega}_2, \boldsymbol{\theta}) - \bar{g}(\boldsymbol{\omega}_2, \boldsymbol{\theta})\rangle - \langle \boldsymbol{\omega}_2 - \boldsymbol{\omega}^*, g(O, \boldsymbol{\omega}_2, \boldsymbol{\theta}) - \bar{g}(\boldsymbol{\omega}_2, \boldsymbol{\theta})\rangle| \\
&\leq 2U_\omega \|(g(O, \boldsymbol{\omega}_1, \boldsymbol{\theta}) - g(O, \boldsymbol{\omega}_2, \boldsymbol{\theta})) - (\bar{g}(\boldsymbol{\omega}_1, \boldsymbol{\theta}) - \bar{g}(\boldsymbol{\omega}_2, \boldsymbol{\theta}))\| + 2U_\delta L_v \|\boldsymbol{\omega}_1 - \boldsymbol{\omega}_2\|.
\end{aligned}
$$

It follows that

$$
\begin{aligned}
&\|(g(O, \boldsymbol{\omega}_1, \boldsymbol{\theta}) - g(O, \boldsymbol{\omega}_2, \boldsymbol{\theta})) - (\bar{g}(\boldsymbol{\omega}_1, \boldsymbol{\theta}) - \bar{g}(\boldsymbol{\omega}_2, \boldsymbol{\theta}))\| \\
&= \|(r(s, a) - J(\boldsymbol{\theta}))(\nabla_{\boldsymbol{\omega}} \widehat{V}(\boldsymbol{\omega}_1; s) - \nabla_{\boldsymbol{\omega}} \widehat{V}(\boldsymbol{\omega}_2; s)) \\
&\quad + \widehat{V}(\boldsymbol{\omega}_1; s') \nabla_{\boldsymbol{\omega}} \widehat{V}(\boldsymbol{\omega}_1; s) - \widehat{V}(\boldsymbol{\omega}_2; s') \nabla_{\boldsymbol{\omega}} (\boldsymbol{\omega}_2; s) \\
&\quad + \widehat{V}(\boldsymbol{\omega}_2; s) \nabla_{\boldsymbol{\omega}} \widehat{V}(\boldsymbol{\omega}_2; s) - \widehat{V}(\boldsymbol{\omega}_1; s) \nabla_{\boldsymbol{\omega}} \widehat{V}(\boldsymbol{\omega}_1; s)\| \\
&\leq \|\widehat{V}(\boldsymbol{\omega}_1; s') \nabla_{\boldsymbol{\omega}} \widehat{V}(\boldsymbol{\omega}_1; s) - \widehat{V}(\boldsymbol{\omega}_1; s') \nabla_{\boldsymbol{\omega}} \widehat{V}(\boldsymbol{\omega}_2; s) \\
&\quad + \widehat{V}(\boldsymbol{\omega}_1; s') \nabla_{\boldsymbol{\omega}} \widehat{V}(\boldsymbol{\omega}_2; s) - \widehat{V}(\boldsymbol{\omega}_2; s') \nabla_{\boldsymbol{\omega}} \widehat{V}(\boldsymbol{\omega}_2; s)\| \\
&\quad + \|\widehat{V}(\boldsymbol{\omega}_2; s) \nabla_{\boldsymbol{\omega}} \widehat{V}(\boldsymbol{\omega}_2; s) - \widehat{V}(\boldsymbol{\omega}_1; s) \nabla_{\boldsymbol{\omega}} \widehat{V}(\boldsymbol{\omega}_2; s) \\
&\quad + \widehat{V}(\boldsymbol{\omega}_1; s) \nabla_{\boldsymbol{\omega}} \widehat{V}(\boldsymbol{\omega}_2; s) - \widehat{V}(\boldsymbol{\omega}_1; s) \nabla_{\boldsymbol{\omega}} \widehat{V}(\boldsymbol{\omega}_1; s)\| + 2U_r H_v \|\boldsymbol{\omega}_1 - \boldsymbol{\omega}_2\| \\
&\leq 2U_v H_v \|\boldsymbol{\omega}_1 - \boldsymbol{\omega}_2\| + 2L_v^2 \|\boldsymbol{\omega}_1 - \boldsymbol{\omega}_2\| + 2U_r H_v \|\boldsymbol{\omega}_1 - \boldsymbol{\omega}_2\| \\
&= (2U_v H_v + 2L_v^2 + 2U_r H_v) \|\boldsymbol{\omega}_1 - \boldsymbol{\omega}_2\|.
\end{aligned}
$$

Therefore, we obtain

$$
|\Psi(O, \boldsymbol{\omega}_1, \boldsymbol{\theta}) - \Psi(O, \boldsymbol{\omega}_2, \boldsymbol{\theta})| \leq C_2 \|\boldsymbol{\omega}_1 - \boldsymbol{\omega}_2\|,
$$

where $C_2 = 2U_\delta (U_v H_v + L_v^2 + U_r H_v + L_v)$.

**Step 3:** show that for tuples $O_t = (s_t, a_t, s_{t+1})$ and $\widetilde{O}_t = (\widetilde{s}_t, \widetilde{a}_t, \widetilde{s}_{t+1})$. Conditioning on $s_{t-\tau+1}$ and $\boldsymbol{\theta}_{t-\tau}$, we have

$$
\mathbb{E}[\Psi(O_t, \boldsymbol{\omega}_{t-\tau}, \boldsymbol{\theta}_{t-\tau}) - \Psi(\widetilde{O}_t, \boldsymbol{\omega}_{t-\tau}, \boldsymbol{\theta}_{t-\tau})] \leq U_\delta^2 L_v L_\pi G \tau (\tau + 1)\alpha. \tag{47}
$$

By the definition of total variation norm, we have

$$
\begin{aligned}
&\mathbb{E}[\Psi(O_t, \boldsymbol{\omega}_{t-\tau}, \boldsymbol{\theta}_{t-\tau}) - \Psi(\widetilde{O}_t, \boldsymbol{\omega}_{t-\tau}, \boldsymbol{\theta}_{t-\tau})] \\
&= \mathbb{E}[\langle \boldsymbol{\omega}_{t-\tau} - \boldsymbol{\omega}_{t-\tau}^*, g(O_t, \boldsymbol{\omega}_{t-\tau}, \boldsymbol{\theta}_{t-\tau}) - g(\widetilde{O}_t, \boldsymbol{\omega}_{t-\tau}, \boldsymbol{\theta}_{t-\tau}))] \\
&\leq 2U_\delta^2 L_v d_{TV}(\mathbb{P}(O_t \in \cdot | s_{t-\tau+1}, \boldsymbol{\theta}_{-\tau}), \mathbb{P}(\widetilde{O}_t \in \cdot | s_{t-\tau+1}, \boldsymbol{\theta}_{t-\tau})) \\
&\overset{(1)}{\leq} U_\delta^2 L_v L_\pi \sum_{k=t-\tau}^{t} \mathbb{E}\|\boldsymbol{\theta}_k - \boldsymbol{\theta}_{t-\tau}\| \\
&\leq U_\delta^2 L_v L_\pi G \tau (\tau + 1)\alpha,
\end{aligned}
$$

where (1) follows from Eq. (43).

**Step 4:** show that conditioning on $s_{t-\tau+1}$ and $\boldsymbol{\theta}_{t-\tau}$,

$$\mathbb{E}[\Psi(\widetilde{O}_t, \boldsymbol{\omega}_{t-\tau}, \boldsymbol{\theta}_{t-\tau})] \le 2U_\delta^2 C \rho^{\tau-1} \tag{48}$$

From the definition of $\Psi(O, \boldsymbol{\omega}, \boldsymbol{\theta})$, we have

$$\mathbb{E}[\Psi(O'_{t-\tau}, \boldsymbol{\omega}_{t-\tau}, \boldsymbol{\theta}_{t-\tau})|s_{t-\tau+1}, \boldsymbol{\theta}_{t-\tau}] = 0,$$

where $O'_{t-\tau}$ is the tuple generated by $s'_{t-\tau} \sim \mu_{\boldsymbol{\theta}_{t-\tau}}, a'_{t-\tau} \sim \pi_{\boldsymbol{\theta}_{t-\tau}}, s'_{t-\tau+1} \sim \mathcal{P}$. From Assumption 4.6, we have

$$d_{TV}(\mathbb{P}(\widetilde{s}_t = \cdot | s_{t-\tau+1}, \boldsymbol{\theta}_{t-\tau}), \mu_{\boldsymbol{\theta}_{t-\tau}}) \le C\rho^{\tau-1}.$$

Then, it holds that

$$\begin{aligned}
\mathbb{E}[\Psi(\widetilde{O}_t, \boldsymbol{\omega}_{t-\tau}, \boldsymbol{\theta}_{t-\tau})] &= \mathbb{E}[\Psi(\widetilde{O}_t, \boldsymbol{\omega}_{t-\tau}, \boldsymbol{\theta}_{t-\tau}) - \Psi(O'_{t-\tau}, \boldsymbol{\omega}_{t-\tau}, \boldsymbol{\theta}_{t-\tau})] \\
&= \mathbb{E}\langle \boldsymbol{\omega}_{t-\tau} - \boldsymbol{\omega}^*_{t-\tau}, g(\widetilde{O}_t, \boldsymbol{\omega}_{t-\tau}, \boldsymbol{\theta}_{t-\tau}) - g(O'_{t-\tau}, \boldsymbol{\omega}_{t-\tau}, \boldsymbol{\theta}_{t-\tau})\rangle \\
&\le 2U_\delta^2 L_v d_{TV}(\mathbb{P}(\widetilde{O}_t = \cdot | s_{t-\tau+1}, \boldsymbol{\theta}_{t-\tau}), \mu_{\boldsymbol{\theta}_{t-\tau}} \otimes \pi_{\boldsymbol{\theta}_{t-\tau}} \otimes \mathcal{P}) \\
&= 2U_\delta^2 L_v d_{TV}(\mathbb{P}((\widetilde{s}_t, \widetilde{a}_t) \in \cdot | s_{t-\tau+1}, \boldsymbol{\theta}_{t-\tau}), \mu_{\boldsymbol{\theta}_{t-\tau}} \otimes \pi_{\boldsymbol{\theta}_{t-\tau}}) \\
&= 2U_\delta^2 L_v d_{TV}(\mathbb{P}(\widetilde{s}_t = \cdot | s_{t-\tau+1}, \boldsymbol{\theta}_{t-\tau}), \mu_{\boldsymbol{\theta}_{t-\tau}}) \\
&\le 2U_\delta^2 L_v C \rho^{\tau-1}.
\end{aligned}$$

Combining Eq. (45), Eq. (46), Eq. (47), and Eq. (48), we have

$$\begin{aligned}
\mathbb{E}[\Psi(O_t, \boldsymbol{\omega}_t, \boldsymbol{\theta}_t)] &= \mathbb{E}[\Psi(O_t, \boldsymbol{\omega}_t, \boldsymbol{\theta}_t) - \Psi(O_t, \boldsymbol{\omega}_t, \boldsymbol{\theta}_{t-\tau})] \\
&\quad + \mathbb{E}[\Psi(O_t, \boldsymbol{\omega}_t, \boldsymbol{\theta}_{t-\tau}) - \Psi(O_t, \boldsymbol{\omega}_{t-\tau}, \boldsymbol{\theta}_{t-\tau})] \\
&\quad + \mathbb{E}[\Psi(O_t, \boldsymbol{\omega}_{t-\tau}, \boldsymbol{\theta}_{t-\tau}) - \Psi(\widetilde{O}_t, \boldsymbol{\omega}_{t-\tau}, \boldsymbol{\theta}_{t-\tau})] \\
&\quad + \mathbb{E}[\Psi(\widetilde{O}_t, \boldsymbol{\omega}_{t-\tau}, \boldsymbol{\theta}_{t-\tau})] \\
&\le C_1\|\boldsymbol{\theta}_t - \boldsymbol{\theta}_{t-\tau}\| + C_2\|\boldsymbol{\omega}_t - \boldsymbol{\omega}_{t-\tau}\| \\
&\quad + U_\delta^2 L_v L_\pi G\tau(\tau+1)\alpha + 2U_\delta^2 L_v C\rho^{\tau-1},
\end{aligned}$$

where $C_1 = 2U_\delta^2 L_\pi(1 + \lceil\log_\rho C^{-1}\rceil + \frac{1}{1-\rho}) + 2U_\delta L_J L_v + 2U_\delta L_* L_v$ and $C_2 = 2U_\delta(U_v H_v + L_v^2 + U_r H_v + L_v)$. $\qquad\square$

**Proof of Lemma F.4.**

*Proof.* We will divide the proof of this lemma into four steps.

**Step 1:** show that for any $O, \boldsymbol{\omega}, \boldsymbol{\theta}_1, \boldsymbol{\theta}_2$, we have

$$\|\Xi(O, \boldsymbol{\omega}, \boldsymbol{\theta}_1) - \Xi(O, \boldsymbol{\omega}, \boldsymbol{\theta}_2)\| \le (3U_\delta L_h + 2U_\delta BL_*)\|\boldsymbol{\theta}_1 - \boldsymbol{\theta}_2\| \tag{49}$$

Since $\Xi(O, \boldsymbol{\omega}, \boldsymbol{\theta}) = \langle \boldsymbol{\omega} - \boldsymbol{\omega}^*, (\nabla\boldsymbol{\omega}^*_{\boldsymbol{\theta}})^\top (\mathbb{E}_{O'}[h(O', \boldsymbol{\theta})] - h(O, \boldsymbol{\theta}))\rangle$, we define $\mathbb{E}_{\boldsymbol{\theta}}[h(O', \boldsymbol{\theta})] := \mathbb{E}_{O'}[h(O', \boldsymbol{\theta})]$, where $\mathbb{E}_{\boldsymbol{\theta}}$ is the shorthand of $\mathbb{E}_{O' \sim (\mu_{\boldsymbol{\theta}}, \pi_{\boldsymbol{\theta}}, \mathcal{P})}$. In the following, we will show that each term in $\Xi(O, \boldsymbol{\omega}, \boldsymbol{\theta})$ is Lipschitz with respect to $\boldsymbol{\theta}$.

Term $\boldsymbol{\omega}$ is not related to $\boldsymbol{\theta}$, term $\boldsymbol{\omega}^* := \boldsymbol{\omega}^*(\boldsymbol{\theta})$ is $L_*$-Lipschitz, and term $\nabla\boldsymbol{\omega}^*_{\boldsymbol{\theta}}$ is $L_s$-Lipschitz.

For term $h(O, \boldsymbol{\theta})$, denote $\delta(O, \boldsymbol{\theta}) := r(s, a) - J(\boldsymbol{\theta}) + \widehat{V}(\boldsymbol{\omega}^*(\boldsymbol{\theta}); s') - \widehat{V}(\boldsymbol{\omega}^*(\boldsymbol{\theta}); s)$, we have

$$\begin{aligned}
&\|h(O, \boldsymbol{\theta}_1) - h(O, \boldsymbol{\theta}_2)\| \\
&= \|\delta(O, \boldsymbol{\theta}_1)\nabla\log\pi_{\boldsymbol{\theta}_1}(a|s) - \delta(O, \boldsymbol{\theta}_2)\nabla\log\pi_{\boldsymbol{\theta}_2}(a|s)\| \\
&\le \|\delta(O, \boldsymbol{\theta}_1)\nabla\log\pi_{\boldsymbol{\theta}_1}(a|s) - \delta(O, \boldsymbol{\theta}_1)\nabla\log\pi_{\boldsymbol{\theta}_2}(a|s)\| \\
&\quad + \|\delta(O, \boldsymbol{\theta}_1)\nabla\log\pi_{\boldsymbol{\theta}_2}(a|s) - \delta(O, \boldsymbol{\theta}_2)\nabla\log\pi_{\boldsymbol{\theta}_2}(a|s)\| \\
&\le U_\delta L_l\|\boldsymbol{\theta}_1 - \boldsymbol{\theta}_2\| + B|\delta(O, \boldsymbol{\theta}_1) - \delta(O, \boldsymbol{\theta}_2)| \\
&\le U_\delta L_l\|\boldsymbol{\theta}_1 - \boldsymbol{\theta}_2\| + B(|J(\boldsymbol{\theta}_1) - J(\boldsymbol{\theta}_2)| + \|\widehat{V}(\boldsymbol{\omega}^*(\boldsymbol{\theta}_1); s') - \widehat{V}(\boldsymbol{\omega}^*(\boldsymbol{\theta}_2); s')\| \\
&\quad + \|\widehat{V}(\boldsymbol{\omega}^*(\boldsymbol{\theta}_1); s) - \widehat{V}(\boldsymbol{\omega}^*(\boldsymbol{\theta}_2); s)\| \\
&\le (U_\delta L_l + 2BL_J)\|\boldsymbol{\theta}_1 - \boldsymbol{\theta}_2\| + 2BL_v\|\boldsymbol{\omega}^*(\boldsymbol{\theta}_1) - \boldsymbol{\omega}^*(\boldsymbol{\theta}_2)\| \\
&\le L_h\|\boldsymbol{\theta}_1 - \boldsymbol{\theta}_2\|.
\end{aligned}$$

Hence we have $h(O, \boldsymbol{\theta})$ is $L_h$-Lipschitz, where $L_h = U_\delta L_l + 2BL_v L_* + 4BU_\delta L_J$.

For term $\mathbb{E}_{\boldsymbol{\theta}}[h(O', \boldsymbol{\theta})]$, we have

$$
\begin{aligned}
&\|\mathbb{E}_{\boldsymbol{\theta}_1}[h(O', \boldsymbol{\theta}_1)] - \mathbb{E}_{\boldsymbol{\theta}_2}[h(O', \boldsymbol{\theta}_2)]\| \\
&\leq \|\mathbb{E}_{\boldsymbol{\theta}_1}[h(O', \boldsymbol{\theta}_1)] - \mathbb{E}_{\boldsymbol{\theta}_1}[h(O', \boldsymbol{\theta}_2)]\| + \|\mathbb{E}_{\boldsymbol{\theta}_1}[h(O', \boldsymbol{\theta}_2)] - \mathbb{E}_{\boldsymbol{\theta}_2}[h(O', \boldsymbol{\theta}_2)]\| \\
&\leq \mathbb{E}_{\boldsymbol{\theta}_1}[\|h(O', \boldsymbol{\theta}_1) - h(O', \boldsymbol{\theta}_2)\|] + \|\mathbb{E}_{\boldsymbol{\theta}_1}[h(O', \boldsymbol{\theta}_2)] - \mathbb{E}_{\boldsymbol{\theta}_2}[h(O', \boldsymbol{\theta}_2)]\| \\
&\leq L_h\|\boldsymbol{\theta}_1 - \boldsymbol{\theta}_2\| + \|\mathbb{E}_{\boldsymbol{\theta}_1}[h(O', \boldsymbol{\theta}_2)] - \mathbb{E}_{\boldsymbol{\theta}_2}[h(O', \boldsymbol{\theta}_2)]\| \\
&\leq L_h\|\boldsymbol{\theta}_1 - \boldsymbol{\theta}_2\| + 2BU_\delta d_{TV}(\mu_{\boldsymbol{\theta}_1} \otimes \pi_{\boldsymbol{\theta}_1}, \mu_{\boldsymbol{\theta}_2} \otimes \pi_{\boldsymbol{\theta}_2}) \\
&\leq (L_h + 2BU_\delta L_\pi (1 + \lceil \log_\rho C^{-1} \rceil + \frac{1}{1-\rho}))\|\boldsymbol{\theta}_1 - \boldsymbol{\theta}_2\| \\
&\leq (L_h + 2BU_\delta L_J)\|\boldsymbol{\theta}_1 - \boldsymbol{\theta}_2\| \\
&\leq 2L_h\|\boldsymbol{\theta}_1 - \boldsymbol{\theta}_2\|.
\end{aligned}
$$

Then we have $\boldsymbol{\omega} - \boldsymbol{\omega}_{\boldsymbol{\theta}}^*$ is $U_\delta$-bounded and $L_*$-Lipschitz; $\nabla \boldsymbol{\omega}_{\boldsymbol{\theta}}^*$ is $L_*$-bounded and $L_s$-Lipschitz; $\mathbb{E}_{\boldsymbol{\theta}}[h(O', \boldsymbol{\theta})] - h(O, \boldsymbol{\theta})$ is $2U_\delta B$-bounded and $3L_h$-Lipschitz. By the triangle inequality, we have

$$
\|\Xi(O, \boldsymbol{\omega}, \boldsymbol{\theta}_1) - \Xi(O, \boldsymbol{\omega}, \boldsymbol{\theta}_2)\| \leq (2U_\delta BL_*^2 + 2U_\delta^2 BL_s + 3U_\delta L_* L_h)\|\boldsymbol{\theta}_1 - \boldsymbol{\theta}_2\| \leq C_3\|\boldsymbol{\theta}_1 - \boldsymbol{\theta}_2\|,
$$

where $C_3 := 3U_\delta L_*(U_\delta L_l + 4BU_\delta L_J + 2BL_v L_*) + 2U_\delta BL_*^2 + 2U_\delta^2 BL_s$.

**Step 2:** show that

$$
\|\Xi(O, \boldsymbol{\omega}_1, \boldsymbol{\theta}) - \Xi(O, \boldsymbol{\omega}_2, \boldsymbol{\theta})\| \leq 2U_\delta BL_*\|\boldsymbol{\omega}_1 - \boldsymbol{\omega}_2\|. \tag{50}
$$

Actually, we have

$$
\begin{aligned}
\|\Xi(O, \boldsymbol{\omega}_1, \boldsymbol{\theta}) - \Xi(O, \boldsymbol{\omega}_2, \boldsymbol{\theta})\| &= \|\langle \boldsymbol{\omega}_1 - \boldsymbol{\omega}_2, (\nabla \boldsymbol{\omega}_{\boldsymbol{\theta}}^*)^\top \mathbb{E}_{O'}[h(O', \boldsymbol{\theta})] - h(O, \boldsymbol{\theta})\rangle\| \\
&\leq 2U_\delta BL_*\|\boldsymbol{\omega}_1 - \boldsymbol{\omega}_2\|.
\end{aligned}
$$

**Step 3:** show that for tuples $O_t = (s_t, a_t, s_{t+1})$ and $\widetilde{O}_t = (\widetilde{s}_t, \widetilde{a}_t, \widetilde{s}_{t+1})$. Conditioning on $s_{t-\tau+1}$ and $\boldsymbol{\theta}_{t-\tau}$, we have

$$
\mathbb{E}[\Xi(O_t, \boldsymbol{\omega}_{t-\tau}, \boldsymbol{\theta}_{t-\tau}) - \Xi(\widetilde{O}_t, \boldsymbol{\omega}_{t-\tau}, \boldsymbol{\theta}_{t-\tau})] \leq 2U_\delta^2 BL_\pi \sum_{k=t-\tau}^{t} \mathbb{E}\|\boldsymbol{\theta}_k - \boldsymbol{\theta}_{t-\tau}\|. \tag{51}
$$

By definition of $\Xi(O, \boldsymbol{\omega}, \boldsymbol{\theta})$, we have

$$
\begin{aligned}
&\|\mathbb{E}[\Xi(O_t, \boldsymbol{\omega}_{t-\tau}, \boldsymbol{\theta}_{t-\tau}) - \Xi(\widetilde{O}_t, \boldsymbol{\omega}_{t-\tau}, \boldsymbol{\theta}_{t-\tau})]\| \\
&= \|\mathbb{E}[\langle \boldsymbol{\omega}_{t-\tau} - \boldsymbol{\omega}_{t-\tau}^*, (\nabla \boldsymbol{\omega}_{t-\tau}^*)^\top (h(\widetilde{O}_t, \boldsymbol{\theta}_{t-\tau}) - h(O_t, \boldsymbol{\theta}_{t-\tau}))]\| \\
&\leq 4U_\delta^2 BL_* d_{TV}(\mathbb{P}(O_t \in \cdot | s_{t-\tau+1}, \boldsymbol{\theta}_{t-\tau}), \mathbb{P}(\widetilde{O}_t \in \cdot | s_{t-\tau+1}, \boldsymbol{\theta}_{t-\tau})),
\end{aligned} \tag{52}
$$

where the inequality comes from the definition of total variation distance. The total variation norm between $O_t$ and $\widetilde{O}_t$ has been computed in Eq. (43). Plugging Eq. (43) into Eq. (52), we get

$$
\begin{aligned}
\|\mathbb{E}[\Xi(O_t, \boldsymbol{\omega}_{t-\tau}, \boldsymbol{\theta}_{t-\tau}) - \Xi(\widetilde{O}_t, \boldsymbol{\omega}_{t-\tau}, \boldsymbol{\theta}_{t-\tau})]\| &\leq 2U_\delta^2 BL_* L_\pi \sum_{k=t-\tau}^{t} \mathbb{E}\|\boldsymbol{\theta}_k - \boldsymbol{\theta}_{t-\tau}\| \\
&\leq 2U_\delta^2 BL_* L_\pi G\tau(\tau+1)\alpha.
\end{aligned}
$$

**Step 4:** Show that conditioning on $s_{t-\tau+1}$ and $\boldsymbol{\theta}_{t-\tau}$, we have

$$
\|\mathbb{E}[\Xi(\widetilde{O}_t, \boldsymbol{\omega}_{t-\tau}, \boldsymbol{\theta}_{t-\tau})]\| \leq 4U_\delta^2 BC\rho^{\tau-1}. \tag{53}
$$

It can be shown that

$$
\begin{aligned}
\|\mathbb{E}[\Xi(\widetilde{O}_t, \boldsymbol{\omega}_{t-\tau}, \boldsymbol{\theta}_{t-\tau})]\| &\stackrel{(1)}{=} \|\mathbb{E}[\Xi(\widetilde{O}_t, \boldsymbol{\omega}_{t-\tau}, \boldsymbol{\theta}_{t-\tau}) - \Xi(O'_{t-\tau}, \boldsymbol{\omega}_{t-\tau}, \boldsymbol{\theta}_{t-\tau})]\| \\
&\stackrel{(2)}{\leq} 4U_\delta^2 BL_* d_{TV}(\mathbb{P}(\widetilde{O}_t \in \cdot | s_{t-\tau+1}, \boldsymbol{\theta}_{t-\tau}), \mu_{\boldsymbol{\theta}_{t-\tau}} \otimes \pi_{\boldsymbol{\theta}_{t-\tau}} \otimes \mathcal{P}),
\end{aligned}
$$

where (1) is due to the fact that $O'_t$ is from the stationary distribution which satisfies $\mathbb{E}[\Xi(O'_{t-\tau}, \boldsymbol{\omega}_{t-\tau}, \boldsymbol{\theta}_{t-\tau})|\boldsymbol{\theta}_{t-\tau}, s_{t-\tau+1}] = 0$ and (2) follows from the definition of total variation distance. From Assumption 4.6, we know that

$$d_{TV}(\mathbb{P}(\widetilde{s}_t \in \cdot), \mu_{\boldsymbol{\theta}_{t-\tau}}) \leq C\rho^{\tau-1}.$$

Therefore, we have

$$\begin{aligned}
\|\mathbb{E}[\Xi(\widetilde{O}_t, \boldsymbol{\omega}_{t-\tau}, \boldsymbol{\theta}_{t-\tau})]\| &\leq 4U_\delta^2 BL_* d_{TV}(\mathbb{P}(\widetilde{O}_t = \cdot|s_{t-\tau+1}, \boldsymbol{\theta}_{t-\tau}), \mu_{\boldsymbol{\theta}_{t-\tau}} \otimes \pi_{\boldsymbol{\theta}_{t-\tau}} \otimes \mathcal{P}) \\
&= 4U_\delta^2 BL_* d_{TV}(\mathbb{P}((\widetilde{s}_t, \widetilde{a}_t) \in \cdot|s_{t-\tau+1}, \boldsymbol{\theta}_{t-\tau}), \mu_{\boldsymbol{\theta}_{t-\tau}} \otimes \pi_{\boldsymbol{\theta}_{t-\tau}}) \\
&= 4U_\delta^2 BL_* d_{TV}(\mathbb{P}(\widetilde{s}_t = \cdot|s_{t-\tau+1}, \boldsymbol{\theta}_{t-\tau}), \mu_{\boldsymbol{\theta}_{t-\tau}}) \\
&\leq 4U_\delta^2 BL_* C\rho^{\tau-1}.
\end{aligned}$$

Combining Eq. (49)-Eq. (53), we can decompose the Markovian bias as

$$\begin{aligned}
\mathbb{E}[\Xi(O_t, \boldsymbol{\omega}_t, \boldsymbol{\theta}_t)] &= \mathbb{E}[\Xi(O_t, \boldsymbol{\omega}_t, \boldsymbol{\theta}_t) - \Xi(O_t, \boldsymbol{\omega}_t, \boldsymbol{\theta}_{t-\tau})] \\
&\quad + \mathbb{E}[\Xi(O_t, \boldsymbol{\omega}_t, \boldsymbol{\theta}_{t-\tau}) - \Xi(O_t, \boldsymbol{\omega}_{t-\tau}, \boldsymbol{\theta}_{t-\tau})] \\
&\quad + \mathbb{E}[\Xi(O_t, \boldsymbol{\omega}_{t-\tau}, \boldsymbol{\theta}_{t-\tau}) - \Xi(\widetilde{O}_t, \boldsymbol{\omega}_{t-\tau}, \boldsymbol{\theta}_{t-\tau})] \\
&\quad + \mathbb{E}[\Xi(\widetilde{O}_t, \boldsymbol{\omega}_{t-\tau}, \boldsymbol{\theta}_{t-\tau})] \\
&\leq C_3\|\boldsymbol{\theta}_t - \boldsymbol{\theta}_{t-\tau}\| + 2U_\delta BL_*\|\boldsymbol{\omega}_t - \boldsymbol{\omega}_{t-\tau}\| \\
&\quad + 2U_\delta^2 BL_* L_\pi G\tau(\tau+1)\alpha + 4U_\delta^2 BL_* C\rho^{\tau-1}.
\end{aligned}$$

Thus we conclude our proof. $\qquad\square$

**Proof of Lemma F.6.**

*Proof.* We will divide the proof of this lemma into three steps.

**Step 1:** show that

$$|\Theta(O, \boldsymbol{\theta}_1) - \Theta(O, \boldsymbol{\theta}_2)| \leq (2U_\delta BL_{J'} + 3L_J L_h)\|\boldsymbol{\theta}_1 - \boldsymbol{\theta}_2\|, \tag{54}$$

where $L_h = U_\delta L_l + 2BL_v L_* + 4BU_\delta L_J$ is defined in the proof of Lemma F.4.

Since $\Theta(O, \boldsymbol{\theta}) = \langle \nabla J(\boldsymbol{\theta}), \mathbb{E}_{O'_\theta}[h(O'_{\boldsymbol{\theta}}, \boldsymbol{\theta})] - h(O, \boldsymbol{\theta})\rangle$, we will show that each term in $\Theta(O, \boldsymbol{\theta})$ is Lipschitz.

For the term $\nabla J(\boldsymbol{\theta})$, we know it's $L_J$-bounded and $L_{J'}$-Lipschitz. For term $\mathbb{E}_{\boldsymbol{\theta}}[h(O', \boldsymbol{\theta})] - h(O, \boldsymbol{\theta})$, we have shown in the proof of Lemma F.4 that it's $2U_\delta B$-bounded and $3L_h$-Lipschitz. By the triangle inequality, we have

$$|\Theta(O, \boldsymbol{\theta}_1) - \Theta(O, \theta_2)| \leq (2U_\delta BL_{J'} + 3L_J L_h)\|\boldsymbol{\theta}_1 - \boldsymbol{\theta}_2\|$$

**Step 2:** show that conditioning on $s_{t-\tau+1}$ and $\boldsymbol{\theta}_{t-\tau}$, we have

$$|\mathbb{E}[\Theta(O_t, \boldsymbol{\theta}_{t-\tau}) - \Theta(\widetilde{O}_t, \boldsymbol{\theta}_{t-\tau})]| \leq 2U_\delta BL_J L_\pi \sum_{k=t-\tau}^{t} \|\boldsymbol{\theta}_k - \boldsymbol{\theta}_{t-\tau}\| \tag{55}$$

By definition of $\Theta(O, \boldsymbol{\theta})$, we have

$$\begin{aligned}
&|\mathbb{E}[\Theta(O_t, \boldsymbol{\theta}_{t-\tau}) - \Theta(\widetilde{O}_t, \boldsymbol{\theta}_{t-\tau})]| \\
&= |\mathbb{E}[\langle \nabla J(\boldsymbol{\theta}_{t-\tau}), h(\widetilde{O}_t, \boldsymbol{\theta}_{t-\tau}) - h(O_t, \boldsymbol{\theta}_{t-\tau})\rangle]| \\
&\leq 4U_\delta BL_J d_{TV}(\mathbb{P}(O_t \in \cdot|s_{t-\tau+1}, \boldsymbol{\theta}_{t-\tau}), \mathbb{P}(\widetilde{O}_t \in \cdot|s_{t-\tau+1}, \boldsymbol{\theta}_{t-\tau})), \tag{56}
\end{aligned}$$

where the inequality comes from the definition of total variation distance. The total variation distance between $O_t$ and $\widetilde{O}_t$ has been computed in Eq. (43). Plugging Eq. (43) into Eq. (56), we get

$$|\mathbb{E}[\Theta(O_t, \boldsymbol{\theta}_{t-\tau}) - \Theta(\widetilde{O}_t, \boldsymbol{\theta}_{t-\tau})]| \leq 2U_\delta BL_J L_\pi \sum_{k=t-\tau}^{t} \|\boldsymbol{\theta}_k - \boldsymbol{\theta}_{t-\tau}\|.$$

**Step 3:** show that conditioning on $s_{t-\tau+1}$ and $\boldsymbol{\theta}_{t-\tau}$, we have

$$|\mathbb{E}[\Theta(\widetilde{O}_t, \boldsymbol{\theta}_{t-\tau}) - \Theta(O'_{t-\tau}, \boldsymbol{\theta}_{t-\tau})]| \leq 4U_\delta BL_J C\rho^{\tau-1}. \tag{57}$$

From the definition of $\Theta(O, \boldsymbol{\theta})$, we have

$$\begin{aligned}
|\mathbb{E}[\Theta(\widetilde{O}_t, \boldsymbol{\theta}_{t-\tau}) - \Theta(O'_{t-\tau}, \boldsymbol{\theta}_{t-\tau})]| &= |\mathbb{E}[\langle \nabla J(\boldsymbol{\theta}_{t-\tau}), h(O'_t, \boldsymbol{\theta}_{t-\tau})\rangle - \langle \nabla J(\boldsymbol{\theta}_{t-\tau}), h(\widetilde{O}_t, \boldsymbol{\theta}_{t-\tau})\rangle]| \\
&\leq 4U_\delta BL_J d_{TV}(\mathbb{P}(\widetilde{O}_t \in \cdot | s_{t-\tau+1}, \boldsymbol{\theta}_{t-\tau}), \mu_{\boldsymbol{\theta}_{t-\tau}} \otimes \pi_{\boldsymbol{\theta}_{t-\tau}} \otimes \mathcal{P}) \\
&= 4U_\delta BL_J d_{TV}(\mathbb{P}((\widetilde{s}_t, \widetilde{a}_t) \in \cdot | s_{t-\tau+1}, \boldsymbol{\theta}_{t-\tau}), \mu_{\boldsymbol{\theta}_{t-\tau}} \otimes \pi_{\boldsymbol{\theta}_{t-\tau}}) \\
&= 4U_\delta BL_J d_{TV}(\mathbb{P}(\widetilde{s}_t \in \cdot | s_{t-\tau+1}, \boldsymbol{\theta}_{t-\tau}), \mu_{\boldsymbol{\theta}_{t-\tau}}) \\
&\leq 4U_\delta BL_J C\rho^{\tau-1},
\end{aligned}$$

where the last inequality follows from Assumption 4.6. Therefore, we have

$$|\mathbb{E}[\Theta(\widetilde{O}_t, \boldsymbol{\theta}_{t-\tau}) - \Theta(O'_{t-\tau}, \boldsymbol{\theta}_{t-\tau})]| \leq 4U_\delta BL_J C\rho^{\tau-1}.$$

Combining Eq. (54), Eq. (55), and Eq. (57), we can decompose the Markovian bias as

$$\begin{aligned}
\mathbb{E}[\Theta(O_t, \boldsymbol{\theta}_t)] &= \mathbb{E}[\Theta(O_t, \boldsymbol{\theta}_t) - \Theta(O_t, \boldsymbol{\theta}_{t-\tau})] \\
&\quad + \mathbb{E}[\Theta(O_t, \boldsymbol{\theta}_{t-\tau}) - \Theta(\widetilde{O}_t, \boldsymbol{\theta}_{t-\tau})] \\
&\quad + \mathbb{E}[\Theta(\widetilde{O}_t, \boldsymbol{\theta}_{t-\tau}) - \Theta(O'_{t-\tau}, \boldsymbol{\theta}_{t-\tau})] \\
&\quad + \mathbb{E}[\Theta(O'_{t-\tau}, \boldsymbol{\theta}_{t-\tau})],
\end{aligned}$$

where $\widetilde{O}_t$ is from the auxiliary Markovian chain defined in Eq. (8) and $O'_{t-\tau}$ is from the stationary distribution which satisfies $\mathbb{E}[\Theta(O'_{t-\tau}, \boldsymbol{\theta}_{t-\tau}) | \boldsymbol{\theta}_{t-\tau}] = 0$.

Then we have

$$\begin{aligned}
\mathbb{E}[\Theta(O_t, \boldsymbol{\theta}_t)] &\leq (2U_\delta BL_{J'} + 3L_J L_h)\mathbb{E}\|\boldsymbol{\theta}_t - \boldsymbol{\theta}_{t-\tau}\| \\
&\quad + 2U_\delta BL_J L_\pi \sum_{k=t-\tau}^{t} \|\boldsymbol{\theta}_k - \boldsymbol{\theta}_{t-\tau}\| + 4U_\delta BL_J C\rho^{\tau-1} \\
&\leq (2U_\delta BL_{J'} + 3L_J L_h) \sum_{k=t-\tau+1}^{t} \mathbb{E}\|\boldsymbol{\theta}_k - \boldsymbol{\theta}_{k-1}\| \\
&\quad + 2U_\delta BL_J L_\pi \sum_{k=t-\tau+1}^{t} \sum_{j=t-\tau+1}^{k} \mathbb{E}\|\boldsymbol{\theta}_j - \boldsymbol{\theta}_{j-1}\| + 4U_\delta BL_J C\rho^{\tau-1} \\
&\leq (2U_\delta BL_{J'} + 3L_J L_h) \sum_{k=t-\tau+1}^{t} \mathbb{E}\|\boldsymbol{\theta}_k - \boldsymbol{\theta}_{k-1}\| \\
&\quad + 2U_\delta BL_J L_\pi \tau \sum_{j=t-\tau+1}^{t} \mathbb{E}\|\boldsymbol{\theta}_j - \boldsymbol{\theta}_{j-1}\| + 4U_\delta BL_J C\rho^{\tau-1} \\
&\leq C_4(\tau+1) \sum_{k=t-\tau+1}^{t} \mathbb{E}\|\boldsymbol{\theta}_k - \boldsymbol{\theta}_{k-1}\| + C_5 C\rho^{\tau-1} \\
&\leq C_4(\tau+1)^2 G\alpha + C_5 C\rho^{\tau-1}
\end{aligned}$$

where $C_4 = \max\{2U_\delta BL_{J'} + 3L_J L_h, 2U_\delta BL_J L_\pi\}$ and $C_5 = 4U_\delta BL_J$. Substituting $L_h$ into $C_4$, we conclude the proof. $\qquad\square$

