# OpenReview forum: "Provable Convergence of Single-Timescale Neural Actor-Critic in Continuous Spaces"
_ICLR.cc/2025/Conference — Submitted to ICLR 2025_

### Official Review · Reviewer_FrEi · 2024-11-03

**Soundness:** 2
**Presentation:** 2
**Contribution:** 2
**Rating:** 3
**Confidence:** 4

**Summary:**

This paper provides a theoretical analysis of an actor critic-type algorithm in continuous spaces, and contributions are theoretical in nature.

**Strengths:**

- The paper is written well.
- The mathematical analysis is easy to read.

**Weaknesses:**

There are some weaknesses in the proposed approaches, as listed below:

1. Continuous spaces are important and challenging, but the analysis seems to make significant assumptions that are not realistic in practice. Assumption 4.7 seems unrealistic, and the authors have not discussed it in detail. I think this assumption is standard in tabular settings, where one can easily find an upper bound, but for continuous, it could be tricky.  It would be nice to give some realistic examples where this assumption would hold.

2. Given the existing analysis in the literature, because the authors are operating in parametrized settings, extending the analysis to continuous settings seems incremental and novelty is limited. it would be nice to further add the specific challenges that exist in extending the analysis from tabular to continuous settings and what efforts were made to deal with such challenges. The proof is almost similar to the following paper, with only some additional generalizations of the assumptions.

Chen X, Zhao L. Finite-time analysis of single-timescale actor-critic. Advances in Neural Information Processing Systems. 2024 Feb 13;36.

3. Also assumption 4.6 also requires discussions; the papers authors have cited are mostly in tabular settings. Is it valid to assume the same assumption for continuous spaces as well?

4. How assumption 4.5 is related to exploration is not clear. Is there any scenarios where this can hold?

5. Discussing all these assumptions is important because the paper's motivation is to study the analysis in the most practical settings; if the assumptions are not realistic, that would defeat the purpose.

**Questions:**

N/A

---

> ### Author Response · Authors · 2024-11-21
> **Authors Response--Part I**
>
> **(Q1: on Assumption 4.7)**
>
> Thank you for seeking clarification. Compared to the standard tabular case, we proposed a more general condition in Condition (c) to handle continuous policies. It holds for a broad class of policy functions as proved in Proposition 4.8. Conditions (a) and (b) can be easily verified through direct computation. For example, Assumption 4.7 can be satisfied by truncated Gaussian distributions with bounded mean values parameterized by MLPs or Transformers, and we adopt this parameterization for the policy in our newly added experiment section. Moreover, a uniform distribution with an interval length $\theta > 0$ also fulfills Assumption 4.7.
>
> **(Q2: on the novelty of the analysis)**
>
> Our novelty lies in the following: (1) we formulate a general condition in Assumption 4.7(c) and show that it is satisfied by a broad class of neural network policies in continuous spaces, as detailed in Proposition 4.8, which plays a crucial role in the proof. (2) We developed new operators to efficiently handle the complexities of continuous settings. (3) To address Markovian noise in continuous settings, we proved a sharp bound on the distance between stationary distributions, as characterized by Lemma C.1. These techniques have never been developed in the single-timescale actor-critic framework. They are the enablers for the analysis of the continuous state and action spaces.
>
> We would like to highlight that our work extends beyond simply combining existing methods. In fact, (Chen\&Zhao, 2024) only considers the learning with the linear function approximation. This strong assumption greatly simplifies the assumptions and transfers the dynamic analysis of the actor and critic to the corresponding simple matrix analysis. (Tian et al., 2024) focuses on the neural network learning problem in the discounted case and i.i.d. sampling of the actor. We note that the discounted problem greatly relieves the rate analysis of the critic analysis due to the well-defined Bellman equation in this case. In contrast, our work considers the most general setting. The convergence analysis of the neural network is coupled with the non-i.i.d. sampling and the additional reward estimation error from the Bellman equation of the time-average MDPs. We are the first to integrate neural network estimation analysis into the coupling relationship between three error sources, including the reward estimation error, critic error, and actor error defined in Eq. (12). Moreover, we established sufficient conditions for convergence over an infinite action space, a result not previously achieved. Our work bridges the gap between theoretical analysis and practical applications.

---

> ### Author Response · Authors · 2024-11-21
> **Authors Response--Part II**
>
> **(Q3: on Assumption 4.6)**
>
> Assumption 4.6 states that the induced Markov chain is geometrically mixing, which holds for both finite and continuous cases. To justify this assumption in the continuous space, we note that all the distributions specified by the Ornstein–Uhlenbeck process satisfy this property. The OU process can be defined as follows
> \begin{align*}
>     dX_{t}=\theta_{\pi}(\mu_{\pi}-X_t)dt+\sigma d W_t,
> \end{align*}
> where $\theta_{\pi}>0$ and the drift $\mu_{\pi}$ are induced by the policy $\pi$. This process converges to a Gaussian distribution with the exponential mixing time. Moreover, it can also be shown that this property holds for more general diffusion processes[1].
>
> **(Q4: on Assumption 4.5)**
>
> This assumption was first introduced by us for the continuous setting with general function approximation classes.  To demonstrate its connection to exploration, we show that if exploration is insufficient, the assumption fails to hold. Consequently, when the assumption holds, it implies sufficient exploration. First note that the operator $D_{\theta}$ essentially multiplies the stationary distribution $\mu_{\theta}$ to the function on its right (see the definition in Eq. (1) in the manuscript). If the policy $\pi_{\theta}$ does not sufficiently explore, there exists a subset of the state space $U \subset \mathcal{S}$ such that $\mu_{\theta}(U)=0$. Furthermore, we can choose $\widehat{V}(\omega)$ such that $\widehat{V}(\omega;s) = 0, \forall s \in \mathcal{S}\setminus U$ and $\widehat{V}(\omega;s) \ge 0, \forall s \in U$. With this choice, the left-hand side of the inequality evaluates to 0, while the right-hand side becomes positive. This violates the condition stated in Assumption 4.5. Thus, the contrapositive holds: if Assumption 4.5 is satisfied, it ensures sufficient exploration of the state space under the policy $\pi_\theta$.
>
> Note that sufficient exploration assumption is standard in the literature of analyzing the convergence of on-policy RL algorithms. We can also drop this condition by analyzing the off-policy version of the algorithm under some sufficiently-exploring behavior policy that can be arbitrarily specified, and relates to the target policy by importance sampling. However, this is not the core focus of the problem. Therefore, we adopt Assumption 4.5 directly and concentrate on the primary challenge of analyzing the algorithm in the continuous state-action space.
>
> Thanks for taking your time to review our paper. We are looking forward to your reply and we would be delighted to address any further concerns during the remaining discussion time.
>
> [1] Pierre Del Moral and Denis Villemonais. Exponential mixing properties for time inhomogeneous diffusion processes with killing. 2018

---

> > ### Comment · Reviewer_FrEi · 2024-12-01
> >
> > Thank you for your response. I have checked them in detail, and I would like to keep my score. After reading the detailed response, I realized that the contributions are incremental and do not completely justify the motivation to study the theoretical insights of a "practical" algorithm. As shown in the examples, the assumption does not hold for the most straightforward Gaussian parameterization, which is the most practical parametrization used in practice.

---

> > > ### Author Response · Authors · 2024-12-01
> > >
> > > 1. The Truncated Gaussian is utilized because, in the pendulum environment, the action space is constrained to the range [-2,2], necessitating the truncation of the Gaussian distribution to align with this boundary. In our newly added Assumption 4.7(c), we will show the following proof that the Gaussian policy is satisfied.
> > >
> > > We demonstrate that two multivariate Gaussians, with mean values parameterized by two neural networks ($\theta_1$) and ($\theta_2$), can also satisfy Assumption 4.7(c).
> > >
> > > Let these two multivariate Gaussians be denoted as $\mathcal{N}_{\theta_1} := \mathcal{N}(\theta_1, \Sigma)$ and $\mathcal{N} _{\theta_2} := \mathcal{N}(\theta_2, \Sigma)$. The KL divergence between them can be expressed as:
> > >
> > > \begin{align*}
> > > D _{KL}(\mathcal{N} _{\theta _1}||\mathcal{N} _{\theta _2})=&\int\mathcal{N} _{\theta _1}(x)\times[-\frac{1}{2}(x-\theta _1)^\top\Sigma^{-1}(x-\theta _1)+\frac{1}{2}(x-\theta _2)^\top\Sigma^{-1}(x-\theta _2)]dx\\\ =&-\frac{1}{2}{\rm tr}(\mathbb{E}[(x-\theta _1)(x-\theta _1)^\top]\Sigma^{-1})+\frac{1}{2}\mathbb{E}[(x-\theta _2)^\top\Sigma^{-1}(x-\theta _2)]\\\ =& \frac{1}{2}(\theta _1-\theta _2)^\top \Sigma^{-1}(\theta _1-\theta _2)-\frac{1}{2}{\rm tr(\Sigma^{-1}\Sigma)}+\frac{1}{2}{\rm tr(\Sigma^{-1}\Sigma)}\\\ \leq & \frac{1}{2\lambda _{\rm min}}\|\theta _1-\theta _2\|^2. \end{align*} Therefore, from Pinsker's inequality, we have \begin{align*} d _{TV}(\mathcal{N} _{\theta_1},\mathcal{N} _{\theta_2})\leq &\sqrt{\frac{1}{2}D _{KL}(\mathcal{N} _{\theta_1}||\mathcal{N} _{\theta_2})}\\\ \leq & \frac{1}{2\sqrt{\lambda _{\rm min}}}\|\theta_1-\theta_2\|,
> > > \end{align*}
> > > which shows Gaussian distribution satisfies Assumption 4.7 (c).
> > >
> > > 2. We would like to highlight that there is no infinite magnitude control signal in the realistic applications, it is truncated due to physical limitations of devices. The Gaussian distribution is rather an idealized approximation for these practical control-limited settings.
> > >
> > > 3. In fact, truncated Gaussian is widely used for continuous control in practical applications [1].
> > >
> > > 4. Constrained control poses greater challenges compared to unconstrained control due to its increased system nonlinearity, a longstanding topic of interest in both the learning and control communities. Incorporating truncated Gaussian distributions enhances the practicality of the analysis, and bridges the gap between theory and application.
> > >
> > > 5. More importantly, the analytical gap between the standard Gaussian and its truncated counterpart is trivial— the same asymptotic results holding with high probability can be easily obtained by some standard concentration analysis.
> > >
> > > We sincerely hope that the reviewer's decision will be based on a more comprehensive evaluation, rather than being disproportionately influenced by small and trivial details.
> > >
> > > [1] Silver, David, et al. "Deterministic policy gradient algorithms." International conference on machine learning. Pmlr, 2014.

---

### Official Review · Reviewer_rt7P · 2024-11-03

**Soundness:** 3
**Presentation:** 3
**Contribution:** 3
**Rating:** 5
**Confidence:** 3

**Summary:**

The authors analyze single time scale actor critic algorithms for continuous state and action space control problems. The results show that the convergence to a stable point at a rate that depends on the number of iterates and the width of the controlling network.

**Strengths:**

Strong results for an important problem.

Considering Markovian sampling makes the result even stronger.

The assumptions are explained very clearly. This is commendable as there are many assumptions ....

**Weaknesses:**

It is not clear how $c$ is chosen for THM 4.9 to hold. This seems to be something the algorithm would have to know a-priori. But how can that be? This seems like a pretty complex constant AFAIU.

I don't understand why one would not take $m \to \infty$: in that case the second term is 0, but wouldn't neural network convergence issues emerge? I mean, this is something that should be reflected in the bound somehow. Or am I missing something?

I would appreciate to see a proof outline. Especially, pointing out to the novel elements of the analysis is crucial. Since the paper is really an exposition to the appendix where the proofs are, it is rather hard to follow where the real novelty is.

Finally, if there is any empirical evidence for the (low) dependence on $m$ empirically for large $m$, this would go a long way to convincing me in the utility of the paper.

I'd be happy to raise my score, but I would like to see detailed discussion of the above.

**Questions:**

Please see weaknesses above.

---

> ### Author Response · Authors · 2024-11-21
>
> **(W1: on the choice of $c$)**
>
> For THM 3.9 to hold, it is sufficient to set $c$ smaller than a problem-dependent constant, as specified in Eq. (34) of our revised manuscript (Eq. (30) of the original submission). The complex constant is only an upper bound of $c$, and can be easily satisfied by choosing $c$ a sufficiently small positive number, as is commonly done in practice. Intuitively, it indicates that a small stepsize will be conducive to stable learning. In our experiment, we set $c=1$, which produced satisfactory simulation results.
>
> **(W2: on the dependence of $m$)**
>
> Indeed, the neural network approximation error, denoted by $ \epsilon_{\rm app} $, also depends on the neural network width $ m $. If  $ m $ is too large, the neural network's approximation ability will deteriorate. However, deriving a more analytical bound for $\epsilon_{\rm app} $ in terms of  $ m $ remains an open problem. Our results are better interpreted as characterizing the analytical convergence rate within the range where the increase of  $ m $ does not significantly increase $ \epsilon_{\rm app} $.  Moreover, we empirically show the impact of $ m $ through a newly added inverted pendulum example (see Section 5). It demonstrates that the initial increase of $ m $ (for $m\le 300 $) reduces the learning error, but worsens the performance when $m$ is too large ($ m > 300 $).
>
> **(W3: on the novelty of the proof)**
>
> Thank you for your advice. We provided a proof sketch in our updated manuscript in Appendix. We highlight the new
> techniques developed in detail and labeled in bold for your reference. We would like to summarize our novelty in the following: (1) we formulate a general condition in Assumption 4.7(c) and show that it is satisfied by a broad class of neural network policies in continuous spaces, as detailed in Proposition 4.8, which plays a crucial role in the proof. (2) We developed new operators to efficiently handle the complexities of continuous settings. (3) To address Markovian noise in continuous settings, we proved a sharp bound on the distance between stationary distributions, as characterized by Lemma C.1. These techniques have never been developed in the single-timescale actor-critic framework. They are the enablers for the analysis of the continuous state and action spaces.
>
> **(W4: on the experiment)**
>
> Thanks for your suggestion. As already mentioned in the above, we experimented with the classic "Pendulum" benchmark environment to validate our theoretical findings (Section 5). This environment features continuous state and action spaces. Both the critic and actor are parameterized using the neural network structure defined in Eq. (5) of our paper. We adopt a step size of $5 \times 10^{-6}$ for both the critic and actor, which is a single-timescale setting. The results demonstrate the convergence of this single-timescale neural actor-critic algorithm in continuous spaces when the neural network width $m$ is sufficiently large ($m\ge 200$). We hope this experiment is illustrative.

---

> > ### Comment · Reviewer_rt7P · 2024-12-01
> > **Thank you for your response.**
> >
> > Thank you for your response. While the paper is pretty clear in terms of contribution, I am still not convinced this really matters for a "real' problem. The dependence of $\epsilon_{app}$ in $m$ is not clear. Moreover, I think that the experiments are not convincing enough.
> >
> > I appreciate the authors response, but will keep my score.

---

> > > ### Author Response · Authors · 2024-12-01
> > >
> > > 1. Thanks for recognizing our contribution.
> > > 2. Our work represents a major step toward understanding practical AC algorithms, which cannot be achieved through existing analyses.
> > > 3. We contend that the term $\epsilon_{\rm app}$ represents a classical bias term, which consistently appears in all the relevant papers cited in Table 1.
> > > 4. Could you please clarify which specific aspects of our simulation you find unconvincing? We would appreciate concrete feedback.

---

> > > > ### Comment · Reviewer_rt7P · 2024-12-01
> > > >
> > > > I do not understand how the experiments support and O(1/\sqrt{m}) dependence as claimed.
> > > > In general, a much more thorough experimentation is needed to support such claims - larger networks, other problems, etc. etc.

---

> ### Author Response · Authors · 2024-12-01
>
> Thanks for your response. Note that this rate is only an upper bound. The actual convergence rate in terms of $m$ can be faster. Meanwhile, the convergence rate is also strongly affected by various randomness in the learning process, such as the sampling of the actions. Therefore, experiments show exactly match of the order is difficult, if not impossible. Nevertheless, we are able to demonstrate that the learning error decreases as m increases (for $m \in (10, 400)$) . This relationship is well captured by our theorectical results.

---

> > ### Comment · Reviewer_rt7P · 2024-12-01
> >
> > I do not think that a relationship of \sqrt{m}^-1 is captured. Perhaps this is the right relationship, and perhaps it is not.

---

### Official Review · Reviewer_7fyF · 2024-11-04

**Soundness:** 3
**Presentation:** 4
**Contribution:** 3
**Rating:** 5
**Confidence:** 3

**Summary:**

The authors consider the convergence of single time scale actor critic algorithm in continuous state and action spaces with infinite horizon average reward as the performance metric. The state and action spaces considered are assumed to be compact and the value functions and the policies are parametrized through deep neural networks. The average reward, the relative value function and the policy are all updated on a similar time scale. Finite time convergence guarantees to a stationary point are provided under the overparametrized neural network regime. They resort to a NTK style analysis where by the virtue of overparametrization, the weights do not change much in magnitude by the end of the training period.

**Strengths:**

1. Most prior literature in RL (especially average reward RL) considers finite state and action spaces. Hence the setting of continuous state and action spaces is relatively understudied given its significance.
2. Neural networks (NN) have consistently demonstrated strong capabilities as function approximators, with prior results showing that they can approximate continuous functions to any desired accuracy. However, their theoretical foundations remain relatively underexplored. This paper addresses a crucial problem, particularly given the extensive practical applications of neural networks.
3. They characterize tight bounds in terms of dependence on the number of iterations and number of parameters of the NN.

**Weaknesses:**

1. Since this paper is motivated by the promise of neural networks in policy optimization in the realm of RL, simulations demonstrating the same would increase the strength of the paper. As of now, the paper analyzes the standard actor critic algorithm and provides finite time bounds, but relating these to whats observed in practice would vastly help with understanding (especially in terms of determining the various step sizes required for updating different quantities of interest).
2. The paper provides convergence to a stationary point which can be a local minima. However, some of the recent works have proposed global convergence guarantees for average reward problems. See ref below. Although these are for finite state and action spaces, the optimization landscape might be the same independent of the finiteness of the problem parameters.
3. The assumptions require the nonlinear activations to be differentiable and smooth, this precludes the use of ReLU. However the authors suggest other nonlinear activations which indeed satisfy these assumptions. Once again, a simulation to demonstrate their practicality would help.
4. Regarding Assumption 4.3: Does this assumption require the original value functions to also be smooth with respect to the underlying parameter $\theta$. Since the parameters are initialized from a normal distribution, it is not clear as to how assumption 4.3 can be satisfied in practice (the uniform boundedness of the optimal parameters corresponding to the value function estimate). In case of linear function approximations, this assumption is trivial since typically $\|\phi\| \leq 1$, where $\phi$ is the feature vector. Its not clear whether this assumption can be satisfied in a straight forward manner when using neural networks.
5. Regarding Assumption 4.4: Since this assumption holds for every state, when dealing with uncountably infinite states, it is unclear as to how $\lambda_1$ can be uniformly bounded from below across all states.




[1] Improved Sample Complexity for Global Convergence of Actor-Critic Algorithms
N Kumar, P Agrawal, G Ramponi, KY Levy, S Mannor 2024
[2] On the global convergence of policy gradient in average reward markov decision processes
N Kumar, Y Murthy, I Shufaro, KY Levy, R Srikant, S Mannor 2024

**Questions:**

1. Equation 1, what does the measure $d(a\times s')$ represent?
2. Theorem 4.9 $\alpha = \frac{c}{\sqrt{T}}$, what are the typical values of $c$ in this learning rate?

---

> ### Author Response · Authors · 2024-11-21
>
> **(W1: add experiment)**
>
> Thanks for your suggestion. We added an experiment using the classic "Pendulum" benchmark to validate our theoretical findings (Section 5). This environment features continuous state and action spaces, aligning well with the focus of our study. Both the critic and actor are parameterized using the neural network structure defined in Eq. (5) of our paper. We adopt step sizes of $5 \times 10^{-6}$ for both the critic and actor, resulting in a single-timescale setting. The results demonstrate the convergence of the considered single-timescale neural actor-critic algorithm in continuous spaces when the neural network width $m$ is sufficiently large ($m\ge 200$). This experiment aims to illustrate the practical efficacy of the algorithm and support our theoretical results.
>
> **(W2: on the global convergence)**
>
> Thanks for pointing out the two relevant works. The provided reference [1] additionally assumes strong conditions on the objective function -- the well-known gradient domination lemma, which directly guarantees global convergence when the gradient approaches zero and simplifies the analysis. However, this property does not hold in general. For instance, one prerequisite for the gradient domination lemma is that the initial distribution must be strictly positive across all states---a condition that is unmet in our continuous setting.
> The provided reference [2] only considers the vanilla policy gradient algorithm, which is not of the actor-critic structure and does not parameterize and estimate the critic. Therefore, their optimization landscapes are significantly different from our considered problem.
>
> **(W3: on the activation function)**
>
> Thanks for your advice. To illustrate the practicality, we experiment with the tanh activation function which is nonlinear and satisfies the smoothness assumption.
>
>
> **(W4: on Assumption 4.3)**
>
> Assumption 4.3 does not require the original value functions to be smooth with respect to the underlying parameter. Meanwhile, we would like to clarify that Assumption 4.3 only assumes the Lipschitz continuity and Lipschitz continuous gradient of the optimal critic parameter with respect to the actor parameter, and does not require uniform boundedness of the optimal parameters.
>
> **(W5: on Assumption 4.4)**
> Thank you for your careful reading. This was a typo. Our proof does not require this condition to hold for all individual states; it only requires the norm defined over the state space to hold (that is, independent of any individual state). We have corrected this in the manuscript.
>
>
> **(Q1: on the measure $d(a \times s')$)**
>
> In Eq. (1), the measure $d(a \times s')$ represents the joint measure defined on the space of actions $ a $ and subsequent states $ s'$. It is used exchangeably with  $dads'$.
>
>
> **(Q2: on the type value of $c$)**
>
> In the theoretical analysis, $c$ is set to be smaller than a problem-dependent constant, as specified in Eq. (34) of our revised manuscript (Eq. (30) of the original submission). In our experiment, we set $c=1$, which produced satisfactory simulation results.

---

### Official Review · Reviewer_QDLr · 2024-11-04

**Soundness:** 4
**Presentation:** 3
**Contribution:** 4
**Rating:** 6
**Confidence:** 4

**Summary:**

This paper proves a novel result for average reward MDP's. Prior convergence results for actor critic for average reward were only able to prove convergence for a finite state action spaces. This paper is the first result of its kind and thus deserves acceptance in my opinion. I do however have some concerns.

**Strengths:**

The paper is well laid out and easily readable. The appendix has been arranged in a way that makes it easy to follow. The biggest strength of the paper is the novel contributions that it has.

**Weaknesses:**

1. Even though the results proved in the paper are novel, the proof seemed quite derivative of the paper Chen 2024.


References:


Xuyang Chen and Lin Zhao. Finite-time analysis of single-timescale actor-critic. Advances in Neural Information Processing Systems, 36, 2024.

**Questions:**

1. I have a concern about the assumption 4.5. Specifically the term ${\mu_{\theta}}{\int_{\mathcal{S}{\times}{\mathcal{A}}}}(\pi_{\theta}-1)\mathcal{P}(s^{'}|s,a)d(a{\times}s^{'})$. Since the term $(\pi_{\theta}-1)$ for all state action pairs, is this term not always negative? This would make the assumption 4.5 not true. It is likely I am misunderstanding the notations, so some clarity here would be helpful.
2.  On page 24. In the second left hand side for $Z_{T}$. I am having trouble figuring out where the $\mathcal{O}\left(\frac{1}{\sqrt{m}}\right)$ has come from.  It would be helpful if some information would be given in this regard.

---

> ### Author Response · Authors · 2024-11-21
>
> **(Q1: on the term $\pi_{\theta}-1$)**
>
> Thank you for the careful reading. It is indeed a typo, which should be $1-\pi_\theta$ as can be seen from our proof in Line 1178. We have updated Assumption 4.5 and labeled it with red color for your reference in the newly submitted revised manuscript. Sorry for the confusion caused and thanks for pointing out the issue.
>
> **(Q2: on the term $\widetilde{\mathcal{O}}(\frac{1}{\sqrt{m}})$)**
>
> Thanks for seeking clarification. The term $\widetilde{\mathcal{O}}(\frac{1}{\sqrt{m}})$ comes from the $H_v$-Lipschitz continuous gradient $H_v=\widetilde{\mathcal{O}}(\frac{1}{\sqrt{m}})$ of the critic neural network parameterization (see Lemma C.5, Page 16 of the revised manuscript). We have provided additional explanation of this derivation on Page 26 of the revised manuscript to enhance readability.

---

> > ### Comment · Reviewer_QDLr · 2024-11-26
> > **Response to Authors**
> >
> > Thank you for the clarification. Based on your response I will keep my score.

---

> > > ### Author Response · Authors · 2024-11-26
> > >
> > > Thanks for your reply and recognizing our contribution!

---

### Author Response · Authors · 2024-11-21
**Responses to all Reviewers**

Thank you for taking the time to review our paper! Based on your valuable suggestions, (1) we conducted experiments and added a section to explain the results in the updated manuscript. We hope it can better illustrate our theoretical results on the convergence of the proposed single-timescale neural actor-critic algorithm in continuous spaces. (2) We also have included a proof sketch in the Appendix to emphasize the merits of our analysis and highlight the novel contributions we have developed. (3) Furthermore, we have provided detailed justifications of our assumptions, with all changes marked in red in the revised manuscript for your convenience to review.

We are looking forward to your feedback and would be delighted to address any further concerns during the remaining discussion period.

---

### Meta-Review · Area_Chair_Ywwd · 2024-12-09

**Metareview:**

The paper gives a finite sample analysis for single time-scale actor critic with neural networks in continuous state action spaces. The finite  sample analysis appears correct. However, the reviewers are concerned about the motivation and the technical novelty of the analysis. In particular, the analysis seems derivative from Chen & Zhao (2024) and Tian et al. (2024). The way this paper handles neural networks seem to be derivative from Tian et al (2024) since A4.3 and A4.4 are both from Tian et al (2024). Infinite state space is already studied in Chen & Zhao (2024) so the paper would greatly benefit from additional clarification about whether and why infinite action space poses additional challenges (other than simply require a generalization of the previous assumption). The motivation should also be significantly toned down, since the analyzed algorithm is far from being practical and the assumptions are far from being practical (e.g., A5 does not hold even with a tabular representation since $D(I-P)$ is only positive semidefinite).

**Additional Comments On Reviewer Discussion:**

The reviewers are concerned about the motivation and the novelty of the technical analysis. The authors revised the paper but still did not convince the reviewers (and me). The only positive reviewer is unable or unwilling to argue for acceptance. I read the paper and agree with the reviewers and believe the assumptions are quite strong, in particular, see my comment about A5 in meta review.

---

### Decision · Program_Chairs · 2025-01-22

Reject